 Select

# Dual dressed black holes as the end point of the charged superradiant instability in $\mathcal{N} = 4$ Yang Mills

Sunjin Choi[1,2]⋆, Diksha Jain[3]†, Seok Kim[4]‡, Vineeth Krishna[3,5]∘,
Eunwoo Lee[3,4]§, Shiraz Minwalla[3]¶ and Chintan Patel[3,6]∥

**1** School of Physics, Korea Institute for Advanced Study,
85 Hoegi-ro, Dongdaemun-gu, Seoul 02455, Republic of Korea
**2** Kavli Institute for the Physics and Mathematics of the Universe (WPI),
The University of Tokyo Institutes for Advanced Study,
The University of Tokyo, Kashiwa, Chiba 277-8583, Japan
**3** Department of Theoretical Physics, Tata Institute of Fundamental Research,
Homi Bhabha Rd, Mumbai 400005, India
**4** Department of Physics and Astronomy & Center for Theoretical Physics,
Seoul National University, 1 Gwanak-ro, Gwanak-gu,
Seoul 08826, Republic of Korea
**5** Leinweber Center for Theoretical Physics, University of Michigan,
Ann Arbor, MI 48109, USA
**6** Kavli Institute for Theoretical Physics, Kohn Hall,
Santa Barbara, CA 93106, USA

⋆ sunjin.choi@ipmu.jp , † diksha.2012jain@gmail.com , ‡ seokkimseok@gmail.com ,
∘ vineethbannu@gmail.com , § eunwoo.lee@tifr.res.in ,
¶ minwalla@theory.tifr.res.in , ∥ chintan.patel@tifr.res.in

## Abstract

Charged Black holes in $AdS_5 \times S^5$ suffer from superradiant instabilities over a range of energies. Hairy black hole solutions (constructed within gauged supergravity) have previously been proposed as endpoints to this instability. We demonstrate that these hairy black holes are themselves unstable to the emission of large dual giant gravitons. We propose that the endpoint to this instability is given by Dual Dressed Black Holes (DDBH)s; configurations consisting of one, two, or three very large dual giant gravitons surrounding a core $AdS$ black hole with one, two, or three $SO(6)$ chemical potentials equal to unity. The dual giants each live at $AdS$ radial coordinates of order $\sqrt{N}$ and each carry charge of order $N^2$. The large separation makes DDBHs a very weakly interacting mix of their components and allows for a simple computation of their thermodynamics. We conjecture that DDBHs dominate the phase diagram of $\mathcal{N} = 4$ Yang-Mills over a range of energies around the BPS plane, and provide an explicit construction of this phase diagram, briefly discussing the interplay with supersymmetry. We develop the quantum description of dual giants around black hole backgrounds and explicitly verify that DDBHs are stable to potential tunneling instabilities, precisely when the chemical potentials of the core black holes equal unity. We also construct the 10-dimensional DDBH bulk solutions.

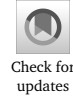

# Contents

# 1   Introduction

Over 15 years ago Gubser pointed out [1] that charged black holes in AdS spacetime are sometimes unstable to the condensation of charged fields. These instabilities were investigated in detail in several 'bottom up' models (such as Einstein Maxwell cosmological constant theory with a single charged scalar field), and their endpoints were demonstrated to be 'hairy' black holes: black holes that live inside the sea of charge condensate [2, 3] and usually interact strongly with it. Hairy black holes solutions are complicated, and have typically been constructed numerically.[1]

The study of charge instabilities in top-down models (i.e. in bulk theories that arise as the dual description of field theories with a known dual description) has received less attention. Familiar examples of such bulk theories are $AdS \times M$ compactifications of 10 or 11-dimensional supergravity. In such models, the $AdS$ gauge fields arise (via the Kaluza-Klein (KK) mechanism) from isometries of the internal manifold $M$. Charged fields arise from modes that carry 'angular momentum' on $M$. As the spectrum of this 'angular momentum' is unbounded, the effective AdS theory hosts an infinite number of charged fields. It is natural to wonder whether this new feature qualitatively changes the nature of the charge condensation phenomenon in such top-down models.[2] In this paper we address this question in the context of a particular top down model: IIB string theory on $AdS_5 \times S^5$, i.e. the bulk dual of $\mathcal{N} = 4$ Yang Mills (SYM) theory at strong coupling. In this theory the various KK modes on $S^5$ are dual to half BPS single trace operators of SYM theories, i.e. to the various superconformal descendants of $O_m^{(I_1 \dots I_m)} = \text{Tr}(X^{(I_1} X^{I_2} \dots X^{I_m)})$ for $m = 2, 3, \dots \infty$.[3] Here we investigate the impact that charged fields at large $m$ have on the charge condensation phenomenon in the bulk dual of $\mathcal{N} = 4$ SYM.

Previous studies of charge condensation in $\mathcal{N} = 4$ SYM have been performed within a consistent truncation of IIB SUGRA - namely gauged supergravity. This consistent truncation gov-

---

[1]However, hairy black holes simplify at small charges and energies. At such charges, black holes are much smaller than the size of the charge cloud (whose size is set by the radius of $AdS$). Form factor effects then ensure that the two components in these solutions are effectively non-interacting at leading order. This non-interacting picture receives corrections at higher orders in a power series expansion in charges. See, e.g. [4–7].

[2]Recall that the infinite number of potentially unstable modes plays a crucial (and, paradoxically simplifying) role in the analysis of the *angular momentum* version of the charged instability [8].

[3]Here $X^I$, $I = 1 \dots 6$ are the six scalar fields of $\mathcal{N} = 4$ SYM, and the brackets $^{()}$ denote traceless symmetrization. Superconformal descendants of $\text{Tr}(X^{(I_1} X^{I_2)})$ include both $SO(6)$ currents (dual to bulk $SO(6)$ gauge bosons), 42 charged scalar operators (which transform in the $\mathbf{20} + \mathbf{10} + \bar{\mathbf{10}} + \mathbf{1} + \mathbf{1}$ of $SO(6)$) as well as the stress tensor (dual to the graviton). For $m \geq 3$, superconformal multiplets include several charged fields in increasingly large representations of $SO(6)$. In particular the primaries themselves transform in the $SO(6)$ representation with $m$ boxes in the first row of the Young Tableaux.

erns the nonlinear dynamics of the fields dual to the operators $O_2^{(I_1 I_2)}$ (and their superconformal descendants) while setting the fields dual to $O_m^{(I_1 \dots I_m)}$ and descendants ($m = 3 \dots \infty$) to zero. Thus gauged supergravity contains only a 'single' Kaluza-Klein mode, and charge condensation in this model is qualitatively similar to that in bottom-up models [5, 7, 9, 10]. In particular, the end point of the charged condensation instability is a hairy black hole that is generically complicated[4] and numerically constructed [9, 10] solution. The authors of [5, 7, 9, 10] proposed that the hairy black hole phase dominates the microcanonical ensemble (of strongly coupled $\mathcal{N} = 4$ SYM) at mass to charge ratios near to the BPS bound.[5] Notice, however, that as the analysis of [5, 7, 9, 10] is performed within gauged supergravity, it sidesteps the question of the impact of the infinite number of charged fields on charge condensation. In particular, it is silent on the question of whether the hairy solutions constructed in [5, 7, 9, 10] are themselves unstable to the emission of fields dual to $O_m^{I_1 \dots I_m}$ at large $m$. The starting point of the current paper is a simple argument that this is indeed the case, though the time scales associated with this process may turn out to be very large.

Our argument uses dual giant gravitons [11, 12]. Recall that dual giants of charge $H$[6] are solutions of the probe D3-brane theory that puff up in $AdS_5$ with a radius (measured as the value of the usual radial $AdS$ coordinate) given by $\sqrt{\frac{H}{N}}$. While dual giant solutions were initially found in pure $AdS$ space, similar solutions have also been constructed in the background charged black holes in $AdS$ [13, 14]. When $H/N$ is large, the dual giants live far from the black hole. At these values of charges their properties are thus essentially identical to those of dual giants in pure AdS space. In particular, their energy $E$ is equals their charge $H$[7] up to corrections that are subleading at large $N$. It follows immediately from simple thermodynamics (see subsection 3.2), that any black hole with $\mu_i > 1$ gains entropy by emitting a mode with $E = H_i$.[8] (Here $i = 1 \dots 3$ range of the three Cartan directions - the three embedding two planes - of $SO(6)$).[9] As a consequence, black holes with $\mu_i > 1$ are always thermodynamically unstable[10] to the emission of large dual giant gravitons that carry all their charge in the $i^{th}$ Cartan direction.[11] However, all the hairy black holes constructed in [5, 7, 9, 10] turn out to have at least one chemical potential greater than unity. It follows immediately that the hairy black holes of [5, 7, 9, 10] are unstable to the emission of large dual giants.

In this paper, we propose that the endpoint of this condensation instability is given by Dual

---

[4]As in the context of bottom-up models, hairy black holes solutions simplify at small charge, and can be constructed analytically in that context [5, 7].

[5]Within the consistent truncation of [5, 9, 10]), usual 'vacuum' black holes turn out to be unstable when their chemical potential exceeds a critical value $\mu_c(Q)$ ($Q$ is the charge). At small $Q$, $\mu_c(Q)$ can be computed analytically; one finds $\mu_c(Q) = 1 + 2\frac{Q^2}{N^4} + \dots$ (see eq 6.13 in [5]). At larger values of the charge $\mu_c(Q)$ is a complicated numerically determined function [9, 10] that, however, turns out to always be strictly greater than unity. [5, 7, 9, 10] proposed that hairy black holes replace unstable vacuum black holes in the phase diagram of $\mathcal{N} = 4$ Yang Mills. Note all hairy black constructed in [5, 7, 9, 10] have $\mu > 1$. Similar comments apply to the one and two charge black holes studied in [7]. See §3.3 for details.

[6]The precise version of this statement is as follows. Dual giant gravitons carry an $SO(6)$ charge which is an $SO(6)$ adjoint element, i.e. a $6 \times 6$ antisymmetric matrix. The charge matrix for dual giant gravitons turns out to be of rank two, and so has eigenvalues $(iH, -iH, 0, 0, 0, 0)$. $H$, in the main text, is the modulus of either of the nonzero eigenvalues of the dual giant charge matrix.

[7]In the absence of the black hole, the dual giant is BPS and so has $E = H$.

[8]On the other hand such an emission causes a seed black hole with $\mu < 1$ to lose entropy.

[9]Recall that any $SO(6)$ charge matrix can be diagonalized by an $SO(6)$ rotation, we always work in an $SO(6)$ frame in which the central black hole carries diagonal charges $(Q_1, Q_2, Q_3)$ and has corresponding chemical potentials $\mu_1, \mu_2, \mu_3$.

[10]As we explain in §5, this instability involves tunneling through a barrier. Consequently, it is perhaps more accurate to say that black holes with $\mu_i > 1$ are 'metastable' in the microcanonical ensemble. We expect the same black holes to have a simpler 'roll down the hill' instability in the grand canonical ensemble (see §3.2).

[11]$S^5$ can be embedded in a six-dimensional space parameterized by three complex numbers ($z_i$). These are dual giants that spin in the $i^{th}$ complex plane. The $SO(6)$ charge matrix for such a dual giant is $6 \times 6$ block diagonal, $i^{th}$ block equalling $iH\sigma_2$, and the remaining $2 \times 2$ blocks equalling zero.

Dressed Black Holes (DDBHs). DDBHs consist of a seed $AdS_5$ black hole with $\mu_i = 1$[12] (for at least one value of $i$) surrounded by one, two or three large dual giant graviton of radius (in $AdS_5$) of order $\sqrt{N}$ and charge of order $N^2$. The dual giant(s) lives so far away from the black hole[13] that they interact very weakly with it.[14] In fact, the leading interaction between the dual giants and the black holes results from the fact that each dual giant reduces the five form flux at the location of the black hole by one unit. The entire effect of this interaction (which, by itself, is already subleading at order $\frac{1}{N}$ compared to the leading order) is to renormalize the thermodynamic charges of the central black hole to those of the $SU(N')$ theory, where $N' = N - m$, and $m$ is the number of dual giants in the solution. All remaining interactions between the dual giants and the black hole can be shown to be of order $\frac{1}{N^3}$, and so are highly subleading.

DDBH solutions contain small number (one, two or three) of dual giants (rather than a gas of such giants) because the effective reduction of flux (that accompanies each additional dual giant) is entropically unfavorable for seed black hole (see §3 for details). Consequently, the thermodynamically dominant configuration is the one with the least number of dual giants needed to carry the required charges. This number is sometimes greater than unity (and as large as three) for the following reason. The main role that dual giants play in DDBH solutions is to act as a sink for $SO(6)$ charge. However single dual giants carry $SO(6)$ charges of a constrained sort: the most general ($SO(6)$ adjoint, i.e. $6 \times 6$, antisymmetric) charge matrix carried by a single dual giant graviton is of rank 2. We need up to 3 dual giants in order to make up the most general (maximal rank) charge matrix.

As we have mentioned, the central black hole in a DDBH has $\mu_i = 1$ for at least one of the three values of $i$.[15] The nature of DDBHs changes depending on how many of the three chemical potentials equal unity. Thus DDBHs can be subclassified as

- DDBHs of rank 2: when central black hole has $\mu_i = 1$ for one value of $i$, with the other two $\mu_j < 1$. Such DDBHs are dressed by a single dual giant which carries charge only in the $i^{th}$ direction.

- DDBHs of rank 4: When two chemical potentials of the central black hole equal unity, with the third chemical potential less than unity. Such DDBHs are generically dressed by two dual giants, which, respectively, carry charges under the two Cartan charges with unit chemical potential.[16]

- DDBHs of rank 6: when all three chemical potentials of the central black hole equal unity.[17] Such DDBHs are generically dressed by three dual giants, which, respectively carry charge in the $i = 1, 2$ and 3 Cartan charge directions respectively.[18]

At leading order in the large $N$ limit, the (collection of) dual giants and the black hole are well separated in space. In this limit, as a consequence, a DDBH can be viewed as a non interacting mix of its components. The energy (and charges) of a DDBH is simply the sum of the energies of its components, and the entropy of a DDBH is simply the Bekenstein-Hawking entropy of its central black hole. This simple fact completely determines the thermodynamics

---

[12] Up to corrections that are subleading at large $N$.

[13] And, in the case that there are multiple dual giants, from each other.

[14] In the absence of the black holes, the various dual giants (recall the number equals 1, 2 or 3) turn out to be mutually BPS: they preserve a $1/8^{th}$ fraction of the 32 supersymmetries of $\mathcal{N} = 4$ Yang Mills. As a consequence, the interactions between the various dual giants do not renormalize the energy of the solution.

[15] And $\mu_j \leq 1$ (for the other two values of $j$).

[16] As a consequence, the net charge matrix of this dual giant configuration is of rank four.

[17] As a consequence, all three charges $Q_i$ of this black hole are also equal to one another.

[18] As a consequence, the net charge matrix of this three dual giant configuration is generically of rank six.

of DDBH phases. At vanishing angular momentum, it seems reasonable to assume that non-hairy black holes (which we refer to as 'vacuum black holes' through the rest of this paper) and DDBHs are the only relevant 5 dimensional black hole phases (see below for a discussion of 10d black holes). Under this assumption, one can use the known thermodynamics of vacuum black holes and DDBHs to construct the phase diagram of $\mathcal{N} = 4$ Yang Mills at strong coupling, as a function of $E, Q_1, Q_2, Q_3$ (at any given value of charges, one simply chooses the one among these phases that has the highest entropy). The resulting phase diagram - which we have worked out in section 3.4 -is intricate as it has phase transitions between vacuum black holes (with all $\mu_i < 1$, the three different rank one DDBH phases, the three different rank 2 DDBH phases and the unique rank 3 DDBH phase. Together with the usual vacuum black holes, the three DDBH phases described above completely fill out the microcanonical phase diagram, all the way down (in energy) to the BPS bound.

We emphasize that the phase diagram of §3.4 has been worked out under the assumption that either $5d$ vacuum black holes or DDBHs are always the dominant phase. When the bulk theory is $AdS_5 \times S^5$, we know this assumption is not always true. Away from extremality,[19] 5d black holes that are small in units of the $S^5$ radius are Gregorry Laflamme unstable in the $S^5$ directions; the end point of this instability is a 10d black hole moving on the $S^5$. The phase diagram presented in §3.4 ignores this phenomenon, and so will correctly reproduce the phase diagram of $\mathcal{N} = 4$ Yang Mills as a function of charges and energies, only when the core (or vacuum) black holes are either large or very near to extremality. We leave a more complete analysis of the full phase diagram - one that accounts for both DDBH phases - as well as 10d black hole phases - to future work.

Recall that black holes locally reduce to black branes when their temperature and chemical potentials are scaled as $T \sim \frac{\tau}{\epsilon}, \mu_i \sim \frac{\nu_i}{\epsilon}$, with $\epsilon$ taken to zero at fixed $\tau$ and $\nu_i$. This limit yields black branes, whose ratio of chemical potential to temperature equals $\frac{\nu_i}{\tau}$. We emphasize that this scaling limit takes $\mu_i \to \infty$, and so, in particular, to values greater than unity. It follows immediately that charged black branes are unstable (to the emission of huge charged duals) at any nonzero charge density. The end point of this instability is a phase in which almost all of the energy is carried by the central black hole, while almost all of the charge is carried by the dual giant, which now lives at a point that is deep in the UV end of the geometry. Surprisingly enough, this implies that the entropy of $\mathcal{N} = 4$ Yang Mills is determined completely by the energy (and is independent of the charge, i.e. the parameter $\alpha$ below) when the energy and charge are both taken large, along a curve of the form $E = \alpha Q^a$ for every value of $a > 1$ (see §3.6 for details). This is in sharp contrast with the naive prediction of 'vacuum' charged black branes (which predict an entropy formula that depends on $\alpha$ when $E$ and $Q$ are both taken large along the curve $E = \alpha Q^{\frac{4}{3}}$), and is related to several previous observations predicting effective instabilities related to the tunneling of branes to 'infinity' (large values of the $AdS_5$ radius), see e.g. [15–19]

Once we turn on angular momentum in addition to charges, the phase diagram of $\mathcal{N} = 4$ Yang Mills theory has additional Grey Galaxy phases (see [8]) that replace the 'vacuum black holes' when $\omega_i > 1$ ($\omega_i$, $i = 1, 2$, are the two angular velocities of the black hole). In order to explain how this works, in §3 we discuss the thermodynamics of the dual to $\mathcal{N} = 4$ Yang Mills at energy $E$, charges $Q_1 = Q_2 = Q_3 = Q$ and angular momenta $J_1 = J_2 = J$ in more detail. Upon lowering energy at fixed values of $Q$ and $J$, we find that our system generically makes a phase transition from the vacuum black hole phase into either the DDBH or the Grey Galaxy phase. The intermediate phase phase is a DDBH at large charge, but a Grey Galaxy at large angular momentum. The BPS plane $E = 3Q + 2J$, of course, forms a boundary of our phase diagram. The 'large charge' portion of this plane bounds the DDBH phase while the

---

[19]As explained in the introduction of [5] charged 5d black holes are stable to Gregory Laflamame type instabilities - even when they are very small - when they are near enough to extremality.

'large angular momentum' part of this plane bounds the Grey Galaxy phase. These two distinct portions of the BPS plane are separated by the formula that determines $Q$ as a function of $J$ in supersymmetric Gutowski-Reall [20, 21] black holes.

As we have explained above, if we lower the energy at fixed $J$ and fixed but large $Q$, we enter the DDBH phase before hitting the BPS plane. On the BPS plane, the DDBH is supersymmetric. It consists of three supersymmetric dual giants surrounding a supersymmetric Gutowski-Reall black hole. The dual giant gravitons here turn out to obey the Kappa symmetry constraint on supersymmetry even when they carry charges of order $N$ and so are located at a finite value of the radial parameter. We thus appear to have found a new 5 parameter set of supersymmetric black hole solutions - supersymmetric DDBHs - whose thermodynamics is precisely reproduced by the non interacting model, as guessed in [5] (see below equation 7.15 of that paper). In the case of black holes with larger angular momentum than charge, we expect that the black hole on the BPS plane is either a supersymmetric Grey Galaxy or a supersymemtric Revolving Black hole (see [8]). In the (hopefully soon) upcoming paper [22], we use these constructions 5 parameter set of SUSY black states in $\mathcal{N} = 4$ Yang Mills to conjecture explicit formulae for the cohomological supersymmetric entropy as a function of its five charges.

We emphasize that non supersymmetric extremal black holes never make an appearance in the phase diagram for $\mathcal{N} = 4$ theory (either on their own or as the core black hole component of a DDBH) as these black holes always have either $\mu_i > 1$ (for some $i = 1 \ldots 3$) or $\omega_i > 1$ (for some $i = 1 \ldots 2$.), and so are always superradiant unstable. Even more strikingly, black holes that are well approximated by charged black branes (at any nonzero value of the charge density) also never make an appearance for the same reason: they are always unstable.

As we have seen above, the thermodynamics of DDBHs can be constructed without reference to details of the DDBH solutions. In order to perform more intricate calculations in such phases, however, we need full control over the relevant bulk solutions. For this reason, in the rest of this paper, we turn to a detailed study of DDBH solutions built around a central black hole that carry energy $E$ and charges $Q_1 = Q_2 = Q_3 = Q$, $J_1 = J_2 = J$).

In section §4 we review and generalize the study of [13,14] of the classical motion of probe D-branes around these black holes. In particular, we determine the effective potential (as a function of radial coordinate) seen by probe solutions with any given value of $SO(6)$ charges. At large enough values of the probe charges, this potential has a (sometimes local) minimum at a value of $r$ near to $\sqrt{\frac{H}{N}}$ (here $H$ is the charge of the probe: see §4 for more accurate versions of these statements). Classically, a probe that sits at this minimum constitutes a stable configuration that (locally) minimizes energy at any fixed value of charges: this is true both of black holes with $\mu > 1$ and $\mu < 1$.[20]

The large charge classical probe dual solutions described above are classically stable around black holes of arbitrary $\mu$, in apparent conflict with the thermodynamic expectation that dual giants should be able to coexist only with black holes with $\mu = 1$.[21] In section 5 we resolve this apparent mismatch by quantizing the motion of the dual giants. We demonstrate that the wave function for this probe brane obeys an effective Klein-Gordon equation. In §5 we then use WKB methods to solve this Schrodinger equation. We study wave functions with time dependence $e^{-i\omega v}$ (here $v$ is the Eddington-Finkelstein time coordinate that is regular on the future horizon of the black hole) and construct the 'lowest energy' solution to this wave

---

[20]Though we find a local minimum both for $\mu > 1$ and $\mu < 1$, there is an interesting difference between these two cases even at the classical level. When the black hole has $\mu > 1$, the minimum described above is global in nature (in the sense that no value of $r > R$ has a larger value of the potential: $R$ is the black hole horizon). When $\mu < 1$, on the other hand, the minimum described above is local in nature: in particular the potential at the horizon has a lower value than that at the local minimum. This point was observed and emphasized in [13,14].

[21]Recall that black holes with $\mu > 1$ are thermodynamically unstable to the emission of large dual giant gravitons while those with $\mu_i < 1$ tend to increase their entropy by 'eating up' dual giants.

equation that is both normalizable at infinity and regular on the future horizon. We find that the real part of $\omega$ (for such solutions) is parametrically near to the minimum of the classical potential, as might have been expected on general grounds. However, $\omega$ also develops an imaginary part. This imaginary part is of order $e^{-H}$, where $H$ is the charge of the probe brane, and so is extremely small. It changes from negative (implying exponential decay) when $\mu < 1$ to positive (implying exponential growth) when $\mu > 1$, and vanishes at $\mu = 1$, at which point the DDBH is dynamically stable.

It follows that a D-brane placed at the minimum of the classical potential is, generically, unstable at the quantum level. When $\mu < 1$ it decays into the black hole. When $\mu > 1$ it grows (being fed by the black hole by the process of superradiance). When $\mu = 1$, on the other hand, our quantum wave function is stable even quantum mechanically. In other words, the thermodynamic instabilities described above are perfectly mirrored by dynamical instabilities, even though the (quantum) time scales for these instabilities are enormous.

In addition to illuminating the stability of dual giants, the quantization of dual solutions has one other advantage: it allows us to precisely determine the $SO(6)$ charges of DDBHs. See §5.9 for details.

We have, so far, dealt with dual giants in the probe approximation. As our probe consists of a single D-brane, the probe approximation works excellently at distances of order unity away from the dual giant. As our probe carries 'classical' energies (i.e. energies of order $N^2$) however, this is no longer the case over length scales of order $\sqrt{N}$. At these distance scales the backreaction of these probe branes is no longer negligible. In particular, this backreaction captures the energy and charge of these branes: in our final (thermodynamically dominant) solution, these are comparable to the energy and charge of the central black hole, and so certainly cannot be ignored. In fact these backreaction effects are easily computed as follows. The supergravity solution corresponding to a single dual giant in $AdS_5$ was presented in the famous paper by Lin, Lunin and Maldacena [23]. The solution of [23] applies to dual giants of any charge: in particular to charges of order $N^2$. Since these branes have a very large radius, it is possible to add a $\mu_i = 1$ black hole to the centre of these solutions (in a matched asymptotic expansion in the radius of the dual giant, i.e. in $\frac{1}{\sqrt{N}}$), yielding a supergravity solution for our new solutions. This solution is presented in detail in §6 below.

The supergravity solution described in the previous paragraph yields the bulk dual description of the new phases constructed in this paper. This dual bulk solution can then be used, for instance, to compute the expectation value of all single trace operators in the corresponding solution. As an example of this point, in section §6.5 we compute the nontrivial expectation value for $\text{Tr}Z^n$ for all $n$. The fact that we obtain nontrivial expectation values for these observables, contrasts these solutions with those (like Kerr RN AdS black holes) obtained within the consistent truncation of gauged supergravity.[22]

This emphasizes the point that DDBH solutions cannot be written down within any bottom up model of 5-dimensional supergravity with a finite number of five dimensional fields. Indeed, the single D-branes that appear as part of these phases contain singularities at the location of the brane, and so cannot really be accurately described everywhere even within 10-dimensional supergravity. A complete bulk description of this phase requires probe D-branes governed by the Born-Infeld action, and so elements of the full 10-dimensional string theory.

The rest of this paper is organized as follows. In section 2 below we review relevant aspects of 'vacuum' charged rotating black hole solutions in $AdS_5 \times S^5$. In §3 we study the

---

[22]Gauged supergravity is believed to form a consistent truncation of the full 10 d supergravity in the following sense. If we decompose 10 d supergravity into 5d fields, denote those fields that lie in gauged supergravity a $a$ type fields, and those fields that lie outside gauged supergravity as $b$ type fields, then it is believed that all couplings between $a$ and $b$ type fields are of quadratic or higher order in $b$. As a consequence, a solution of gauged supergravity never sources $b$ type fields including all those dual to $Tr(Z^n)$ for all $n \geq 3$.

thermodynamics of our new solutions and argue that they dominate over hairy black holes. In §3.4 we use the analysis developed above to construct a detailed conjectured microcanonical phase diagram for $\mathcal{N} = 4$ as a function of charges. In §4 we review and generalize the study of [13,14] for the classical motion of dual giants in black hole backgrounds. In §5 we quantize the motion of these dual giants and demonstrate that they are quantum mechanically stable only around black holes with $\mu = 1$. In §6 we construct backreacted bulk solutions for probes with charge of order $N^2$ around black holes (that also carry charge and energy) of order $N^2$. In §7 we conclude with a discussion of our results. We present supporting material for the main text in several appendices.

## 2 Review of 'vacuum' black hole solutions

In this section, we recall and review well known 'vacuum' black hole solutions in $AdS_5 \times S^5$ presented in [24–27]. In §2.1 below we present a detailed review of the solutions and thermodynamics of black holes with equal values of the three $SO(6)$ Cartan charges, and equal values of the two $SO(4)$ Cartans. In §2.2 below, we present a brief review of black holes that carry three distinct $SO(6)$ Cartan charges, but no angular momenta. The reader who is familiar with the relevant black hole solutions should feel free to skip to the next subsection.

### 2.1 Black holes with $Q_1 = Q_2 = Q_3 = Q$ and $J_1 = J_2 = J$

In this subsection, we review the solutions and thermodynamics of black holes that carry angular momentum $J_L = J$, with $J_R$ set to zero ($J_L$ and $J_R$ are the Cartan charges of $SU(2)_L \times SU(2)_R = SO(4)$). In other words, these solutions carry $SO(4)$ two plane Cartan charges $J_1 = J_2 = J$. They also carry $SO(6)$ Cartan charges $Q_1 = Q_2 = Q_3 = Q$. These black holes appear in a 3-parameter set, parameterized by their energy $E$, $J$ and $Q$.

#### 2.1.1 Structure of the 10d black hole metric

The metric in ten dimensions for the black hole solution[23] takes the form[24]

$$\frac{ds_{10}^2}{R_{AdS}^2} = g_{\mu\nu}dx^\mu dx^\nu = ds_5^2 + ds^2(\mathbb{CP}^2) + (d\Psi + \Theta - A)^2 \,, \tag{1}$$

where $R_{AdS}$ is the radius of $AdS_5$, given by

$$R_{AdS}^4 = 4\pi g_s N (\alpha')^2 \,, \tag{2}$$

$\Psi$ is a coordinate along the 'fibre' direction,[25] and $A$ is a one-form valued in the five dimensional space $ds_5^2$.[26] The appearance of $ds^2(\mathbb{CP}^2)$ in this metric reflects the fact that a solution with $SO(6)$ charges $(Q,Q,Q)$ preserves a $U(3)$ subgroup of the full $SO(6)$ isometry of the sphere. This metric satifies Einstein's equations in 10 dimensions in the presence of $N$ units of flux of the five-form field strength.

---

[23]For orientation, (1) reduces to $AdS_5 \times S^5$ upon letting $ds_5^2$ be the metric on the unit $AdS_5$ and setting $A = 0$. (In this situation, the last two terms on the RHS of (1) combine to give the metric on a unit $S^5$)

[24]The angles $\phi_i$ in (3) are related to those in Eq (2.1) of [24] as $\phi_i^{here} = -\phi_i^{there}$. For this reason $\Theta^{here} = -\Theta$ and $d\Psi^{here} = -d\Psi$ (see (7)). This is why the last term in (1) takes the form $(d\Psi + \Theta - A)^2$ rather than the form $(d\Psi + \Theta + A)^2$ that appears in eqn 2.8 of [24]. We have made this coordinate change for the following reason. If a metric is built out of the the combination $(d\phi - A)^2$, then a mode that moves in the positive $\phi$ direction behaves like $e^{in\phi}$ with $n$ positive. With our choice of coordinates, therefore, a probe D-brane moving in the positive $\phi$ direction carries the same charge as the black hole. This will prove convenient for us below.

[25]The terminology comes from viewing $S^5$ as a $U(1)$ fibration over $\mathbb{CP}^2$; see the next subsection for more details.

[26]Were $A$ have been set to zero, the metric above would have reduced to $ds^2 + d\Omega_5^2$, where $d\Omega_5^2$ is the metric on the unit sphere.

### 2.1.2 The squashed $S^5$ metric in more detail

The internal part of the metric (1) can be rewritten as

$$ds^2(\mathbb{CP}^2) + (d\Psi + \Theta - A)^2 = \sum_{i=1}^{3} dl_i^2 + l_i^2 (d\phi_i - A)^2 \,. \tag{3}$$

Here we have used the coordinates $\phi_i$ $i = 1\ldots 3$ together with three direction cosines $l_i$, subject to the relation

$$\sum_{i=1}^{3} l_i^2 = 1 \,, \tag{4}$$

as coordinates on the squashed $S^5$.[27] The function $\Psi$ is defined as[28]

$$\Psi = \frac{\phi_1 + \phi_2 + \phi_3}{3} \,. \tag{6}$$

The Kahler form $J$ on $\mathbb{CP}^2$ (and its corresponding 'gauge field' $\Theta$) are defined by the following equations:

$$2J = d\Theta = \sum_i d(l_i^2) \wedge d\phi_i \,,$$
$$\Theta = \sum_i l_i^2 d\phi_i - d\Psi = \sum_i \left( l_i^2 - \frac{1}{3} \right) d\phi_i \,. \tag{7}$$

The 10d volume form on the metric (1) can be rewritten, in terms of the volume form $\epsilon^{(5)}$ on the black hole metric $ds_5^2$, and the volume form on $\mathbb{CP}^2$ and the fibre as

$$\epsilon^{(10)} = \epsilon^{(5)} \wedge \epsilon^{(4)}(\mathbb{CP}^2) \wedge (d\Psi + \Theta - A) \,, \qquad \epsilon^{(4)}(\mathbb{CP}^2) = \frac{1}{2} J \wedge J \,. \tag{8}$$

In Appendix H.2 we give a detailed description of the action of the isometry group $U(3)$ on the squashed $S^5$. In particular, we verify that the one form $d\Psi + \Theta$ is $U(3)$ invariant.

### 2.1.3 Structure of the 5d black hole metric and gauge field

The five dimensional spacetime metric, and the one-form $A$, are given by[29,30]

$$ds_5^2 = -\frac{r^2 W(r)}{4b(r)^2} dt^2 + \frac{dr^2}{W(r)} + \frac{r^2}{4}(\sigma_1^2 + \sigma_2^2) + b(r)^2(\sigma_3 + f(r) dt)^2 \,, \tag{9}$$

$$A = \frac{q}{r^2}\left( dt - \frac{j}{2}\sigma_3 \right) \tag{10}$$

(see [25] [26]). $q$ and $j$ in (9) and (10) are constant parameters, which together with $p$ (see (16) (17) (18)) parameterize the black holes we study. $W(r)$, $b(r)$ and $f(r)$ are functions whose explicit form is listed in (16) (17) (18). The coordinates on this five dimensional

---

[27]We can obtain a completely explicit set of coordinates by expressing the direction cosines in terms of two angles in any convenient way. For instance, we could set

$$l_1 = \cos\alpha \,, \qquad l_2 = \sin\alpha\cos\beta \,, \qquad l_3 = \sin\alpha\sin\beta \,. \tag{5}$$

[28]$\Psi$ is the coordinate along the fibre (when, for instance, we write $S^5$ as a $U(1)$ fibration over $\mathbb{CP}^2$.)

[29]We will sometimes refer to these black holes as "Cvetic-Lu-Pope (CLP) black holes"

[30]We can check that the gauge field below gives the correct charge when integrated upon at infinity using $Q = \frac{1}{16\pi G} \int_{S^3} *F$.

spacetime are the radius $r$, the time $t$, and three angles on a warped $S^3$ ($\theta_a, \phi_a, \psi$). $\sigma_1, \sigma_2$ and $\sigma_3$ are the usual right invariant one-forms on an $S^3$ that transform in the three dimensional (vector) representation of $SU(2)_L$. Explicitly

$$\sigma_1 = \cos\psi \, d\theta_a + \sin\psi \sin\theta_a \, d\phi_a \,, \tag{11}$$

$$\sigma_2 = -\sin\psi \, d\theta_a + \cos\psi \sin\theta_a \, d\phi_a \,, \tag{12}$$

$$\sigma_3 = d\psi + \cos\theta_a \, d\phi_a \,. \tag{13}$$

These one-forms are defined so that

$$\sum_i \frac{\sigma_i^2}{4}$$

is the metric on the unit $S^3$. The fact that the metric in (9) is built entirely out of right invariant one-forms reflects the fact that our black hole is charged only under $J_L^z$, and so preserves rotations under $SU(2)_R \times U(1)_L$. (the last factor is the rotation of $\sigma_1$ and $\sigma_2$ into each other: note that $\sigma_1$ and $\sigma_2$ appear in the metric only in the combination $\sigma_1^2 + \sigma_2^2$).

The angular and charge chemical potentials of these black holes are given by[31]

$$\begin{aligned} \Omega &= f(R) \,, \\ \mu &= A_t(R) - \Omega A_{\sigma_3}(R) \,, \end{aligned} \tag{14}$$

where $R$ is the radius of the outer event horizon of the black hole.

In (10) we have chosen the gauge field to vanish at infinity but be nonvanishing at the horizon. While this choice is convenient for some purposes (because the metric, with this choice of coordinates, is manifestly $AdS_5 \times S^5$ at infinity), it is inconvenient for other purposes. The chief drawback is that, with this choice of gauge $\zeta^\mu A_\mu \neq 0$ (here $\zeta^\mu$ is the killing generator of the horizon). This makes this gauge singular in good coordinates (like Eddington-Finkelstein coordinates) at the horizon. This problem is easily remedied: all we have to do is to make the gauge transformation $A_t \to A_t - \mu$, with $\mu$ a constant, chosen to ensure that $\zeta^\mu A_\mu$ vanishes. It follows from (14), that this condition is met when we choose $\mu$ to be the chemical potential. In this choice of gauge (10) is replaced by

$$A = \left( \frac{q}{r^2} - \mu \right) dt - \frac{jq}{2r^2} \sigma_3 \,. \tag{15}$$

### 2.1.4 Details of the 5d metric

As we have mentioned above, the 5 dimensional metric (9) is parameterized by the three constants $p$, $q$ and $j$.[32] The functions $b(r)$, $W(r)$ and $f(r)$, that appear in (9), are given in terms of these parameters by

$$b(r)^2 = \frac{r^2}{4} \left( 1 - \frac{j^2 q^2}{r^6} + \frac{2j^2 p}{r^4} \right) \,, \tag{16}$$

$$f(r) = -\frac{j}{2b^2} \left( \frac{2p-q}{r^2} - \frac{q^2}{r^4} \right) \,, \tag{17}$$

$$W(r) = 1 + 4b^2 - \frac{1}{r^2}(2p - 2q) + \frac{1}{r^4}\left( q^2 + 2pj^2 \right) \,. \tag{18}$$

---

[31]In a gauge in which the gauge field vanishes at infinity, $\mu$ of the black hole equals $A_\mu \zeta^\mu$ at the horizon, where $\zeta^\mu$ is the generator of the horizon, normalized so that it equals $\partial_t - \Omega(\partial_\psi + \cos\theta \, \partial_\phi)$. This equation may be understood as follows. In Euclidean space we are forced to work with a gauge in which $A_\mu \zeta^\mu = 0$. As this condition is not met in the original gauge, we perform a gauge transformation $A_t \to A_t - \mu$, with the constant $\mu$ chosen to be the original value of $A_\mu \zeta^\mu$. In this new gauge $A_t = -\mu$ at the boundary, allowing us to identify $\mu$ as the chemical potential.

[32]Our solution is labeled by three parameters, because the black holes we study come in a three parameter family, parameterized by their energy, charge and left angular momentum.

The parameters $(p, q, j)$ are related to the conserved charges $(M, Q, J)$[33] as follows:[34]

$$M = \frac{1}{2}(3p - 3q + pj^2),\tag{20}$$

$$Q = \frac{q}{2},\tag{21}$$

$$J = \frac{1}{2}j(2p - q).\tag{22}$$

The temperature of these black holes is given in terms of $W'(R)$ by

$$T = \frac{RW'(R)}{8\pi b(R)},\tag{23}$$

where (as mentioned above) $R$ is the radius of the outer horizon, i.e. the largest root of the equation $W(r) = 0$.

### 2.1.5 The five form potential and field strength

In addition to the metric, the black hole background is characterized by its 5-form field strength given by

$$\frac{F^{(5)}}{R^4_{AdS_5}} = G^{(5)} + *_{10}G^{(5)},$$
$$G^{(5)} = -4\epsilon^{(5)} + J \wedge *_5 F^{(2)},\tag{24}$$

where[35] $\epsilon^{(5)}$ is the volume form in $AdS_5$ and $F^{(2)} = dA$. A four-form potential which gives rise to the five form field strength (24) according to

$$dC = F^{(5)},\tag{25}$$

takes the following simple form:

$$\frac{C}{R^4_{AdS_5}} = C_V + \Theta \wedge (-*_5 F + A \wedge F) + l_2^2 d(l_3^2) \wedge d\phi_1 \wedge d\phi_2 \wedge d\phi_3 - A \wedge (d\Psi + \Theta) \wedge J,\tag{26}$$

where

$$C_V = \frac{r^4 - R^4}{8} dt \wedge \sigma_1 \wedge \sigma_2 \wedge \sigma_3\tag{27}$$

(see Appendix B.1 for a demonstration). We have chosen four-form gauge to ensure that $C_V$ vanishes on the horizon (so that our choice of gauge is regular in Eddington-Finkelstein coordinates).

---

[33]These charges are in units of $N^2$(note that these differ from [25] by factor of $\frac{\pi}{2}$, such that $M_{there} = \frac{\pi}{2}M_{here}, J_{there} = \frac{\pi}{2}J_{here}, Q_{there} = 2Q_{here}$).

[34]Our black holes are supersymmetric when they obey the BPS bound

$$M - 2J - 3Q = 0.\tag{19}$$

[35]In the conventions of Appendix A, the five form field strength $F_{\mu_1 \dots \mu_5}$ had mass dimension 1, and so $F_{\mu_1 \dots \mu_5} dx^{\mu_1} \wedge \dots dx^{\mu_5}$ had mass dimension $-4$ (because the coordinates in Appendix A were all assumed to have the dimensions of length). The coordinate invariant statement is that $F_{\mu_1 \dots \mu_5} dx^{\mu_1} \wedge \dots dx^{\mu_5}$ has dimension $-4$. If we now work with coordinates that are dimensionless - as we have chosen to do in this section - then $F_{\mu_1 \dots \mu_5}$ has dimension $-4$. Consequently the LHS of the first of (24) - and hence all terms in (24) - are dimensionless.

## 2.2  Nonrotating black holes with $SO(6)$ Cartans $(Q_1, Q_2, Q_3)$

In this subsection, we briefly recall the supergravity solution for general nonrotating charged black holes in $AdS_5 \times S^5$. Such black holes appear in a four parameter set, parameterized by their three $SO(6)$ Cartan charges and their energy. The relevant black holes take the form[36]

$$ds^2 = -\frac{Y}{R^2} dt^2 + \frac{R\rho^2}{Y} d\rho^2 + R d\Omega_3^2, \tag{28}$$

$$A_i = \frac{r_j r_k - \Gamma r_i}{H_i} dt, \tag{29}$$

$$X_i = \frac{R}{H_i}, \tag{30}$$

where $ds^2$ is the five dimensional metric, $A_i$ are the three five dimensional gauge fields, $X_i$ are the three scalars (that obey the constraint $\prod_i X_i = 1$) and

$$R = (H_1 H_2 H_3)^{\frac{1}{3}}, \tag{31}$$

$$Y = R^3 + \rho^4 + \rho^2 \left( \frac{1}{3}(r_1^2 + r_2^2 + r_3^2) - \Gamma^2 \right) + \frac{1}{3}\Gamma^2(r_1^2 + r_2^2 + r_3^2) \tag{32}$$

$$- 2\Gamma r_1 r_2 r_3 + \left[ \frac{5}{18}(r_1^2 + r_2^2 + r_3^2)^2 - \frac{1}{2}(r_1^4 + r_2^4 + r_3^4) \right],$$

$$H_i = \rho^2 + \frac{\mu(s_i^2 - s_j^2) + \mu(s_i^2 - s_k^2)}{3}, \tag{33}$$

$$\Gamma = \sqrt{\mu}\left[ c_1 c_2 c_3 - s_1 s_2 s_3 \right], \tag{34}$$

$$r_i = \sqrt{\mu}(c_i s_j s_k - s_i c_j c_k), \tag{35}$$

$$c_i = \cosh\delta_i, \tag{36}$$

$$s_i = \sinh\delta_i. \tag{37}$$

The expressions for metric and the gauge fields are not directly written as a function of the charges $(E, Q_1, Q_2, Q_3)$, but rather both the fields and the charges are written in terms of the four parameters $(\mu, \delta_1, \delta_2, \delta_3)$. The charges of the solution are given by,

$$E = \frac{\mu}{4} \left[ \cosh 2\delta_1 + \cosh 2\delta_2 + \cosh 2\delta_3 \right], \tag{38}$$

$$Q_i = \frac{\mu}{4} \sinh 2\delta_i. \tag{39}$$

The entropy (in units of $N^2$) and the temperature of the black hole solution are given by,

$$S = \pi(H_1 H_2 H_3)^{\frac{1}{2}}, \tag{40}$$

$$T = \frac{Y'(\rho_0)}{4\pi\rho_0(H_1 H_2 H_3)^{\frac{1}{2}}}, \tag{41}$$

where $\rho = \rho_0$ is the location of the outer horizon and is given by the largest positive root of $Y(\rho) = 0$.

The chemical potentials of the black holes ($\mu_i$) are given by the values of the gauge field evaluated at the horizon,

$$\mu_i = \frac{3\mu \sinh 2\delta_i}{6\rho_0^2 + \mu(2\cosh 2\delta_i - \cosh 2\delta_j - \cosh 2\delta_k)}. \tag{42}$$

---

[36]Black hole solutions with unequal electric charges in the $\mathcal{N} = 2$ $U(1)^3$ gauged supergravity coupled to two vector multiplets in five dimensions were first written down in [27]. The theory has the following field content: a metric, three $U(1)$ gauge fields and two scalars. We use the notations in [25] (with $l = 0$).

# 3 Thermodynamics and phase diagrams

In this section, we discuss thermodynamic aspects of DDBHs, in the approximation that these objects can be viewed as a non interacting mix of probe dual giants with $E = Q_1 + Q_2 + Q_3$[37] and a central black hole. We first demonstrate that every black hole with $\mu_i > 1$ (for any value of $i$) is necessarily thermodynamically unstable and explain that the thermodynamical end point of this instability (i.e. the maximum entropy configuration with the same charges) consists of a black hole with one or more $\mu_i = 1$, dressed with one, two or three dual giants. In the rest of this section, we use the results of this non interacting model to work out the detailed structure of two different cuts of the microcanonical phase diagram of $\mathcal{N} = 4$ Yang Mills theory.[38] These are the microcanonical ensemble with

- $J_1 = J_2 = 0$ as a function of $E, Q_1, Q_2, Q_3$ ($E$ is the energy).

- $Q_1 = Q_2 = Q_3 = Q$, $J_1 = J_2 = J$, as a function of $E, Q, J$.

## 3.1 Thermodynamics in the non interacting model

As we have explained in the introduction, the central black hole and the dual giants in DDBHs are effectively non interacting except for one effect: each dual giant reduces the effective 5-form flux (the effective value of $N$) seen by the central black hole by one unit.

Consider a system of total energy $E^{\text{tot}}$, total $i^{th}$ charge $Q_i^{\text{tot}}$ $i = 1 \ldots 3$, and total $j^{th}$ angular momentum $J_j^{\text{tot}}$. For definiteness, we assume that all of these quantities are positive. Let us suppose that of these charges $Q_i^D$ is carried by a total of $m$ dual giants. The dual giants, therefore, carry energy $E^D = \sum_{i=1}^{3} Q_i^D$. It follows also that effective flux, $N_{\text{eff}}$, of central black hole, its energy, charge and angular momentum are given by

$$
\begin{aligned}
N_{\text{eff}} &= N - m\,, \\
E^{BH} &= E^{\text{tot}} - \sum_{i=1}^{3} Q_i^D\,, \\
Q_i^{BH} &= Q_i^{\text{tot}} - Q_i^D\,, \\
J_i^{BH} &= J_i^{\text{tot}}\,.
\end{aligned}
\tag{43}
$$

Now the full entropy of our solution is that of its central black hole. The black hole entropy formula takes the form

$$
S_{BH} = N_{\text{eff}}^2\, s_{BH}\left( \frac{E^{BH}}{N_{\text{eff}}^2}, \frac{Q_i^{\text{BH}}}{N_{\text{eff}}^2}, \frac{J_j^{BH}}{N_{\text{eff}}^2} \right).
\tag{44}
$$

Let us assume that all charges (total charges, those of the dual giants, and those of the black holes) are of order $N^2$, but that the total number of dual giants, $m$, is of order unity. In this situation the second term on the RHS of the expression

$$
N_{\text{eff}} = N\left( 1 - \frac{m}{N} \right),
\tag{45}
$$

---

[37]This relation follows because the duals we study carry no angular momentum and are approximately supersymmetric. Note that duals with zero angular momentum have highest charge to mass ratio and so extract charge out of the black hole in the thermodynamically most favourable manner, at least when black hole angular velocities $\omega_i$ all obey $\omega_i < 1$.

[38]As explained in the introduction, the phase diagram presented in this section ignores Gregory Laflamme instabilities and is conjectured to be correct only in the 'half' of energy and charge space in which all participating 5d black holes are Gregory Lafflamme stable. The completion of this phase diagram to include 10d black holes is left to future work.

is of order $\frac{1}{N}$ compared to the first and therefore fractionally small.[39] Taylor expanding, and retaining only terms of first order in smallness, we find

$$S_{BH} = N^2 s_{BH}\left(\frac{E^{BH}}{N^2}, \frac{Q_i^{BH}}{N^2}, \frac{J_j^{BH}}{N^2}\right) + 2Nm\beta\left(\frac{E^{BH}}{N^2} - \mu_i\frac{Q_i^{BH}}{N^2} - \omega_j\frac{J_j^{BH}}{N^2} - s_{BH}T\right), \qquad (46)$$

where $\mu_i$ and $\omega_j$ and $\beta$ are defined by

$$\begin{aligned}
\beta &= N^2\frac{\partial s_{BH}}{\partial E^{BH}}, \\
\beta\mu_i &= -N^2\frac{\partial s_{BH}}{\partial Q_i^{BH}}, \\
\beta\omega_j &= -N^2\frac{\partial s_{BH}}{\partial J_j^{BH}}.
\end{aligned} \qquad (47)$$

Within the non interacting model, $Q_D^i$ are determined by choosing those values that maximize the entropy listed in (46).

### 3.1.1 Extremization at leading order

Let us now try to find the values of $Q_i^D$ that maximize entropy at fixed total charges. While the first term on the RHS of (46) is of order $N^2$, the second term is of order $N$. For the purposes of determining $Q_i^D$, therefore, we can simply ignore the second term and simply extremize the first term. Using (43) and (47), we see that

$$\frac{\partial S_{BH}}{\partial Q_i^D} = \beta(\mu_i - 1), \qquad Q^{BH} < Q^{\text{tot}}. \qquad (48)$$

We see that the entropy has a local maximum only if $\mu_i = 1$. Using the fact that $\mu_i$ is an increasing function of $Q_i^{BH}$, it is easy to convince oneself that this extremum is a maximum (for variations of $Q_i^D$ at fixed $Q_j^D$, $j \neq i$). Now $Q_i^D$ is an intrinsically positive quantity.[40] As a consequence, the function $S_{BH}$ has a second 'endpoint maximum' at $Q_i^D = 0$ if[41]

$$\mu_i(Q_i^{BH} = Q_i^{\text{tot}}) < 1. \qquad (49)$$

It follow that the extremization procedure of this subsection yields one of four phases.

- If $\mu_i(Q_i^{\text{tot}}) < 1$ for all $i = 1\ldots3$ then the dominant phase is the vacuum black hole phase, and the solution does not have dual giant gravitons.

---

[39]The analysis presented in this subsection applies whenever $\frac{m}{N}$ is small. This is certainly the case when $m$ is of order unity, as assumed in the main text in the rest of this subsection. However it is also the case when $m = \zeta N$ with $\zeta \ll 1$ (see §7 for a discussion of bulk solutions corresponding to this case). The probe analysis of the rest of this section (see (50)) tells us that the reduction of flux at the centre of the black hole results in a fractional lowering of entropy of order $\mathcal{O}(\zeta)$. The analysis of §6.8.2 then tells us that interaction effects correct probe estimates at order $\frac{\zeta}{d^4} = \mathcal{O}(\zeta^3)$, and so are negligible at small $\zeta$. This discussion strongly suggests that the solutions described in this paper maximize entropy locally (in configuration space). We emphasize that nothing presented in this paper rules out the possible existence of new nonlinear solutions of supergravity that have even higher entropy than the solutions presented in this paper. For instance, we cannot rule out the possibility that thermodynamics is dominated by a new nonlinear solution - which can be, in some sense, thought of as '$\zeta$ of order unity'.

[40]The correct formula for the energy of the dual giant is $E = \sum_i |Q_i|$, which reduces to the second of (43) only when all $Q_i$ are positive. Adding negative charge always decreases entropy.

[41]Recall that $\mu_i$ is an increasing function of $Q_i^{BH}$ (at fixed values of $Q_j^{BH}$ for $j \neq i$). Also that $Q_i^{BH}\mu_i$ vanishes when $Q_i^{BH} = 0$. Using these facts, it is easy to convince oneself that, for every choice of $Q_j^{BH}$ and $E$, either (48) holds (for some $Q_i^{BH}$ in the range$(0, Q_i^{\text{tot}})$) or (49) holds (both cannot simultaneously be true).

- The black hole at the centre could have $\mu_i = 1$ for one $i$, and $\mu_j < 1$ for the remaining $j$. In this case, the solution generically carries dual giants charged in the $i^{th}$ direction. This is a DDBH of rank 2.

- The black hole at the centre could have $\mu_i = 1$ for two values of $i$, and $\mu_j < 1$ for the remaining $j$. In this case the solution generically carries dual giants charged in the two $i$ directions. This is a DDBH of rank 4.

- The black hole at the centre could have $\mu_i = 1$ all three values of $i$. In this case the solution generically carries dual giants charged all three $i$ directions. This is a DDBH of rank 6.

### 3.1.2 Extremization at subleading order

The analysis of the previous subsubsection has served to determine $Q_i^D$ for all $i$. We now turn to the determination of $m$, the total number of D-branes in a DDBH phase. Referring to (46), we see that we are instructed to determine $m$ in a manner that maximizes the quantity[42]

$$2Nm\beta \left( \frac{E^{BH}}{N^2} - \mu_i \frac{Q_i^{\text{BH}}}{N^2} - \omega_j \frac{J_j^{\text{BH}}}{N^2} - s_{BH} T \right). \tag{50}$$

This quantity is proportional to the Gibbs Free energy of the black hole. We have numerically checked that the Gibbs free energy of all black holes with at least one $\mu_i = 1$, and all other $\mu_j < 1$ is always negative. Consequently the quantity above is maximized when $m$ takes the smallest value consistent with being in the phase of interest. Consequently $m = 1$ for DDBHs of rank 2, $m = 2$ for DDBHs of rank 4, and $m = 3$ for DDBHs of rank 6.

We emphasize that any black hole with $\mu_i > 1$ is thermodynamically unstable. From the analysis above, this follows because (49) is not obeyed, so the black hole can always increase its entropy by emitting duals charged under the $i^{th}$ charge. As this statement follows from general thermodynamical considerations it applies equally well to vacuum black hole, hairy black holes as well as DDBH solutions.[43]

## 3.2 Thermodynamics in the canonical ensemble

DDBH phases can also be analysed in the canonical ensemble. At inverse temperature $\beta$ and chemical potential $\mu$, the effective Boltzmann factor of a large dual giant graviton of charge (in the $i^{th}$ Cartan direction of $SO(6)$) and mass both equal to $q_i$

$$e^{-\beta q_i (1 - \mu_i)}. \tag{51}$$

It follows immediately that the black hole saddle, at fixed $\beta$ and $\mu_i$, is unstable to the Bose condensation of large dual giants when $\mu_i > 1$ (we expect that this instability will be seen in the divergence of the one loop determinant around the relevant black hole saddle).[44]

---

[42]Even though we are working with a subleading term in the entropy, we have proceeded ignoring the entropy of the dual giant gas (associated with the various ways of dividing the charge between dual giants). This is justified as this dual giant entropy is of order unity (more quantitatively, this entropy $S(m, Q^D)$ is of order $m \ln \frac{Q^D}{m}$ when $m \ll Q^D$) and so further subleading compared to the term we have retained (which is of order $N$). While the D-brane entropy is of order $N$ when $m$ is of order $N$, it is still naively subleading compared to the reduction in black hole entropy (which is of order $N^2$ in this case). Of course a proper analysis of this case goes beyond the strict non interacting model: see the discussion of '$\zeta$ solutions' in §7.

[43]In a situation in which a black hole has $\mu_i > 1$ for more than one value of $i$, the final end point of the instability - the global maximum of entropy - need not contain a dual carrying charges for all unstable values of $i$. This can come about as follows. If $\mu_2 > \mu_1 > 1$, then the end point of the instability will definitely carry a dual giant carrying $Q_2$ charge. It may, however, be that the core black hole of this DDBH ends up having $\mu_1 < 1$. We will see an example of this situation at the end of section §3.4.

[44]This is the case even though these black holes have negative Gibbs free energy and so are Hawking Page stable.

More quantitatively, the partition function for a single dual giant is given by a formula of the schematic form

$$Z^i_{\text{Dual}}(\beta, \mu_i) = \sum_{n=1}^{\infty} \left( Z^{(n)}_{\text{excit}}(\beta)\, e^{-\beta(1-\mu)n} \right) \approx \frac{Z^{(\infty)}_{\text{excit}}(\beta)}{\beta(1-\mu_i)}. \tag{52}$$

Here $Z^{(n)}_{\text{excit}}(\beta)$, the partition function of excitations over a single large dual giant of fixed charge $n$, is of $\mathcal{O}(1)$ (in terms of its scaling with $N$), and tends to $Z^{(\infty)}_{\text{excit}}(\beta)$ in the large $n$ limit. As this quantity is of order 1 and nonsingular, it plays no role in what follows, and we ignore it. Keeping only the relevant terms we have

$$\ln Z_{\text{Dual}} = -\ln\left(\beta(1-\mu_i)\right). \tag{53}$$

Within the non interacting model, the logarithm of the full partition function of our system is the sum of the log of the black hole partition function, and one term of the form (53) for each value of $i$. In equations

$$\ln Z_{\text{Tot}} = \ln Z_{\text{BH}} - \sum_{i=1}^{3} \ln \beta(1-\mu_i), \tag{54}$$

$SO(6)$ charge in the $i^{th}$ direction is obtained by actiing on the log of the partition function by the operator $\frac{1}{\beta}\partial_{\mu_i}$. It follows that

$$Q^i_{\text{Tot}}(\beta, \mu_i) = Q^i_{BH}(\beta, \mu_i) + \frac{1}{\beta(1-\mu_i)}. \tag{55}$$

As $Q^i_{BH}(\beta, \mu_i)$ is of order $N^2$, the second term on the RHS of (55) is negligible compared to the first unless $1-\mu_i \sim \mathcal{O}(1/N^2)$. In this later situation the two terms on the RHS are comparable, yielding a DDBH phase.[45]

In summary, we obtain the rank 2, rank 4 or rank 6 DDBH phase in the canonincal ensemble upon setting

$$\mu_i = 1 - \frac{\chi_i}{N^2}, \tag{56}$$

for (respectively) one, two or three values of $i$. In these phases, the partition function is a nontrivial function of $\beta$, $\alpha_i$ (for the values of $i$ for which (56) holds), and $\mu_j$ (for the values of $j$ for which (56) does not hold).

### 3.3 Comparison with hairy black holes

In the introduction we have mentioned that earlier attempts to determine the end point of the superradiant instability in IIB theory on $AdS_5 \times S^5$ led to the construction of hairy black holes. These black holes have been constructed at zero angular momentum, and with either three equal $SO(6)$ charges, or two equal $SO(6)$ charges with the third charge set to zero, or one $SO(6)$ charge with the last two set to zero. We examine these in turn.

#### 3.3.1 Three equal charges

Hairy black holes with $Q^{\text{tot}}_1 = Q^{\text{tot}}_2 = Q^{\text{tot}}_3 = Q^{\text{tot}}$ and $J^{\text{tot}}_1 = J^{\text{tot}}_2 = 0$. were first constructed analytically at small values of $Q^{\text{tot}}$ in [5], and then constructed numerically, at arbitrary values of $Q^{\text{tot}}$ in [9].

---

[45] At these values of $\mu_i$, the entropy following from the dual giant partition function is much smaller than $N^2$ and so can be ignored.

Let us first examine the analytic hairy black hole solutions valid at small $Q^{\text{tot}}$.[46] The chemical potential of these black holes was computed in [5] in a power series in $Q^{\text{tot}}$, and presented in equation 6.16 of that paper. Eq 6.16 reports a chemical potential that is unity at leading order, but has a positive correction at order $Q^{\text{tot}}$. It follows, therefore, that the chemical potential of the hairy black holes is greater than unity (at least at small values of the charge). By the analysis presented above, these black holes are thus unstable to the emission of dual giants.

The entropy of the hairy black hole was also calculated in [5] in a power series in $Q^{\text{tot}}$. At leading order the result equals that of the non interacting model presented above. At first subleading order, however, one finds a negative correction to this leading order result (see equation (6.17) of [5].). It follows, therefore that DDBHs carry larger entropy than hairy black holes, at least at small charge.

Let us now turn to the numerical construction of hairy black holes valid at all values of the charge. [9] reports that these solutions all always have $\mu > 1$ (see the caption on fig. 10 of [9]). We conclude that the hairy black holes constructed in [9] are unstable to the emission of duals at all values of the charge. We believe this means that their entropy is always smaller than that of DDBHs, but the numerical nature of these solutions has prevented us from verifying this directly.

### 3.3.2 Two charge and one charge black holes

Always working at zero angular momentum, hairy black holes with charges $Q_3^{\text{tot}} = 0$, $Q_1^{\text{tot}} = Q_2^{\text{tot}} = Q^{\text{tot}}$, and (separately) black holes with $Q_2^{\text{tot}} = Q_3^{\text{tot}} = 0$ and $Q_1^{\text{tot}} = Q^{\text{tot}}$ and have recently been constructed [7], both analytically (at small $Q^{\text{tot}}$) as well as numerically (at general values of $Q^{\text{tot}}$). As in the case of their three charge cousins, the chemical potential of these new hairy black holes always turns out to be greater than unity. This has been verified both analytically at small charges (see e.g. Eq 3.75 and 4.74a and 4.74b of [7]) as well as numerically at all charges (see Fig. 7 and 16 of [7]). We have also used the small charge formulae presented in [7] to explicitly verify that the entropy of these black holes is smaller than that of the non interacting model at first nontrivial order in the small charge expansion. As in the last subsubsection, we expect this inequality to continue to hold at larger charges.

## 3.4 Phase diagram with $J_1 = J_2 = 0$

In this section we describe the microcanonical phase diagram, at $J_1 = J_2 = 0$, that follows from the non interacting model. In other words we find a phase diagram as a function of four variables, the energy $E$ and the three charges $Q_1, Q_2, Q_3$.

In the four dimensional space parameterized by these charges, we have four important three dimensional surfaces. These are

- The BPS plane $E = Q_1 + Q_2 + Q_3$.

- The three sheets $S_i$ defined by the condition $\mu_i = 1$ (see (42)) together with the inequalities $Q_i \geq Q_j$ (for both $j \neq i$).[47]

In addition to these three dimensional sheets, we also need to keep track of the two dimensional submanifolds, $S_{ij}$ defined to be the intersection of $S_i$ and $S_j$. On $S_{ij}$ we have $Q_i = Q_j \geq Q_k$ (here $k$ is the third index, the one that is neither $i$ nor $j$). Finally we have

---

[46]Interestingly enough, the non interacting model presented above first appeared in [5] as an explanation of the leading order thermodynamics at small charge. In that context the non interacting nature was a result of form factor effects; tiny black holes interact very little with an $AdS_5$ sized charged cloud.

[47]The inequality is imposed to guarantee that $\mu_j < \mu_i = 1$, so that points on the sheet $S_i$ are not unstable to the superradiant emission of charges $Q_j$, $j \neq i$.

the distinguished one dimensional curve $S_{123}$ defined to be the curve on which each of the three $\mu_i = 1$.[48]

Points with $Q_i \geq Q_j$ (for both $j \neq i$), and that lie above[49] the sheet $S_i$ all lie in the 'vacuum black hole' phase. The dominant solutions, for such points, are simply the black holes reviewed in §2.2.

The $i^{th}$ rank-2 DDBH phase is created as follows. We start at any point on the sheet $S_i$ and shoot upwards at 45 degrees in the $E - Q_i$ plane. Since our starting sheet is 3 dimensional, and the lines along which we shoot are one dimensional, this construction gives us a four dimensional region of charge space, or a 'phase'. It is easy to check that the intersection of the surface $S_i$ to any two plane of constant $Q_j$, $j \neq i$ (at any value of $Q_j$), is a curve that everywhere has slope greater than unity in the $E - Q_i$ plane, when we regard $E$ as the $y$ axis and $Q_i$ as the $x$ axis. As a consequence of this fact, all the 45 degree rays shot out from $S_i$ - and so all points in the $i^{th}$ rank 2 phase - lie below the sheet $S_i$. Indeed, the sheet $S_i$ forms the phase boundary between the vacuum black hole phase and the $i^{th}$ rank 2 DDBH phase.

The sheet $S_i$ constitutes the upper bound of the $i^{th}$ rank 2 DDBH phase. This phase also has a lower bound, which can be constructed as follows. Recall that the $i^{th}$ rank two phase is constructed by shooting 45 degree lines from points on $S_i$. But $S_i$ itself has two boundaries, namely the surfaces $S_{ij}$ for the two values of $j \neq i$. If we shoot the upward moving $E = Q_i$ lines out of these two surfaces, we obtain the lower boundary of the $i^{th}$ rank 2 phase. This boundary separates this phase from the $(ij)^{th}$ rank 4 phase. Note that the two phase boundaries are each three dimensional, we denote them by $B_{i,j}$.[50] The two $B_{i,j}$ meet along the two dimensional sheet obtained by shooting upward directed $E = Q_i$ from the curve $S_{123}$. This two dimensional sheet lies along the plane $Q_{j_1} = Q_{j_2}$ (where $j_1$ and $j_2$ are the two values of $j \neq i$).[51]

Let us now turn to considering the rank 6 DDBH phase. This phase is constructed starting from the curve $S_{123}$ (along which $Q_1 = Q_2 = Q_3$) and shooting 'positive' upward directed lines, in any positive direction[52] in the manner so that

$$\Delta E = \Delta Q_1 + \Delta Q_2 + \Delta Q_3 \,. \tag{57}$$

The set of positive rays that originate at any point on $S_{123}$ is parameterized by two parameters. The points along any one of these rays are parameterized by one additional parameter. Finally, points on the curve $S_{123}$ are, themselves, also parameterized by a single parameter. Adding up, we see that the construction above yields a four dimensional region, i.e. a phase. The lower boundary of this phase is simply the BPS plane (we approach this boundary by shooting lines out of the origin (along this boundary the black hole component of the phase vanishes: the DDBH reduces to dual giants in pure $AdS$ space). The upper boundary of this phase is obtained when we restrict the lines shot out of $S_{123}$ to lie at constant value of either $Q_1$, $Q_2$ or $Q_3$ (rays on such planes lie on the edge of positivity). The rank 6 DDBH phase thus has 3 upper boundaries, each of which is three dimensional. We define the boundary generated by lines with $\Delta Q_i = 0$ to be the sheets $D_{jk}$ (where $j$ and $k$ are the two indices $\neq i$). The sheets $D_{jk}$ separate the unique rank 6 DDBH phase from the $(jk)^{th}$ rank 4 DDBH phase.

---

[48]This curve is, for instance, the intersection of $S_{12}$ and $S_3$.

[49]Through this discussion, we think of the energy axis parameterizing a vertical direction, and refer to points with larger energy as lying 'above' points with smaller energy, but the same value of other charges.

[50]We re-emphasize that $B_{i,j}$ is the lower boundary of the $i^{th}$ rank 2 DDBH phase, created by shooting upwards 45 degree lines - in the $Q_i E$ plane - starting from $S_{ij}$.

[51]Suppose $Q_1 > Q_2 > Q_3$. If we now lower energies at fixed charges, we pass from the vacuum black hole to the Rank 2 charge 1 phase to the Rank 4 with charge (12) phase. If we now take $Q_1$ to $Q_2$ from above, the upper and lower end of the middle phase (rank 2 with charge 1) approach each other. When $Q_1 = Q_2$ we pass directly from the vacuum black hole to the rank 4 with charge (12) phase.

[52]We say a line is positive if each of the $Q_i$ are increasing - or more precisely are non decreasing - as $E$ increases.

As we have mentioned above, the upper boundary of the rank 6 DDBH phase has three components, namely $D_{12}$, $D_{23}$ and $D_{31}$. These three component boundaries meet at two dimensional sheets. For instance, the boundary between $D_{12}$ and $D_{13}$ is given by the set of upward pointing 45 degree lines, in the $Q_1 - E$ plane, shot out from $S_{123}$. But this is exactly the same as the two dimensional sheet we get at the intersection of $B_{1,2}$ and $B_{1,3}$. Both these sheets occur at $Q_2 = Q_3$, but with $Q_1$ larger than this common value. It follows that, at these charges, the lower bound of the $i^{th}$ rank 2 DDBH phase is the same as the upper bound of the rank 6 DDBH phase. Upon lowering energies at these charges, therefore, we pass directly from the rank 2 to the rank 6 phase.

Let us summarize. The most physical way to understand this phase diagram is to imagine holding the charges $Q_i$ fixed and lowering the energy starting from the vacuum black hole phase (see Fig. 1). Suppose $Q_1 > Q_2 > Q_3$. Upon lowering energies we go from the vacuum black hole phase to the rank 2 DDBHs in which the dual giants carry only $Q_1$ charge, to the rank 4 DDBHs in which the dual giants carry $Q_1$ and $Q_2$ charges, to the rank 6 DDBHs in which they carry all the three charges and finally hit the BPS bound (where the solution is a combination of three pure dual giants in global $AdS$). The thickness (range in energies) one spends in the rank 2 phase depends on $Q_1$ minus $Q_2$, and goes to zero when this difference goes to zero. When $Q_1 = Q_2$ we pass directly from the vacuum black hole to the rank 4 case. Similarly, thickness of the rank 4 phase depends on $Q_2$ minus $Q_3$ when this is zero we pass directly from the rank 2 to the rank 6 phase. Finally, in the special case $Q_1 = Q_2 = Q_3$ we pass directly from the vacuum black hole to the rank 6 DDBH phase. The above four cases[53] of the phase diagram are depicted in Figure 1.

In order to convey a sense of the global structure of the phase diagram, we have plotted a two dimensional slice of the four dimensional charge space (studied in this section), and the phase diagram in this cut of charge space. The two dimensional slice we have chosen is $\frac{Q_2}{N^2} = 3$ and $\frac{Q_3}{N^2} = 1.5$. In Fig 2 we plot the curves along which $\mu_i = 1$ hypersurfaces (for ordinary vacuum black holes) intersect our two dimensional slice of charge space. The 'stable' region (where vacuum black holes have $\mu_i < 1$ $\forall i$) is shaded blue, while the unstable region (where atleast one $\mu_i > 1$) is shaded green. In Fig 3, we present the actual phase diagram, that displays rank 2, rank 4 and rank 6 phases. An interesting point is that, over a small range of energies, the rank 2 DDBH phase with $Q_2$ has a core black hole with $\mu_1 < 1$ (and so continues to be stable to condensation of $Q_1$ charge) even though vacuum black holes at the same charge have $\mu_1 > 1$ (and so are locally unstable to the emission of $Q_1$ charge).

## 3.5 Phase diagram with $Q_1 = Q_2 = Q_3 = Q$, $J_1 = J_2 = J$

In this subsection we present our conjecture for the microcanonical phase diagram for $\mathcal{N} = 4$ Yang Mills theory on the special slice of charges $Q_1 = Q_2 = Q_3 = Q$, $J_1 = J_2 = J_L = J$. This phase diagram is presented as a function of $Q, J$ and the energy $E$ of all solutions.

The three dimensional space parameterized by $Q, J, E$ has four distinguished two dimensional surfaces. The three dimensional space of AdS black holes has four distinguished sheets.

- The BPS plane (the blue line in Fig 4).

---

[53] Above four cases arise when all $Q_i \neq 0$. When only one of the $Q_i = 0$, we get a similar phase diagram but with two cases: (i) $Q_j = Q_k$ - as we lower energy from vacuum BH phase, we directly go to the Rank 4 phase which exists until the BPS bound, (ii) $Q_j > Q_k$ - as we lower energy from vacuum BH phase, we first encounter a Rank 2 phase, followed by a Rank 4 DDBH phase which exists until the BPS bound. When two of the $Q_j = 0$ and the black hole carries only one of the $Q_i$, as we increase $Q_i$ from zero, we encounter a critical value of the non-zero charge $Q_i = Q_i^c$, until which the vacuum black hole phase exists all the way down to BPS energy, where the black hole becomes zero size. This behavior was first pointed out in [7]. For $Q_i < Q_c$, the DDBH black hole phase is absent. For $Q_i > Q_i^c$, as one lowers energy starting from the vacuum black hole phase, we encounter a Rank-2 DDBH phase which exists until BPS energy. See Appendix D for more details.

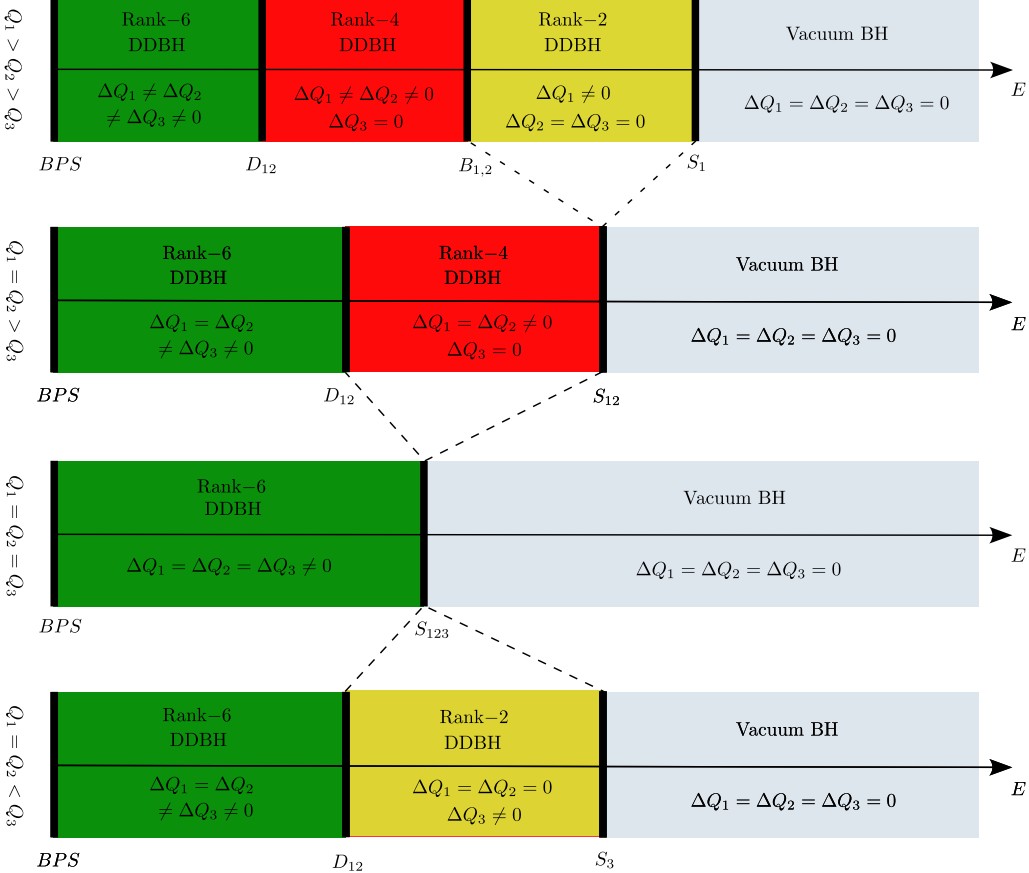

Figure 1: In this diagram we depict a three parameter set of straight lines within the four dimensional phase diagram of $\mathcal{N} = 4$ Yang Mills theory at vanishing values of angular momentum. Each of the lines we study is held at fixed values of the three charges $Q_i$, which, consequently, parameterize these lines. One moves along any of these lines by changing energy at fixed $Q_i$. At high energies (extreme right of all these diagrams) we are always in the vacuum black hole phase. Upon lowering the energy we undergo several phase transitions before hitting the BPS plane. The details of the phases we encounter on the way depends on the charges: we encounter 3 DDBH phases if none of the three $Q_i$ are equal, two such phases if two $Q_i$ are equal, or a unique rank 6 phase if they are all equal.

- The sheet of extremal black holes (the orange line in Fig 4).

- The sheet of black holes with $\omega = 1$ (the black line in Fig 4).

- The sheet of black holes with $\mu = 1$ (the red line in Fig 4).

Remarkably enough, these four sheets intersect on a single line:[54] the line of supersymmetric Gutowski-Reall black holes (the black dot in Fig. 4). All points below the $\mu = 1$ sheet are unstable to the emission of dual giants: the stable phase in this region is given by DDBH. The entropy of the DDBH phase is given by that of the $\mu = 1$ black hole at the end of the 45 degree line (in the $E\,Q$ plane, at constant $J$) that originates at the point of interest. For example, the entropy of the DDBH at the point $A$ in Fig 5, is that of the $\mu = 1$ black hole at the arrow tip in

---

[54]This 'remarkable' fact is of course well known and easy to understand. Susy black holes must have $\mu_i = \omega_j = 1$; this is how the Boltzmann factor $e^{-\beta(E - \mu_i Q_i - \omega_j J_j)}$ projects out all non BPS states as $\beta \to \infty$.

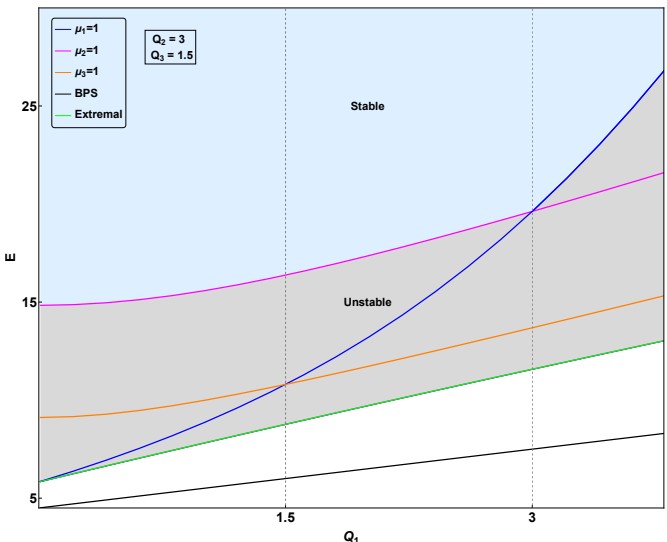

Figure 2: The intersection of the hypersurfaces $\mu_i = 1$ with the two dimensional plane $\frac{Q_2}{N^2} = 3$ and $\frac{Q_3}{N^2} = 1.5$.

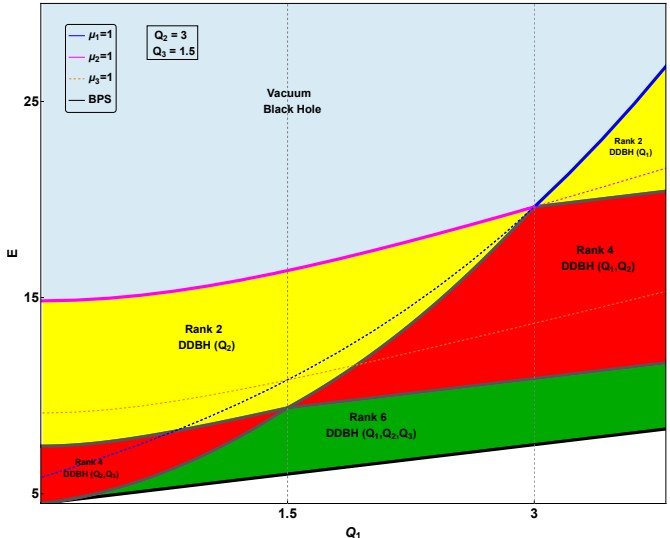

Figure 3: The phase diagram on the two dimensional plane $\frac{Q_2}{N^2} = 3$ and $\frac{Q_3}{N^2} = 1.5$.

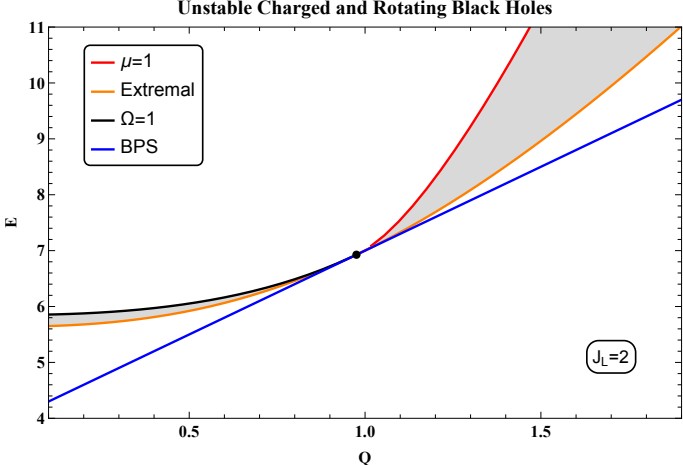

Figure 4: The intersection of the BPS plane (blue), the $\omega = 1$ sheet (black), the $\mu = 1$ sheet (red) and the sheet of extremal black holes (orange) with the slice $J_L = 2$. The black hole solutions of Cvetic-Lu-Pope (9) in the shaded grey region are unstable.

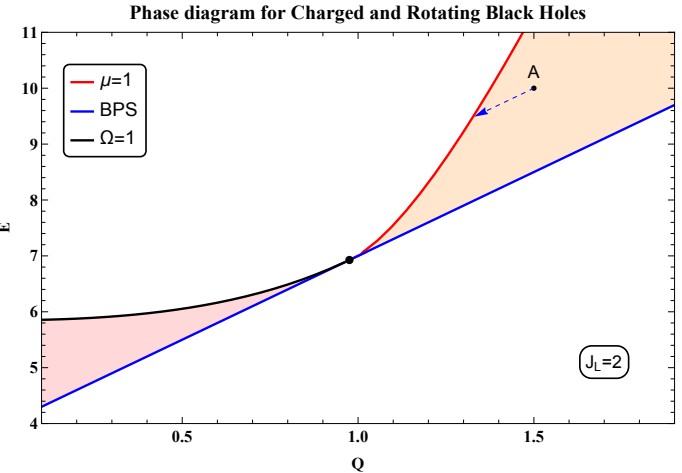

Figure 5: A constant $J$ slice of the phase diagram of $\mathcal{N} = 4$ Yang Mills theory, as a function of $E$ and $Q$. $E$ is the energy, $Q$ parameterizes the (equal) $SO(6)$ Cartans (which are given by $(Q, Q, Q)$. Here $J_1 = J_2 = J_L$ are all fixed at 2. The region between the $\mu = 1$ (red) curve and the BPS plane (blue line) is the DDBH phase. The region between the $\Omega = 1$ (black) curve and the BPS plane (blue line) is the Grey Galaxy phase. The white region in the upper left of the diagram (above the black and the red curves) is the vacuum black hole phase. The entropy of the point $A$ in the DDBH phase equals the entropy of the corresponding $\mu = 1$ black hole (end of the arrow) in the diagram.[55]

the figure.[56]

In a similar manner, all points below the $\omega = 1$ curve are unstable to the superradiant instability: the stable phase in this region is given by 5 dimensional grey galaxies (see [8]).

---

[55]To obtain the entropy of a black hole solution in the Grey Galaxy phase, one has to look at the constant $Q$ slice and draw a $45^o$ line from a given point to the black curve at constant $Q$. The entropy is then given by the black hole solution at the intersection of that line and the black curve.

[56]For large $J$, the $\mu = 1$ curve contains an interval with a negative slope. This does not affect the phase diagram of the DDBH, as the slope of the $\mu = 1$ curve remains greater than 45 degrees as long as it is positive. See Appendix C for a detailed explanation.

The entropy of the grey galaxy phase is given by that of the $\omega = 1$ black hole at the end of the 45 degree line (in the $E\,J$ plane, at constant $Q$) that originates at the point of interest.

A slice of this diagram at constant $J$ is depicted in Fig. 5.

## 3.6   The black brane scaling limit

Through this paper we have focussed on the study of black holes in global $AdS_5 \times S^5$, dual to $\mathcal{N} = 4$ Yang Mills theory on $S^3$. In this section we explore the coordinated large charge and large energy scaling limit of our results, with scalings chosen so as to turn vacuum black holes into vacuum black branes (and so to turn the results on $S^3$ into those on $R^3$). We work in the simplest nontrivial context, namely at energy $E$, zero angular momentum and $R$ charges $Q_1 = Q_2 = Q_3 = Q$. In this special case, it is easy to check (using the formulae presented in §2.1.4) that black holes with $\mu = 1$ carry energy and charge related by

$$E = 3Q^2 + 3Q \approx 3Q^2\,, \tag{58}$$

where the last approximation holds at large charge (all through this subsection, all charges are listed in units of $N^2$).

Let us now compare this to the scaling limit that turns vacuum black holes into vacuum black branes. Such a limit is achieved when we take

$$T = \frac{\tau}{\epsilon}\,, \qquad \mu = \frac{\nu}{\epsilon}\,,$$

and[57] scale $\epsilon \to 0$ at fixed $\tau$ and $\nu$. It follows from extensivity that, in this limit, the energy $E$ and charge $Q$ of vacuum black holes scale like $E \sim \frac{1}{\epsilon^4}$ and $Q \sim \frac{1}{\epsilon^3}$; more precisely we find

$$
\begin{aligned}
E &= \frac{3(1+x^2)}{4\epsilon^4}\,, \\
Q &= \frac{x}{2\epsilon^3}\,, \\
E &= cQ^{\frac{4}{3}}\,,
\end{aligned}
\tag{59}
$$

where $c = \frac{3}{4}(1 + x^2)\left(\frac{2}{x}\right)^{4/3}$ and $x$ is defined to satisfy $\frac{\nu}{\tau} = \frac{2\pi x}{2 - x^2}$. This scaling takes us to a black brane with chemical potential to temperature ratio given by $\frac{\mu}{T} = \frac{\nu}{\tau}$. The black brane metric takes its usual ($\epsilon$ independent) form, when written in terms of the coordinates

$$
\begin{aligned}
r' &= \epsilon r\,, \\
t' &= \frac{t}{\epsilon}\,, \\
x_i' &= \frac{\theta_i}{\epsilon}\,,
\end{aligned}
\tag{60}
$$

where $\theta_i$ are infinitesimal orthogonal angular coordinates (at any point in) the tangent space of $S^3$.[58]

The curve given in the third of (59) lies well below the curve (58) at any given fixed values of $\tau$ and $\nu$. It follows, therefore, that charged vacuum black branes are thermodynamically unstable to the nucleation of huge dual giants at *every nonzero value of the chemical potential to temperature ratio*. See §7.6 of [18] and [17, 28, 59] for a discussion of similar instabilities in various contexts.

---

[57]Note, immediately, that $\mu$ is taken to infinity in this scaling limit.

[58]The absolute value of the chemical potential and temperature of the black brane are unimportant as they can be changed by a scaling. The ratio $\frac{\mu}{T}$ is meaningful as it is scale invariant.

In order to work out the end point of the instability described in the previous paragraph, it is useful, once again, to return to the description of a black brane as a very highly charged (and massive) black hole, with energy and charge scaling like the third of (59). Let us start with a black brane whose charges are listed in the first two of (59). The non interacting model described in this subsection predicts that the end point is three dual giant gravitons, each carrying a charge of $Q_{gg}$

$$Q_{gg} = \frac{x}{2\epsilon^3} - \mathcal{O}\left(\frac{1}{\epsilon^2}\right),$$ (61)

together with a central black hole of charge $Q_{BH}$ and energy $E_{BH}$ given by

$$Q_{BH} = \frac{\sqrt{1+x^2}}{2\epsilon^2} + \dots,$$
$$E_{BH} = \frac{3(1+x^2)}{4\epsilon^4} - \mathcal{O}\left(\frac{1}{\epsilon^3}\right).$$ (62)

Note, that in the strict limit $\epsilon \to 0$, the black hole carries all the energy of the solution, while the dual giant carries all of its charge. It follows, therefore, that the entropy at these values of energies and charges is (to leading order in $\epsilon$) a function only of energy (independent of charge),[59] in complete contrast to the prediction that follows from usual charged black branes (whose entropy function is a nontrivial function of the charge as well as the energy density).

While the event horizon of the black hole is at $r \sim \frac{1}{\epsilon}$ (and so at a finite value of $r'$), the dual giant lives at the radial location

$$r = \sqrt{\frac{Nx}{2}} \frac{1}{\epsilon^{3/2}}, \qquad \text{i.e} \qquad r' = \sqrt{\frac{Nx}{2}} \frac{1}{\epsilon^{1/2}}.$$ (63)

In the strict scaling $\epsilon \to 0$, the dual giant lives at $r' = \infty$, and so in the deep UV of our field theory.[60]

In §5 below we compute the one shot decay rate of black holes into dual giants. In the scaling limit (60), it turns out that this decay rate scales like $e^{-\frac{\zeta}{\epsilon^3}}$ (where $\zeta$ is held fixed as $\epsilon$ is taken to zero) and so goes to zero in the strict $\epsilon \to 0$ limit. We believe that this result should be interpreted as follows. When measured in terms of the scaled variables $x'$ our boundary field theory has a spatial volume of order $\frac{1}{\epsilon^3}$. In field theory language, we are studying the decay of the 'false vacuum'. An instanton that mediates a one shot decay of the false vacuum, of course, has an action of order the volume, and so of order $\frac{1}{\epsilon^3}$. This fact is, however, irrelevant to the computation of the lifetime of the false vacuum. This decay proceeds via the nucleation of a finite size bubble of the true vacuum and its subsequent growth; the instanton governing such a nucleation process has finite action. Motivated by these considerations, we suspect that the decay of charged black branes also proceeds via bubble nucleation (so its rate is finite, though exponentially suppressed in $N$). We leave the further investigation of this interesting point to future work.

## 4 Classical dual giants around $Q_1 = Q_2 = Q_3 = Q$, $J_1 = J_2 = J$ black holes

In the rest of this paper, we analyze DDBHs in more detail, mainly focusing on the special case of DDBHs built around central black hole with $Q_1 = Q_2 = Q_3 = Q$ and $J_1 = J_2 = J$. In this

---

[59]We find the same result if we scale the energy and charge according to the relationship $E = Q^a$ for any value of $2 > a > 1$. Note $a = 1$ is the BPS curve.

[60]We could, of course, perform another scaling which brings the dual to a finite radial location. This scaling would, however, set the radial location of the black hole event horizon to zero.

section we review and generalize the discussion of [13, 29] to formulate and analyze the classical motion of probe dual giant gravitons on the background metric (1). In the next section, we quantize this motion. And in the subsequent section we present backreacted gravitational solutions for DDBHs.

## 4.1 World volume action for dual giant gravitons

All through this paper (see (1)) we work 'in units of $R_{AdS_5}$', i.e. we define the bulk metric $g_{\mu\nu}$ so that the bulk line element takes the form

$$ds^2 = R_{AdS_5}^2 \, g_{\mu\nu} dx^\mu dx^\nu \tag{64}$$

(see (1)). Correspondingly, we define the induced metric on the world volume of the probe D3-branes (that we will study in this section) as

$$ds^2 = R_{AdS_5}^2 \, h_{ab} dx^a dx^b \,. \tag{65}$$

We also work with a similarly rescaled 4-form potential $C$ and 5-form field strength $F_5$, see ((24) and (26)). In terms of these rescaled quantities, the D-brane action (A.6) (with $F_{ab}$ set to zero) can be rewritten as

$$
\begin{aligned}
S &= \frac{R_{AdS_5}^4}{(2\pi)^3 \alpha'^2 g_s} \left( -\int d^4 y \sqrt{-\det h_{ab}} + \frac{1}{4!} \int \epsilon^{abcd} C_{\mu_1\mu_2\mu_3\mu_4} \frac{dx^{\mu_1}}{dy^a} \frac{dx^{\mu_2}}{dy^b} \frac{dx^{\mu_3}}{dy^c} \frac{dx^{\mu_4}}{dy^d} d^4 y \right) \\
&= \frac{N}{2\pi^2} \left( -\int d^4 y \sqrt{-\det h_{ab}} + \frac{1}{4!} \int \epsilon^{abcd} C_{\mu_1\mu_2\mu_3\mu_4} \frac{dx^{\mu_1}}{dy^a} \frac{dx^{\mu_2}}{dy^b} \frac{dx^{\mu_3}}{dy^c} \frac{dx^{\mu_4}}{dy^d} d^4 y \right),
\end{aligned}
\tag{66}
$$

where we have used $R_{AdS_5}^4 = 4\pi g N \alpha'^2$.

(66) holds for any D3-brane motion in the metric (1). We now restrict (66) to 'dual giant graviton' configurations, defined as follows. Recall that black hole spacetime (1) has an $SU(2)_R \times U(1)_L$ killing symmetry.[61] We define 'dual giant gravitons' as D3-brane configurations that preserve this rotational symmetry group.[62] In other words, dual giant gravitons are D3-brane configurations chosen so that $SU(2)_R \times U(1)_L$ killing vectors are tangent to the D3-brane.[63] In the coordinates of (9), this condition is satisfied provided the vectors $\partial_{\theta_a}, \partial_\phi$ and $\partial_\psi$ are tangent to the D3-brane world volume. In other words, dual giant D3-brane configurations necessarily wrap the warped $S^3$ (spanned by $\theta_a, \phi$ and $\psi$) at every given value of $t$, $r$ and $S^5$ coordinates. In other words dual giant graviton action is that for an effective point particle propagating in an effective 6+1 dimensional spacetime (parameterized by $t$, $r$ and $S^5$ coordinates). In Appendix E we show that the effective action in this seven dimensional spacetime is given by

$$
\begin{aligned}
S &= S_{DBI} + S_{WZ} \\
&= 2N \int r^2 \sqrt{b^2 + A_{\sigma_3}^2} \sqrt{-\tilde{g}_{\mu\nu} dx^\mu dx^\nu} + 8N \int \left[ \left( \frac{r^4 - R^4}{8} \right) dt \right],
\end{aligned}
\tag{67}
$$

---

[61]This killing symmetry group is defined to be generated by the killing vectors that reduce, at infinity, to the usual generators of $SU(2)_R$ and $U(1)_L$. This last requirement clearly specifies the $U(1)_L$, distinguishing it, for instance, from an admixtures of the rotational $U(1)_L$ (as defined above) with time translations.

[62]This also means that the brane does not carry any angular momentum along the $S^3$ in $AdS_5$. Following thermodynamic arguments in section 3, it is easy to see that it is thermodynamically unfavorable for the black hole with $\Omega < 1$(which is the regime we will be interested in) to emit angular momentum.

[63]The condition of $SU(2)_R \times U(1)_L$ invariance is simply implemented in the coordinates used in (9) (recall that this metric was written in a gauge in which the gauge field vanishes at infinity rather than at the horizon). If we view the codimension 6 D3-brane world volume as being specified by 6 equations on bulk coordinates, the requirement of symmetry under $SU(2)_R \times U(1)_L$ is satisfied provided that each of these six equations are $SU(2)_R \times U(1)_L$ invariant, i.e. are independent of the three angular coordinates $\theta, \psi, \phi$.

where

$$ds_7^2 = \tilde{g}_{\mu\nu}dx^\mu dx^\nu = -\frac{r^2 W}{4b^2}dt^2 + \frac{dr^2}{W} + ds^2(\mathbb{CP}^2) + \frac{b^2}{b^2 + A_{\sigma_3}^2}\left(d\Psi + \Theta + (fA_{\sigma_3} - A_t)dt\right)^2, \quad (68)$$

where the gauge fields $A_t$ and $A_{\sigma_3}$ are $dt$ and $d\sigma_3$ components of the gauge field listed in (9), the function $b(r)$, $W(r)$ and $f(r)$ are, respectively, defined in (16), (18) and (17).

In the rest of this section we will investigate solutions to the action (67). We will find it convenient to use the notation

$$\Delta = \tilde{g}_{\mu\nu}\frac{dx^\mu}{dt}\frac{dx^\nu}{dt},$$
$$m = 2r^2\sqrt{b^2 + A_{\sigma_3}^2}. \quad (69)$$

Note that the 'effective particle mass' $m$ is a function of the radial coordinate $r$ in $AdS_5$.

## 4.2  Charge matrix for dual giant solutions

The Lagrangian (67) enjoys invariance under the isometry group $U(3)$. As a consequence, dual giant motion is subject to 9 conserved charges, one for each of the 9 generators of $U(3)$. Two questions immediately pose themselves.

- On scanning overall solutions to the equations of motion that follow from (67), do we find all possible $U(3)$ charges, or do we, instead, obtain a constrained set of such charges?

- Once we have established which class of $U(3)$ charges appear in our analysis, do we have to analyze each charge separately, or do the charges organize themselves into a smaller set of equivalence classes of $U(3)$?

The questions posed above are answered in Appendix F, where we demonstrate that the set of all solutions of (67) produce only those $U(3)$ charge matrices that are $U(3)$ similarity equivalent to the matrix

$$\begin{pmatrix} q_1 & 0 & 0 \\ 0 & q_2 & 0 \\ 0 & 0 & 0 \end{pmatrix}, \quad (70)$$

where the two charges $q_1$ and $q_2$ necessarily carry opposite signs. For given generic (i.e. nonzero and non equal) values of $q_1$ and $q_2$, the action of $U(3)$ similarity transformations on (70) produces a 6 parameter set of $U(3)$ charges. Varying over all $q_1$ and $q_2$ gives an 8 parameter set of such charges. In other words matrix of $U(3)$ charges is subject to a single restriction, namely that its determinant vanishes (this condition cuts down the 9 parameters in unconstrained $U(3)$ charge matrices to 8 parameters in the set of $U(3)$ matrices similarity equivalent to (70).[64]) These 8 charges, together with two angular initial conditions, give us 10 coordinates in phase space.[65]

The answer to the second question posed above is now also clear. Though the $U(3)$ charge matrix for generic motion has 8 parameters, these charges arrange themselves into equivalence classes (under $U(3)$ similarity transformations). These equivalence classes are labeled by the two charges $q_1$ and $q_2$. Consequently, it is sufficient to study the solutions whose $U(3)$ charges

---

[64]The special case $q_2 = 0$ will turn out to be of particular physical relevance. The $U(3)$ orbits of such charges are 4 dimensional (these orbits are $U(3)/(U(2) \times U(1)))$ giving a 5 parameter set of such charges.

[65]As we are studying the motion of an effective particle in 6 spatial dimensions, phase space is 12 dimensional. The remaining two coordinates on phase space can be chosen to be the conserved energy of the solution and a radial initial location.

are given by (70). All other solutions (solutions with all other similarity equivalent charges) can be obtained from these by performing the appropriate $U(3)$ rotation.

The Noether procedure yields an expression for the corresponding (conserved) charge matrix $Q_{ij}$. $Q_{ij}$ is a $U(3)$ adjoint element, and so is a $3 \times 3$ Hermitian matrix. We find

$$Q_{ij} = \frac{i}{2}\left(z_i p_{z_j} - \bar{z}_j p_{\bar{z}_i}\right),\tag{71}$$

where[66]

$$
\begin{aligned}
z_i &= l_i e^{i\phi_i}, \\
p_{z_i} &= \frac{\partial L}{\partial \dot{z}_i} = -\frac{i}{z_i}\left(\frac{\partial L}{\partial \dot{\phi}_i}\right) + \frac{l_i}{z_i}\frac{\partial L}{\partial \dot{l}_i}, \\
p_{\bar{z}_i} &= \frac{\partial L}{\partial \dot{\bar{z}}_i} = \frac{i}{\bar{z}_i}\left(\frac{\partial L}{\partial \dot{\phi}_i}\right) + \frac{l_i}{\bar{z}_i}\frac{\partial L}{\partial \dot{l}_i}.
\end{aligned}
\tag{72}
$$

The Noether procedure yields expressions for the charges in terms of $\dot{\phi}_i$, $l_i$ and $\dot{l}_i$. It proves useful to eliminate the three $\dot{\phi}_i$ in favour of the three diagonal charges (dual to translations of the angles $\phi_i$)

$$Q_{ii} \equiv N q_i = \frac{\partial L}{\partial \dot{\phi}_i},\tag{73}$$

and present our expression for $Q_{ij}$ as a function of $q_i$, $l_i$ and $\dot{l}_i$. We find

$$N\begin{pmatrix} q_1 & \frac{e^{i(\phi_1-\phi_2)}\left(l_2^2 q_1 + l_1^2 q_2 - 2il_1^2 l_2^2(\dot{l}_1-\dot{l}_2)\right)}{2l_1 l_2} & \frac{e^{i(\phi_1-\phi_3)}\left(l_3^2 q_1 + l_1^2 q_3 - 2il_1^2 l_3^2(\dot{l}_1-\dot{l}_3)\right)}{2l_1 l_3} \\ -\frac{e^{-i(\phi_1-\phi_2)}\left(l_2^2 q_1 + l_1^2 q_2 - 2il_1^2 l_2^2(\dot{l}_1-\dot{l}_2)\right)}{2l_1 l_2} & q_2 & \frac{e^{i(\phi_2-\phi_3)}\left(l_3^2 q_2 + l_2^2 q_3 - 2il_2^2 l_3^2(\dot{l}_2-\dot{l}_3)\right)}{2l_2 l_3} \\ -\frac{e^{-i(\phi_1-\phi_3)}\left(l_3^2 q_1 + l_1^2 q_3 - 2il_1^2 l_3^2(\dot{l}_1-\dot{l}_3)\right)}{2l_1 l_3} & -\frac{e^{-i(\phi_2-\phi_3)}\left(l_3^2 q_2 + l_2^2 q_3 - 2il_2^2 l_3^2(\dot{l}_2-\dot{l}_3)\right)}{2l_2 l_3} & q_3 \end{pmatrix}.\tag{74}$$

As we have explained in the previous subsection, the most general solution is $U(3)$ equivalent to a solution with diagonal $U(3)$ charges with two nonzero diagonal elements. We choose these nonzero elements to be $q_1$ and $q_2$, and set $q_3$ to zero (i.e. we choose the $U(3)$ matrix to be of the form (70)). Setting all off-diagonal charges - and $q_3$ - to zero in (74) yields the equations

$$\dot{l}_1 = \dot{l}_2 = 0, \qquad l_3 = 0, \qquad \frac{l_1^2}{q_1} = -\frac{l_2^2}{q_2}.\tag{75}$$

These equations (together with the constraint $\sum_i l_i^2 = 1$) tell us that the three $l_i$ take the following constant values

$$l_1 = \sqrt{\frac{q_1}{q_1 - q_2}}, \qquad l_2 = \sqrt{\frac{q_2}{q_2 - q_1}}, \qquad l_3 = 0.\tag{76}$$

Note $q_1$ and $q_2$ have opposite signs, in agreement with (F.5).

In addition, the (rather complicated) equations that determine $\dot{\phi}_1$ $\dot{\phi}_2$ in terms of $q_1, q_2$ are

$$
\begin{aligned}
q_1 &= m\frac{l_1^2\dot{\phi}_1 - l_1^2\left(A_t - \frac{A_t A_{\sigma_3} + b^2 f - \sum_i\left(l_i^2 A_{\sigma_3}\dot{\phi}_i\right)}{b^2 + A_{\sigma_3}^2}A_{\sigma_3}\right)}{\sqrt{-\Delta}}, \\
q_2 &= m\frac{l_2^2\dot{\phi}_2 - l_2^2\left(A_t - \frac{A_t A_{\sigma_3} + b^2 f - \sum_i\left(l_i^2 A_{\sigma_3}\dot{\phi}_i\right)}{b^2 + A_{\sigma_3}^2}A_{\sigma_3}\right)}{\sqrt{-\Delta}},
\end{aligned}
\tag{77}
$$

where $\Delta$ and $m$ were defined in (69).

---

[66]The expression $\frac{\partial L}{\partial \dot{z}_i}$ can be understood as follows. We set $z_i = l_i e^{\phi_i}$ and pretend that the $l_i$ are unconstrained. This procedure works because the $U(3)$ generators are themselves tangent to the surface $l_1^2 + l_2^2 + l_3^2 = 1$.

## 4.3 The 'Routhian' and Hamiltonian at fixed charges

Once we have fixed charges as above, the motion of our effective particle in $r$ (as a function of time) is obtained in the usual manner (i.e. from the principle of least action) that follows from the 'Routhian'

$$L_{\text{eff}} = L - p_i \dot{\phi}_i. \tag{78}$$

Using (77) and (76), we find the Routhian to be

$$
L_{\text{eff}} = N\left( -\frac{1}{2b^2} \sqrt{\frac{(r^2 W^2 - 4b^2 \dot{r}^2)\left((q_1+q_2)^2 A_{\sigma_3}^2 + b^2 (m^2 + (q_1-q_2)^2)\right)}{W}} \right.
$$
$$
\left. + (q_1+q_2)\left(f A_{\sigma_3} - A_t\right) + r^4 - R^4 \right). \tag{79}
$$

The Hamiltonian that follows from (79) is

$$H = N\left( \frac{r\sqrt{W}}{2b^2} \sqrt{(q_1+q_2)^2 A_{\sigma_3}^2 + b^2\left(m^2 + W q_r^2 + (q_1-q_2)^2\right)} + (q_1+q_2)\left(A_t - f A_{\sigma_3}\right) - r^4 + R^4 \right), \tag{80}$$

where $q_r$ is the momentum conjugate to $\dot{r}$. Setting $q_r$ to zero in (83) yields the effective potential

$$V_{\text{cl}}(r) = N\left( \frac{r\sqrt{W}}{2b^2} \sqrt{(q_1+q_2)^2 A_{\sigma_3}^2 + b^2 (m^2 + (q_1-q_2)^2)} + (q_1+q_2)\left(A_t - f A_{\sigma_3}\right) - r^4 + R^4 \right). \tag{81}$$

It is important to keep in mind that $q_2$ and $q_1$ have opposite signs, so $|q_1-q_2| > |q_1+q_2|$. Also, we are in the gauge such that $V_{cl}(R) = 0$.

In Fig. 6 we present a plot of the effective potential for black holes at two different values of the black hole chemical potential.

### 4.3.1 Special case of non rotating black holes

In the case of non rotating black holes (i.e. when $j = 0$) $f(r)$ and $A_{\sigma_3}$ both vanish (see (16), (17) and (18)), and the expressions presented above simplify somewhat to

$$L_{\text{eff}} = N\left( -(q_1+q_2)A_t(r) - \sqrt{\frac{(r^6 + (q_1-q_2)^2)(W(r)^2 - \dot{r}^2)}{W(r)}} + r^4 - R^4 \right), \tag{82}$$

$$H = N\left( (q_1+q_2)A_t + \sqrt{W(r)\left((q_1-q_2)^2 + q_r^2 W(r) + r^6\right)} - r^4 + R^4 \right), \tag{83}$$

and

$$V_{\text{cl}}(r) = N\left( (q_1+q_2)A_t + \sqrt{W(r)\left((q_1-q_2)^2 + r^6\right)} - r^4 + R^4 \right), \tag{84}$$

with

$$
W(r) = \frac{(r-R)(r+R)\left(r^4 + r^2(R^2+1) - \mu^2 R^2\right)}{r^4},
$$
$$
A_t(r) = \mu\left(\frac{R^2}{r^2} - 1\right). \tag{85}
$$

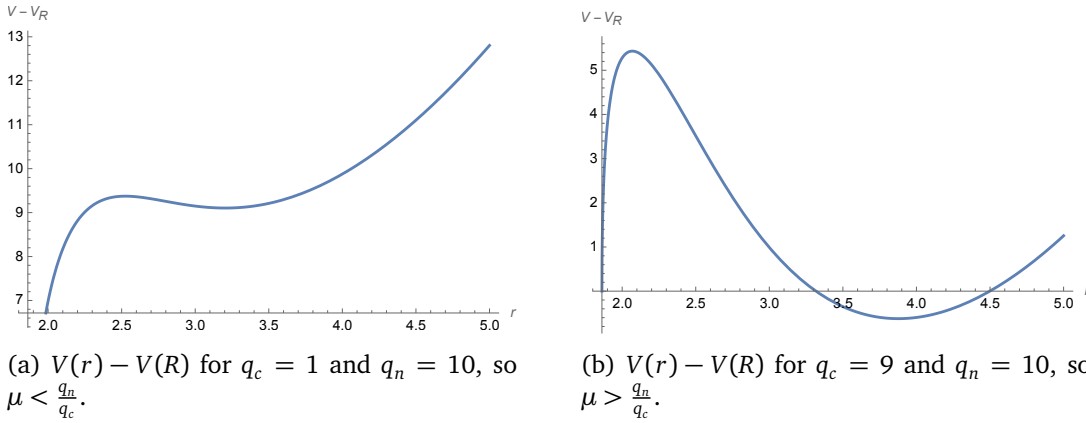

(a) $V(r) - V(R)$ for $q_c = 1$ and $q_n = 10$, so $\mu < \frac{q_n}{q_c}$.

(b) $V(r) - V(R)$ for $q_c = 9$ and $q_n = 10$, so $\mu > \frac{q_n}{q_c}$.

Figure 6: Energy of the probe at distance $r$ minus the energy at the horizon. The above graphs are plotted for black holes with $p = 19.236$, $q = 6$, $j = 0.12318$ and therefore $\mu = 1.68$. For $\mu > \frac{q_n}{q_c}$, the energy at minima is less than $V(R)$ but for $\mu < \frac{q_n}{q_c}$, energy at minima is greater than $V(R)$.

## 4.4 More about the effective potential

Let us define

$$
\begin{aligned}
q_c &= q_1 + q_2, \\
q_n &= q_1 - q_2.
\end{aligned}
\tag{86}
$$

$q_c$ is the $U(1)$ charge of our probe (the $U(1)$ in question is the diagonal part of $U(3)$).[67] Recall that $q_n > q_c$. Note that while $q_n$ is always positive, $q_c$ can be of either sign.

The effective potential can be rewritten as

$$
V_{\rm cl}(r) = \frac{Nr\sqrt{W}}{2b^2}\sqrt{q_c^2 A_{\sigma_3}^2 + b^2\left(m^2 + q_n^2\right)} + q_c\left(A_t - fA_{\sigma_3}\right) - r^4 + R^4.
\tag{87}
$$

### 4.4.1 $V_{\rm cl}(r)$ in the neighbourhood of the horizon

The expression (87) for the potential simplifies at $r = R$ (here $R$ is the radius of the outer event horizon). Working in the gauge (10) we find (see (14)) that

$$
V_{\rm cl}(R) = N\mu q_c,
\tag{88}
$$

a result that might have been anticipated on physical grounds.[68] On the other hand $V_{\rm cl}(R) = 0$ in the gauge (15).

In both gauges, It is also possible to check that $V_{\rm cl}'(R) > 0$ for all values of $\mu$ ($R$ is the radius of the outer horizon of the black hole). Consequently, the potential increases as we move away from the horizon, towards larger values of $r$.

---

[67]$q_n$ can roughly be understood as follows: in the case that the breaking of $SO(6)$ down to $U(3)$ can be ignored, $q_n$ is the unique nonzero eigenvalue of the $SO(6)$ charge matrix corresponding to the probe D3-brane. For this reason it determines the mass of the dual solution in the limit that $q_n$ is large (so that the effect of the black hole can be ignored).

[68]Recall that the ten dimensional generator of the killing horizon has zero norm at the horizon. This suggests that the charge $E - \Omega J - \mu q_c$ of this probe will vanish, as the red shift factor that converts the proper 'momentum' into charge vanishes at the horizon. Note that our probe carries $J = 0$, and so has $E = \mu q_c$ when at the horizon. Note the gauge of (10) is well suited to the analysis of energy, as the metric reduces to $AdS_5 \times S^5$ in this gauge.

### 4.4.2   $V_{\text{cl}}(r)$ at large $r$

At large $r$ it may be checked that[69]

$$V_{\text{cl}}(r) = N\frac{r^2}{2} + \mathcal{O}(1), \tag{89}$$

so that $V_{\text{cl}}(r)$ increases quadratically at $r = \infty$.

### 4.4.3   $V_{\text{cl}}(r)$ in a coordinated large $q_n$ and large $r$ limit

(89) correctly captures the behavior of the potential when $r$ is the largest quantity around. When $q_n \sim q_c$ are also large then the effective potential takes a more interesting form for $r \sim \sqrt{q_n}$.

   We work in the gauge (10), and take $r$ and $q_n$ both large with $\frac{q_n}{r^2}$ held fixed, the effective potential simplifies to the potential in *AdS* space, namely[70,71]

$$V_{\text{cl}}(r) = N\left(\sqrt{(1+r^2)(q_n^2 + r^6)} - r^4\right). \tag{90}$$

$V_{\text{cl}}(r)$ in (90) has a minimum at $r = \sqrt{q_n}$, and that the value of this potential at the minimum equals $q_n$ (in agreement with the BPS bound and the analysis of [12]).

   In order to compute the location and energy of the probe D-brane more accurately (in a power series in $\frac{1}{q_n}$) we expand the potential $V_{\text{cl}}(r)$ in (87) in a power series in $\frac{1}{q_n} \sim \frac{1}{q_c} \sim \frac{1}{r^2} \sim \mathcal{O}(\epsilon)$.

   We find

$$V_{\text{cl}}(r) = N\left(\frac{q_n^2 + r^4}{2r^2} + \left((j^2 - 1)p + \frac{qq_c}{r^2} + q - \frac{(q_n^2 - r^4)^2}{8r^8} + R^4\right)\right), \tag{91}$$

where the first term in above equation is $\mathcal{O}(\frac{1}{\epsilon})$ and the second term is $\mathcal{O}(1)$ and we have neglected all terms that are $\mathcal{O}(\epsilon)$ or higher. We find the minimum of $V_{\text{cl}}(r)$ by solving the equation $V'_{\text{cl}}(r_{min}) = 0$ order by order in $\epsilon$ (at the end of this exercise we set the formal parameter $\epsilon$ to unity: physically the role of $\epsilon$ is played by inverse charges). We find

$$r_{\min} = \sqrt{|q_{\text{n}}|} + \frac{qq_c}{2q_n^{3/2}} \tag{92}$$

(here $q$ is the parameter that enters in the black hole solution). We also find

$$E = V_{\text{cl}}(r_{\min}) = N\left(q_n + \left((j^2 - 1)p + \frac{qq_c}{q_n} + q + R^4\right)\right). \tag{93}$$

   Assuming that $q_n$ and $q_c$ are comparable, it follows that the first correction to the leading energy of the probe occurs at relative order $1/q_n$.

   From (93) and (88), it follow that (at large $q_n$) $V_{\text{cl}}(r_{\min}) > V_{\text{cl}}(R)$ for $\mu < \frac{q_n}{q_c}$, while $V_{\text{cl}}(r_{\min}) < V_{\text{cl}}(R)$ for $\mu > \frac{q_n}{q_c}$ (see fig. 6). This point - which was observed in [13,30] - already suggests that dual giants should be unstable $\mu < \frac{q_n}{q_c}$ (as they would like to tunnel into the black hole), but should be stable at $\mu > \frac{q_n}{q_c}$. In the next section we will see that this expectation is correct.

---

[69]This result is true both in the gauge (10) as well as the gauge (15).

[70]In the rest of this section we present the potential $V_{\text{cl}}(r)$ in the gauge (10). The potential in the gauge (15) may be obtained by subtracting $\mu q_c$ from the expressions for $V(r)$ presented below.

[71](89) can be checked to follow from a Taylor expansion of (90) at large $r$.

### 4.5 Thermodynamic significance

At leading order the charge to energy ratio of our solution is $\frac{q_c}{q_n}$ (note that the modulus of this ratio is smaller than unity). The discussion in the introduction thus suggests that our black hole is thermodynamically unstable to the super radiant emission of large charge dual giants whenever

$$\mu_c > \frac{q_n}{q_c}. \tag{94}$$

Clearly, the solution that first go unstable are those with $q_n = q_c$. In this case (93) simplifies to

$$E = V_{\rm cl}(r_{\rm min}) = N\left(q_n + \left((j^2 - 1)p + 2q + R^4\right)\right). \tag{95}$$

We have checked numerically that

$$(j^2 - 1)p + 2q + R^4 \tag{96}$$

is always positive for black holes with $\mu = 1$[72] and so the charge to energy ratio of these probe solutions is largest in the limit that $q_c$ is large.[73] As the finite charge corrections to the energy of dual giants is positive, dual giants have the largest charge to energy ratio at large $q_n = q_c$. Thermodynamically, this is yet another reason why black holes with $\mu > 1$ 'want' to decay into a single, very large dual giant, with $q_c = q_n$, surrounding the central black hole.

### 4.5.1 Non-rotating black holes

For the special case of non-rotating black holes, using (85), we can express the equations (91), (92), and (93) in terms of the chemical potential of the black hole and the radius of horizon.

$$V_{\rm cl}(r) = N\left(\frac{q_n^2 + r^4}{2r^2} + \frac{-q_n^4 + 2q_n^2 r^4 + 8\mu q_c r^6 R^2 + r^8\left(4R^4 - 4\left(\mu^2 + 1\right)R^2 - 1\right)}{8r^8}\right), \tag{97}$$

which has a minimum at

$$r_{\rm min} = \sqrt{q_n} + \frac{\left(\mu q_c R^2\right)}{2q_n^{3/2}} \tag{98}$$

(where we have set $\epsilon$ to one). The energy at the minima is given by

$$E = V_{\rm cl}(r_{\rm min}) = q_n + \frac{R^2\left(q_n\left(-\mu^2 + R^2 - 1\right) + 2\mu q_c\right)}{2q_n}. \tag{99}$$

We have checked (analytically for $j = 0$ and numerically for $j \neq 0$) that for $\mu = 1$, the correction to the energy (i.e. the second term in the above equation) is always positive (for $q_c = q_n$) and hence the charge to energy ratio of these probe solutions is largest in the limit that $q_c$ is large.

As we have mentioned above, solutions with $q_c = q_n$ have the largest charge to mass ratio. As these are the solutions that will show up in the thermodynamically stable DDBHs, we pause to note that there is a 6 parameter family of these solutions (recall they carry $q_2 = 0$ and $q_c = q_1$), parameterized by $q_c$, the four elements of $U(3)/(U(2) \times U(1))$ and one initial angular variable.

As we will explain in detail in the next section (and the Appendices, see §H.3), the quantization of this phase space produces $U(3)$ representations with $n$ boxes in the first row of the

---

[72]Except in the case of susy black holes for which (96) vanishes. This is expected, as such duals are susy and so should obey the BPS bound $E = Nq_n$.

[73]The argument presented above establishes this fact only at large $q_c$, but we believe this result holds generally: we have checked this numerically at several (allowed) values of $p$, $q$ and $j$.

Young Tableuax, and no boxes in any other row. The approximate map between $q_c$ and $n$ is $q_c \approx nN$.[74]

## 4.6 Minima of the effective potential away from the large $q_n$ limit

Away from the large charge limit, the potential $V_{cl}(r)$ is complicated: depending on details, this potential has either zero, one or two minima outside the horizon. One way to organize thinking about this problem is to hold all black hole parameters, as well as the ratio $\alpha = \frac{q_c}{q_n}$ fixed, and study the potential $V_{cl}(r)$ as a function of $q_n$, with $q_n$ varying from $\infty$ down to zero. In fact it is convenient to take $\alpha$ and $-\alpha$ in one line, and have $q_n$ varying from $\infty$ down to $-\infty$ (where by negative values of $q_n$ we mean the modulus value of $q_n$ but with negative $\alpha$. As we perform the scan described above, we see many different behaviors as we change black hole parameters and $|\alpha| \in (0, 1)$.

For one set of black hole and parameters (roughly when the black hole is small), $V_{cl}$ (of course) admits a single minimum at large positive $q_n$. As $q_n$ is decreased this minimum disappears into the horizon. For a range of lower $q_n$ the potential has no minima. At still lower $q_n$, a new minimum pops out of the horizon, and survives all the way to $q_n = -\infty$.

For another set of black hole and $\alpha$ parameters (roughly for larger black holes), we (of course) have one minimum at large $q_n$. As $q_n$ is decreased another minimum (together with a new maximum) appears, so that $V_{cl}$ has two minima. As $q_n$ is further decreased, first the larger minimum merges with a neighboring maximum and disappears. Then the smaller minimum merges with a neighboring maximum and disappears. For a range of lower charges the potential has no minima. On further lowering $q_n$ a new minimum emerges out of the horizon, and continues to exist all the way down to $q_n = -\infty$

At other values of black hole parameters and $\alpha$, still other behaviors can occur. As solutions at finite values of $q_n$ will play no role in what follows, we do not investigate this further, leaving further development of this analysis to the interested reader.

## 4.7 Dual giants around SUSY black holes

The three parameter set of CLP black holes (9) become supersymmetric on a codimension two surface (a curve). Black holes on this curve are called Gutowski-Reall or GR black holes [20,21]. These special black holes can be parameterized by their outer horizon radius $R$. Their metric is given by (9) with

$$q = R^2 \left(1 + \frac{R^2}{2}\right), \tag{100}$$

$$p = 2R^2 \left(1 + \frac{R^2}{2}\right)^2, \tag{101}$$

$$j = \frac{R^2}{2l} \left(1 + \frac{R^2}{2}\right)^{-1}. \tag{102}$$

The authors of [32] have demonstrated that GR black holes admit supersymmetric dual giants even at finite values of the charge of the dual. As a consistency check of our formulae, we rederive this point using the effective potential $V_{cl}(r)$.

A supersymmetric probe must obey the BPS equation $E = q_c$. Clearly, the probes that are most likely to satisfy this relation are those with the smallest mass to charge ratio, i.e. those

---

[74]This quantization was first performed in pure AdS in e.g. [31] (the solutions above are singled out as the are $1/8^{th}$ BPS in that context).

with $|q_c| = q_n$. Making this choice and substituting (100) in (87), we find that the potential function simplifies to

$$V_{\rm cl}(r) = q_c + \sqrt{A^2 + B} - A, \tag{103}$$

where $A$ and $B$ are functions of $r$ given by

$$\begin{aligned}
A &= q_c \left( \frac{2(r^2 - R^2)\left(2r^4 - r^2 R^4 + R^6\right)}{4r^6 + 4r^2 R^6 - R^8} \right) + r^4 - R^4, \\
B &= \frac{\left(r^2 - R^2\right)^2 \left(-2q_c \left(r^2 + R^2\right) + 2r^4 - r^2 R^4 + R^6\right)^2}{4r^6 + 4r^2 R^6 - R^8}.
\end{aligned} \tag{104}$$

Note that the numerator of $B$ is a perfect square, while its denominator is clearly positive for $r > R$. It follows that $B \geq 0$ for $r > R$. It follows that

$$\sqrt{A^2 + B} - A \geq 0, \tag{105}$$

and that the inequality is saturated only at a value of $r$ at which $B = 0$ and $A > 0$. Clearly, a value of $r > R$ that meets these conditions is a global - and so, in particular, a local - minimum of $V_{\rm cl}(r)$. At values of $r$ at which $B$ vanishes, $V_{\rm cl}(r) = q_c$ and so the solution obeys the BPS bound and so is supersymmetric (see Appendix G and [32] for a direct $\kappa$ symmetry verification of this point).[75]

We have seen that supersymmetric minima occur when $B = 0$, i.e. when

$$q_c = f(r); \qquad f(r) = \frac{2r^4 - r^2 R^4 + R^6}{2\left(r^2 + R^2\right)}. \tag{106}$$

One can solve (106) to determine $r^2$ as a function of $q_c$.[76,77] However, the question of existence of solutions is more easily understood simply by using (106) to plot $f(r)$ versus $r^2$. The nature of this plot changes depending on the value of $R$. Let us first note that $f(R) = \frac{R^2}{2}$. When $R < \sqrt{3}$, then $f(r)$ then increases monotonically as a function of $r$ for all $r$. It follows immediately, that in this case, we have a single supersymmetric minimum of $V_{\rm cl}(r)$ when $q_c > \frac{R^2}{2}$, and no such minima when $q_c < \frac{R^2}{2}$.[78] When $R \geq \sqrt{3}$, $f(r)$ first decreases (as $r$ is increased starting from $R$, reaches a minimum at $r^2 = R^2(-1 + \sqrt{1 + R^2})$ (at this value $f(r) = \frac{R^2}{2}\left(4\sqrt{R^2 + 1} - R^2 - 4\right)$.) and then increases monotonically at all larger $r$. We see that, in this case, $V(r)$ has two supersymmetric minima when

$$\frac{R^2}{2}\left(4\sqrt{R^2 + 1} - R^2 - 4\right) \leq q_c \leq \frac{R^2}{2},$$

but a single supersymmetric minimum when $q_c > \frac{R^2}{2}$.[79] Note that the lower limit of this range is negative for $R > 2\sqrt{2}$. Consequently, at these values of $R$ we have a supersymmetric minimum of $V_{\rm cl}(r)$ over a range of negative values for $q_c$.

---

[75]In addition, $V_{\rm cl}(r)$ also has an additional nonsupersymmetric minimum at small enough values of $q_c$.

[76]We find a quadratic equation for $r^2$.

[77]

$$r^2 = \frac{R^4 + 2q_c \pm \sqrt{(R^4 + 2q_c)^2 - 8\left(R^4 - 2q_c\right)R^2}}{4}. \tag{107}$$

[78]In this case, i.e. when $R < \sqrt{3}$ we also have a single nonsupersymmetric below a critical value of $q_c$, smaller than $\frac{R^2}{2}$ and all the way down to $-\infty$.

[79]We believe that $V_{\rm cl}(r)$ has a nonsupersymmetric minimum for all values of $q_c < \frac{R^2}{2}\left(4\sqrt{R^2 + 1} - R^2 - 4\right)$.

It is not difficult to verify that $\dot{\phi}_1 = \dot{\phi}_2 = \dot{\phi}_3 = 1$ for every supersymmetric minimum of $V_{\rm cl}(r)$.[80] It follows that, for supersymmetric minima, the vector $\partial_t - \partial_\phi - \partial_\psi$ is everywhere tangent to the world volume of the dual giant, as might have been expected.[81]

# 5 Quantization of dual giants in equal charge black hole backgrounds

In the previous section, we have found the lowest energy classical solutions at any given values of the charges $q_1$ and $q_2$ (equivalently $q_n$ and $q_c$). In this section, we will find the quantum wave function corresponding to these classical solutions.

As the overall actions (67), (79) are both proportional to $N$, it follows that the effective value of $\hbar$ for our problem is of order $\frac{1}{N}$. As a consequence, quantum effects are small in the large $N$ limit. The reader may be excused for (at first) suspecting that the only effect of these quantum effects is to smear perfectly localized classical solution (located at the minimum of the effective potentials plotted in Fig. 7) into a harmonic oscillator wave function, of width $\frac{1}{\sqrt{N}}$.

While the broadening described above does happen, there is a qualitatively more interesting effect, associated with the fact that our particle lives in a spacetime with a horizon. When we quantize particle motion, we demand that the wave function is 'ingoing' at the horizon, i.e. regular in Eddington-Finkelstein coordinates. This boundary conditions allows our wave function to leak into (or out of, see below) the black hole horizon. This leakage is a tunneling effect, and so occurs with an amplitude of order $e^{-Nq_n}$. While this effect is quantitatively tiny, it is qualitatively significant: it gives the energy $\omega$ of our wave function an imaginary part. Interestingly enough, the sign of this imaginary part changes, depending on whether $\mu$ is less than or greater than $\frac{q_n}{q_c}$. In the former case the imaginary part of $\omega$ turns out to be negative, describing a state that decays into the horizon. In the latter case the imaginary part of $\omega$ is positive, describing a state that grows due to a super radiant charge flux out of the horizon. At the critical value of $\mu$, the state neither grows nor decays, but is stationary. These dynamical results are in perfect agreement with the thermodynamical expectations spelt out around (94).

In addition, the quantization of the motion of duals in black hole backgrounds produces representations of $U(3)$, and so produces a detailed understanding of the $SO(6)$ quantum numbers of the states of DDBHs.

In the rest of this section we present a detailed quantization of the classical action (67), and use this quantum description to derive all the results mentioned above.[82]

---

[80]In fact it is not difficult to show that the orbits with $\dot{\phi}_1 = \dot{\phi}_2 = \dot{\phi}_3 = 1$ and with any constant values of $l_i$ are satisfy the BPS bound $E = q_1 + q_2 + q_3$ where $q_i$ are the momenta along $\phi_i$ and are related to $l_i$ by the constraint $l_i = \sqrt{\frac{q_i}{\sum_j q_j}}$.

[81]Every BPS solution has $E - J_1 - J_2 - Q_1 - Q_2 - Q_3 = 0$, and so has zero value of the charge that generates translation in $\partial_t - \partial_\phi - \partial_\psi$. For this reason, we expect 'translations' by this vector to leave supersymmetric solutions invariant.

[82]Recall, however, that (67) itself was obtained by consistently truncating classical motion to configurations that preserve the $SU(2)_L \times U(1)_R$ symmetry, and also setting all fermionic fields to zero. A full quantization of the dual D3-brane would require accounting for these degrees of freedom in detail would be a formidable task. The end result of such an exercise would yield a 'renormalized' effective Schrodinger equation governing the motion of the 'ground state' D3-brane under study. As the effective corrections to (111) will be fractionally subleading in $\frac{1}{N}$ (and so will, for instance, produce a fractional correction to the imaginary part of the energy that is only of order $1/N$) we ignore them here. We return briefly to this point around (140) below.

## 5.1 The effective Klein-Gordon equation

The action governing the motion of the dual giant, (67), is the action for a particle of unit mass and unit charge, propagating in the metric

$$h_{\mu\nu} = m(r)^2 \tilde{g}_{\mu\nu}, \tag{108}$$

and subject to the effective gauge field

$$C_t = (r^4 - R^4). \tag{109}$$

As a consequence of the overall factor of $N$ (that multiplies the action (67)), the effective value of $\hbar$ for our action is $\frac{1}{N}$. A natural quantization of this action is given by the Klein-Gordon equation for a particle of unit mass

$$\left( -\frac{1}{N^2} h^{\mu\nu} D_\mu D_\nu + 1 \right) \phi = 0, \tag{110}$$

where $D_\mu = \nabla_\mu - iN C_\mu$. (110) can, of course, be equivalently written as

$$\left( -h^{\mu\nu} D_\mu D_\nu + N^2 \right) \phi = 0. \tag{111}$$

As our background spacetime has an event horizon, it is conceptually clearest to work in (ingoing) Eddington-Finkelstein coordinates. Recall that the Schwarzschild time $t$ and the Eddington-Finkelstein time $v$ are related by

$$t = v - \int \frac{dr}{W(r)}, \tag{112}$$

where $W(r)$ is the function that appears in the black hole metric (9).

In these coordinates, our metric takes the form

$$\tilde{g}_{\mu\nu} dx^\mu dx^\nu = -W(r)dv^2 + 2drdv + ds^2(\mathbb{CP}^2) + \frac{b^2}{b^2 + A_{\sigma_3}^2} \left( d\Psi + \Theta + (f A_{\sigma_3} - A_t)\left( dv - \frac{dr}{W} \right) \right)^2 \tag{113}$$

(where $A_t$ and $A_{\sigma_3}$ are the coefficients of $dt$ and $\sigma_3$ in (15) and the functions $b(r)$ and $f(r)$ appear in the black hole metric (9), and have the explicit form listed in (16) and (17) respectively). As we have explained above, our particle is subject to an effective gauge field $C_\mu$, which - in the current coordinates - is given by

$$C_v = C_t = r^4 - R^4, \qquad C_r = -\frac{C_t}{W(r)} = \frac{R^4 - r^4}{W(r)}. \tag{114}$$

Recall that, in the original $(t, r)$ coordinates, we had chosen to work in a gauge in which the contraction of both gauge fields $A$ and $C$ with the killing vector vanishes on the horizon.[83] This choice guarantees that $C_r$ (see (114)) and the gauge field $A$ (and so the 10 dimensional metric(113) )[84] are both regular at the horizon in our new Eddington Finklestein coordinates.

We are interested in studying the Klein Gordon equation (111) in the metric $h_{\mu\nu} = m(r)^2 \tilde{g}_{\mu\nu}$ with $g_{\mu\nu}$ given in (113). (111) is a partial differential equations; its solutions depend on the time coordinate $v$, the 'fibre' coordinate $\Psi$ and coordinates on $\mathbb{CP}^2$, in addition to the radial coordinate $r$. However the metric $h_{\mu\nu}$ enjoys invariance under time translations and $U(3)$. As a consequence, we can specialize to solutions to (111) that have definite values

---

[83] In equations, our fields obey $A_t - f A_{\sigma_3} = 0$, and $C_t = 0$.

[84] Recall that $W(r)$ vanishes at the horizon. This introduces a potential singularity (at the horizon) in the last term of (113). This singularity is absent because $A_t - f A_{\sigma_3}$ also vanishes precisely where $W(r)$ vanishes.

of energy, $U(1)$ and $SU(3)$ charges. At any definite given values of these charges, we should expect our equation of motion (111) effectively reduces to an ordinary differential equation in $r$. This is indeed the case, as we now pause now to explain.

Let us first account for the $U(1) \in U(3)$ symmetry of of $h_{\mu\nu}$. In the coordinates used in (113) this $U(1)$ act as a translation on the coordinate $\psi$. Note that our metric $h_{\mu\nu} = m(r)^2 \tilde{g}_{\mu\nu}$ (with $\tilde{g}_{\mu\nu}$ given in (113)) takes the Kaluza Klein form with $d\Psi$ as the fibre direction. The effective Kaluza-Klein gauge field equals[85]

$$\left( -\Theta + (A_t - f A_{\sigma_3}) dv - \frac{A_t - f A_{\sigma_3}}{W} dr \right).$$

Using the usual formulae for Laplacians in Kaluza Klein theory (see the previous footnote) the action of the Laplacian on a mode $\phi$ of the form $\phi = e^{ik\Psi}\chi$[86] takes the form

$$
\begin{aligned}
-e^{-ik\Psi}\nabla^2 \phi = {}& -\frac{1}{m^2}(D_v D_r + D_r D_v)\chi - \frac{1}{m^7} D_r\left(m^5 W(r) D^r \chi\right) \\
& + \frac{k^2}{m^2}\left(\frac{b^2 + A_{\sigma_3}^2}{b^2}\right)\chi - \frac{1}{m^2}\nabla^2_{CP^2,k}\chi,
\end{aligned}
\tag{115}
$$

where $D_r = \partial_r + i\frac{k(A_t - f A_{\sigma_3}) - NC_t}{W}$, $D_v = (\partial_v + i(k(A_t - f A_{\sigma_3}) - iNC_t)$, and $D_\Psi = -ik$ and $\nabla^2_{CP^2,k}$ is the Laplacian on $\mathbb{CP}^2$ with an effective gauge field given by the replacement $D_\mu \to D_\mu + ik\Theta$ (see Appendix H.3).

We have explained above that the differential operator (115) acts on $\mathbb{CP}^2$ coordinates through a covariantized Laplacian, describing a particle of charge $k$ subject to a gauge field equal to $-\Theta$ on on $\mathbb{CP}^2$. This Laplacian is standard, and is easily diagonalized. Recall $k$ is an integer. In Appendix H.3 we recall that a basis for the infinnite dimensional space of 'sections' of this charged particle is given by $\mathbb{CP}^2$ 'monopole spherical harmonics', which can be labeled by two positive integers, $n_Z$ and $n_{\bar{Z}}$, subject to the constraint $n_Z - n_{\bar{Z}} = k$. Moreover we demonstrate that $-\nabla^2_{CP^2,k}$ evaluates to $4(n_Z n_{\bar{Z}} + n_Z + n_{\bar{Z}})$ when acting on section in the $n_Z \, n_{\bar{Z}}$ representation. Consequently, if we set $\chi = e^{-i\omega v}\phi_{n_Z,n_{\bar{Z}}}h(r)$, (where $\phi_{n_Z,n_{\bar{Z}}}$ is any function in the given representation) we see that

$$
\begin{aligned}
m^2 \frac{e^{i\omega v}}{\phi_{n_Z,n_{\bar{Z}}}}(-\nabla^2 + N^2)\phi = {}& -(D_v D_r + D_r D_v)h(r) - \frac{1}{m^5}D_r\left(m^5 W(r) D^r h(r)\right) \\
& + \left(\left(\frac{b^2 + A_{\sigma_3}^2}{b^2}\right)(n_Z - n_{\bar{Z}})^2 + 4(n_Z n_{\bar{Z}} + n_Z + n_{\bar{Z}}) + N^2 m^2\right)h(r),
\end{aligned}
\tag{116}
$$

where, now,

$$D_v = -i\omega + i((n_Z - n_{\bar{Z}})(A_t(r) - f A_{\sigma_3}(r)) - NC_t(r)). \tag{117}$$

The Klein-Gordon equation is the assertion that the LHS - and hence the RHS - of (116) vanishes. We have thus achieved the promised reduction of dynamics to a second order differential equation in $r$.

---

[85]Our conventions are as follows. We say that a metric with line element $ds^2 = H_{ij}dx^i dx^j + e^{2\sigma}(d\psi - A_i dx^i)^2$ is of the Kaluza Klein form with $\psi$ as the fibre direction, and with KK gauge field $A_i$. In such a spacetime, the action of the Laplacian on a field of the form $B(x_i)e^{ik\psi}$ is given by

$$-e^{-ik\psi}\nabla^2\left(B(x_i)e^{ik\psi}\right) = -H^{ij}D_i D_j B + e^{-2\sigma}k^2 B,$$

where $D_i = \nabla_i - ikA_i$ and $\nabla_i$ is the usual general relativistic covariant derivative on the base manifold with metric $H_{ij}$.

[86]As $\Psi$ is a periodic variable with periodicity $2\pi$, $k$ is, of course, an integer. This will be made more explicit below.

We will perform a linear redefinition of $h(r)$, that eliminates 'first derivative' terms in (116), and thereby cast this equation in the Schrodinger form. The relevant field redefinition is given by

$$h(r) = \frac{e^{i\omega r_s}}{\sqrt{W}} g(r), \tag{118}$$

where

$$r_s = \int_R^r \frac{dr}{W(r)}. \tag{119}$$

It is not difficult to verify that (116) turns into the Schrodinger equation for $g(r)$

$$\left( -\frac{1}{2N^2} \frac{\partial^2}{\partial r^2} + V(r) \right) g(r) = 0, \tag{120}$$

where the effective potential $V(r)$ is given by

$$\begin{aligned} V(r) = & \frac{(m^5 W)''}{4N^2(m^5 W)} - \frac{((m^5 W)')^2}{8N^2(m^5 W)^2} - \frac{\left( \frac{\omega}{N} + r^4 - R^4 - (A_t - fA_{\sigma_3})\left( \frac{n_Z - n_{\bar{Z}}}{N} \right) \right)^2}{2W^2} + \frac{m^2}{2W} \\ & + \frac{1}{2W} \left( \left( \frac{b^2 + A_{\sigma_3}^2}{b^2} \right) \left( \frac{n_Z - n_{\bar{Z}}}{N} \right)^2 + 4 \left( \frac{n_Z n_{\bar{Z}} + n_Z + n_{\bar{Z}}}{N^2} \right) \right). \end{aligned} \tag{121}$$

Note that the change of variables (118) is singular at the horizon. As the physical condition on solutions is that $h(r)$ is nonsingular at the horizon, we seek solutions $g(r)$ of (120) that have a compensating singularity to ensure that $h(r)$ is regular at the horizon. In the neighborhood of the horizon, in other words, $g(r)$ is required to behave like

$$g(r) \sim \sqrt{W(r)} e^{-i\omega r_s} \sim (r - R)^{-\frac{i\omega}{W'(R)} + \frac{1}{2}}, \tag{122}$$

times an analytic function. We are thus instructed to solve the Schrodinger equation (120) subject to the boundary condition of normalizability at $r = \infty$, and the boundary condition (122) at $r = R$.

### 5.1.1 Recovering the classical potential from the quantum potential in the large $N$

As we have explained above, the wave function of our dual graviton (at spacetime energy $\omega$ and with $U(3)$ charges parameterized by $n_Z$ and $n_{\bar{Z}}$) is the *zero 'energy'* solution to the Schrodinger equation with the potential $V(r)$ listed in (121). We can thus obtain a sort of quantum version of the classical potential $V_{\text{cl}}(r)$ by imposing the equation $V(r) = 0$[87] and using this equation to solve for $\omega$ (the spacetime energy, i.e. the energy under the spacetime klling vector $\partial_v$) as a function of $r$. We should expect this procedure to yield precisely the classical potential $V_{\text{cl}}(r)$ in the classical limit.[88] This expectation is easily verified. If we assume that $n_Z$ and $n_{\bar{Z}}$ are $\mathcal{O}(N)$ (as is appropriate in the classical limit), we find that the first two terms on the RHS of the first line of (121) are of order $\frac{1}{N^2}$, and so can be neglected in comparison to the other terms in the RHS of (121) (which are of order unity in the large $N$ limit). Setting

---

[87]We impose the equation $V(r) = 0$ because we are looking for a zero 'energy' solution to the Schrodinger equation (120). For the purposes of computing the potential, we set the kinetic contribution in the Schrodinger equation to zero. Note that the 'energy' that appears in the Schrodinger equation' is completely distinct from the spacetime energy - the energy under the killing vector $\partial_v$ - which equals $\omega$. From the viewpoint of first quantization, the Schrodinger 'energy' is dual to a shift of the worldline proper time; the vanishing of this 'energy' is a consequence of diffeomorphism invariance on the worldline.

[88]That is in the limit that all charges are first scaled to classical values: that is when $n_Z$, $n_{\bar{Z}}$ and $\omega$ are taken to be of order $N$) and $N$ is then taken to infinity.

the first two terms on the RHS of the first line of (121) to zero, and solving for $\omega$, we obtain precisely[89] $\omega = V_{\text{cl}}(r) - V_{\text{cl}}(R)$ (where $V_{\text{cl}}(r)$ is listed in (81)) once we identify

$$
\begin{aligned}
\frac{n_Z + n_{\bar{Z}}}{N} &= q_n, \\
\frac{n_Z - n_{\bar{Z}}}{N} &= q_c,
\end{aligned}
\tag{123}
$$

(123) give us the formulae that allow us to translate between the quantum charges of the current section and the classical charges of the previous section.

### 5.1.2 The quantum potential a combined large charge limit, large $r$ limit

The quantum potential $V(r)$ simplifies significantly (without trivializing) when $r$ and all charges are taken to be large in a particular coordinated manner. This simplification happens even at finite $N$, as we now explain.

Upon scaling charges and $r$ in the following coordinated manner[90]

$$
r \to \frac{r}{\epsilon}, \qquad \omega \to \frac{\omega}{\epsilon^2}, \qquad n_Z \to \frac{n_Z}{\epsilon^2}, \qquad n_{\bar{Z}} \to \frac{n_{\bar{Z}}}{\epsilon^2},
\tag{124}
$$

and taking $\epsilon$ to be small, we find that the quantum potential (121) simplifies to[91]

$$
V(r) = \frac{r^2}{2} + \frac{(n_Z + n_{\bar{Z}})^2}{2N^2 r^2} - \frac{\omega}{N} + \mu \frac{(n_{\bar{Z}} - n_Z)}{N}.
\tag{126}
$$

In this limit it is very easy to solve the equation $V(r) = 0$ for $\omega$ as a function of $r$ (see the previous subsection). We find

$$
\omega = N\left( \frac{r^2}{2} + \frac{(n_Z + n_{\bar{Z}})^2}{2N^2 r^2} - \mu \frac{n_Z - n_{\bar{Z}}}{N} \right),
\tag{127}
$$

in perfect agreement with the first term in (91) (the classical potential in the same scaling limit).[92] In this limit, therefore, $\omega$ correctly reproduces $V_{\text{cl}}(r)$ even at finite $N$.

As $V(r)$ is linear in $\omega$ in the limit under study (see (126)), the relationship between the quantum and classical potentials is very simple in this limit, and is given by $V_{\text{cl}}(r) = NV(r) + \omega$. In this regime, therefore, $V(r)$, has the same minima as $V_{\text{cl}}(r)$, and so $V(r)$ has a minimum at $r^2 = \sqrt{q_n}$ (see the first term in (92)) with $q_n$ replaced by $\frac{n_Z + n_{\bar{Z}}}{N}$. In other words the potential (126) has a minimum at

$$
r_*^2 = \frac{n_Z + n_{\bar{Z}}}{N} = q_n.
\tag{128}
$$

The potential at this minima vanishes when $\omega$ equals

$$
\omega_* = (n_Z + n_{\bar{Z}}) - \mu(n_Z - n_{\bar{Z}}) = N(q_n - \mu q_c)
\tag{129}
$$

---

[89]The procedure of solving for $\omega$ as a function of $r$ is slightly ambiguous at finite $N$, as the equation is quadratic, and so has two roots for $\omega$ as a function of $r$. In the classical large $N$ limit, however, only one of these two roots survives (the other goes to infinity), yielding an unambiguous answer.

[90]Physically $\epsilon \sim \frac{1}{q_n^{\frac{1}{2}}}$; working at small $\epsilon$ is equivalent to working at large charge.

[91]If we set $r^2 \sim q_n$ and $\frac{\omega}{N} \sim q_n$ (see (124)) we see that the potential scales like $V \sim q_n$. Note, therefore, that the integral

$$
\int \sqrt{V}\, dr \sim q_n,
\tag{125}
$$

where we have assumed that the integral is performed over a range of order $\sqrt{q_n}$.

[92]Recall that we work in the gauge (15) throughout this section.

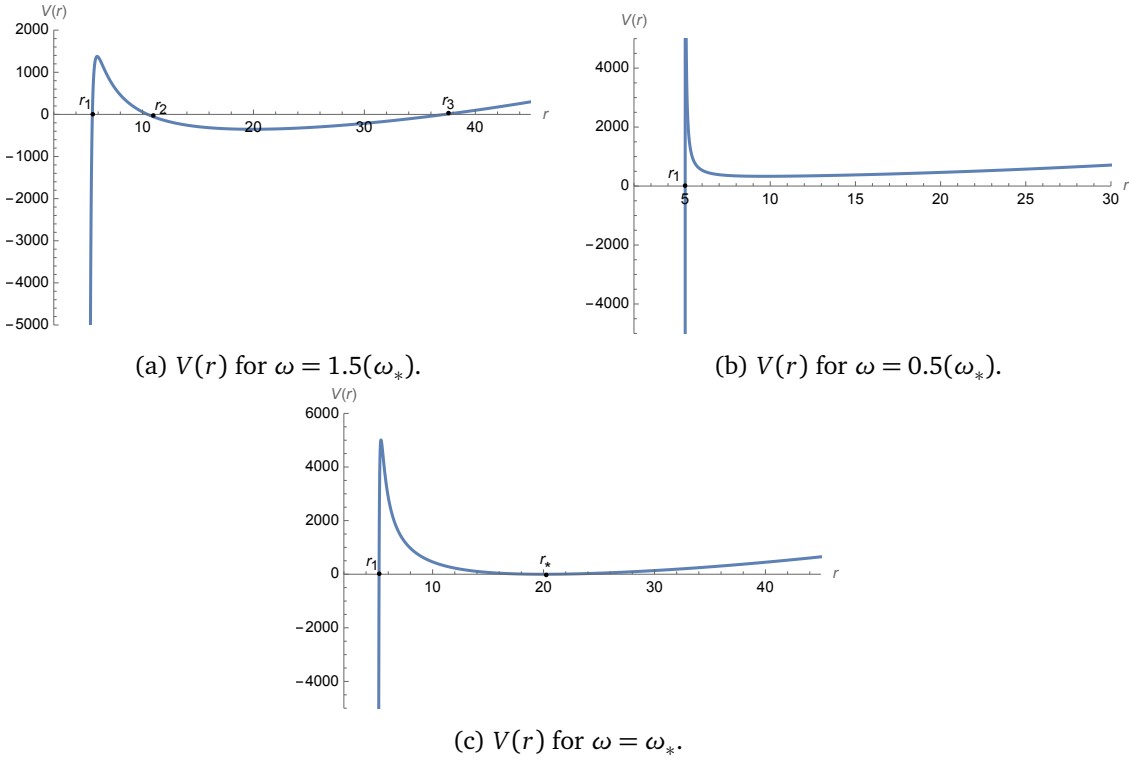

(a) $V(r)$ for $\omega = 1.5(\omega_*)$.        (b) $V(r)$ for $\omega = 0.5(\omega_*)$.

(c) $V(r)$ for $\omega = \omega_*$.

Figure 7: Effective potential for black holes with $\mu = 0.5$.

(this is the analogue (93)). For future use we note that the second derivative of the potential at $r_*$ is given by

$$V''(r_*)\big|_{\omega=\omega_*} = 4\,. \tag{130}$$

In Fig.7 we have plotted the function $V(r)$ (see (120)) at $\mu = \frac{1}{2}$ for three different values of $\omega$, $\omega = \omega_*$, a value of $\omega > \omega_*$ and a value of $\omega < \omega_*$.

### 5.1.3 The potential in the near horizon region

In this paper, we are interested in the large $N$ limit (with charges also scaled to order $N$ or larger). In the previous section we have explained that in this limit, the first two terms on the RHS of (121) can be ignored at generic values of $r$, as these terms are of order $\frac{1}{N^2}$ (in comparison, the remaining terms are of order unity). In this subsubsection we will explain that when $\mu$ is tuned to the neighbourhood of the critical value (defined so that $\omega_*$ in (129) is of order unity rather than order $N$) an exceptional situation arises, and the first two 'quantum' terms in (121) can no longer be neglected when $r$ lies in the neigbbourhood of the horizon $R$. Of course, the neighbourhood of the horizon is a location of particular physical interest, and so warrant study in their own right.

Recall that near the horizon $W(r) \sim W'(R)(r-R)$, so the first two terms on the RHS of (121) - both of which are proportional to $\frac{1}{W^2}$ diverge as $r \to r_H$ like $\frac{1}{(r-R)^2}$. The same is true of the third term on the RHS of (121). Near enough to the horizon, these terms dominate the

quantum potential at every value of $N$; in this region we find[93,94]

$$V(r) = -\frac{(W'(R))^2 + 4\omega^2}{8N^2(W'(R)(r-R))^2}.$$   (132)

In the rest of this subsubsection we study the quantum potential $V(r)$ in the neigbourhood of $r = R$ (and $\omega \sim \omega_*$) first at generic values of $\mu$, and then near the critical value of $\mu$.

**Generic values of $\mu$**

For the purpose of computing the lowest energy state, we will be interested in the quantum potential at $\omega \sim \omega_*$. We see from (129) that at generic values of $\mu$

$$\omega_\star \sim \mathcal{O}(Nq_n). \qquad \text{generic values of } \mu.$$   (133)

It follows that at generic values of $\mu$ the second term in the numerator of (132) is dominant over the first,[95] and the potential in the near horizon region scales like

$$V(r) \sim -\frac{q_n^2}{(r-R)^2}$$

(all scalings with $q_n$ are provided under the assumption that $q_n \gg 1$, and that $q_c$ and $q_n$ are of the same order of largeness). More generally, we find that the potential around the horizon can be expanded as

$$V(r) = q_n^2 \sum_{n=-2}^{\infty} A_n (r-R)^n.$$   (134)

At generic values of $\mu$, all $A_n$ are all of order unity.[96]

We now have a qualitative picture of the potential $V(r)$ at large $q_n$ and generic values of $\mu$. The potential starts out at $-\infty$ at the horizon $R$. Upon increasing $r$, $V(r)$ increases, reaching zero at $r = r_1$ where $r_1 - R \sim O(1)$. The potential continues to increase, reaching its maximum height of $\mathcal{O}(q_n^2)$ at $r - R$ of order unity, and then decreases more slowly, attaining a height

$$V(r_*) \sim q_n - \mu q_c - \frac{\omega}{N},$$

at the local minimum located at

$$r_* \sim \sqrt{q_n}.$$

Subsequently, the potential rises again, reaching $\infty$ at $r = \infty$.

---

[93]The second ter in the numerator of (132) - the term proportional to $\omega^2$- comes from the third term on the RHS of (121).

[94]For non-rotating black holes (132) can be written more explicitly as

$$V(r) = -\frac{\frac{R^2\omega^2}{4(\mu^2 - 2R^2 - 1)^2} + \frac{1}{4}}{2N^2(r-R)^2}.$$   (131)

[95]So - at these values of $\mu$- the first two terms on the RHS of (121) are subdominant compared to the third term in (121) even when $r$ is very near the horizon

[96]We note for future use that, as in (125)

$$\int \sqrt{V} dr \sim q_n,$$   (135)

where we assume that the integral is taken over a range of order unity.

**$\mu$ near the critical value**

The coefficient of the leading term in the expansion (134), $A_{-2}$ is of order $\frac{\max(\frac{\omega}{N},\frac{1}{N})^2}{q_n^2}$), and so is generically of order unity. It is, however, possible to tune $\mu$ in a manner that ensures that $\omega \sim \omega_* \sim \mathcal{O}(1)$ (recall that this quantity is generically of order $N$). In this physically interesting situation $A_{-2}$ becomes anomalously small; it is of order $\frac{1}{N}$ (rather than of order unity, as is the case at generic values of $\mu$).[97]

Focussing on this special situation, let us now examine the structure of the quantum potential $V(r)$ in the neighbourhood of $r = R$. When $r$ is taken to lie sufficiently near to $R$, $V(r)$ continues, of course, to be dominated by the first term - the term proportional to $A_{-2}$- in (134). However, as $A_{-2}$ is of order $\frac{1}{N}$ (while $A_{-1}$ and all subsequent $A_n$ continue to be of order 1), the term proportional to $A_{-1}$ in (134) becomes comparable to the first term in this expansion, already when $r-R$ is of order $\frac{1}{N}$ or larger. On the other hand all subsequent terms in (134) - those proportional to $A_0$, $A_1$ etc - are subdominant (compared to this second term in the expansion) as along as $r-R \ll 1$. It follows that $V(r)$ is well approximated by the first two terms in the series (134), i.e. by[98]

$$V(r) = -\frac{\frac{1}{4}+\frac{\omega_q^{*2}}{W'(R)^2}}{2N^2(r-R)^2} + \frac{b}{N^2(r-R)} \, , \tag{136}$$

all through the region $r-R \ll 1$. The constants in (136) are given by

$$\begin{aligned}
b = \frac{1}{8W'(R)}\Bigg( &4\left(\frac{j^2\mu^2 R^4(n_Z-n_{\bar{Z}})^2}{R^6-j^2(q^2-2pR^2)}+n_Z^2+2n_Z n_{\bar{Z}}+2)+n_{\bar{Z}}^2+4n_{\bar{Z}}\right)+4m^2 N^2 \\
&+ W''(R)+4\frac{W''(R)}{W'^2(R)}\omega^2\Bigg).
\end{aligned} \tag{137}$$

Note that the coefficient of $\frac{1}{r-R}$ - namely $\frac{b}{N^2}$ continues to be of order $\sim \mathcal{O}(q_n^2)$ even when $\omega$ is small, illustrating that $A_{-1}$ (unlike $A_{-2}$) remains of order unity even when $\omega$ is small.

Within the approximation (136), the turning point $r_1$ (see fig. 8) is located at

$$r_1 - R = \frac{\frac{1}{4}+\frac{\omega^2}{W'(R)^2}}{2b} \, . \tag{138}$$

Notice that

$$r_1 - R \sim \mathcal{O}\left(\max\left(\frac{\omega}{N},\frac{1}{N}\right)\right),$$

so $r_1$ approaches very close to the horizon when $\omega$ is small.

## 5.2 Qualitative nature of the solution

As mentioned in the introduction to this section, our goal is to find the quantum state corresponding to the classical configuration in which our dual giant sits at the minimum of the potential $V_{\text{cl}}(r)$, i.e. in a quantum solution at $\omega$ near $\omega_*$. In this brief subsubsection we describe qualitatively how we will proceed (we turn to quantitative analysis in subsequent subsections).

---

[97]In this situation the first term in the numerator of (132) cannot be ignored compared to the second term. As the first term in (132) receives contributions from the 'quantum' terms in (121).

[98]For different values in this range, the term with $n = -2$ is either much larger than, comparable to, or much smaller than the term with $n = -1$. $n = -2$ dominates $n = -1$ for $(r-R) \ll A_{-2}q_n$. The converse is true when the inequality is reversed.

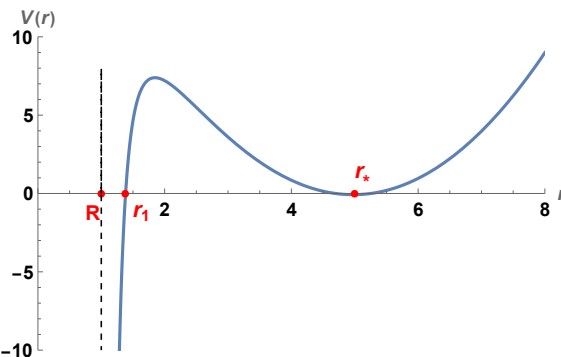

Figure 8: Effective potential for probes for $\omega = \omega_*$. Here $R$ is the radius of horizon, $r_1$ is the first turning point and $r_*$ is the minima of the harmonic oscillator.

We wish to find a zero energy wave function sitting in the potential $V(r)$ with the smallest possible value of $\omega$. As $V(r)$ vanishes at $\omega = \omega_* = N(q_n - \mu q_c)$, we expect to find the lowest energy state (which will be roughly the ground state of the harmonic oscillator centred around $r_*$) at

$$\omega = \omega_* + \dots, \tag{139}$$

where the $\dots$ in (139) represent terms that are subleading in the large $N$ limit.[99] But $\omega_*$ is precisely the energy at which the potential $V(r)$ has a minimum that just 'touches' the real axis at $r = r_*$ (see Fig. 7 c). At the values of $\omega$ that are relevant for the rest of this section, therefore, the potential $V(r)$ takes the qualitative form depicted in the schematic diagram Fig 8. We re emphasize that graph depicted in Fig. 8. has 3 distinguished values of $r$. These are

- $r = R$, the event horizon. $V(r)$ asymptotes to $-\infty$ here.

- $r = r_1$, the point where $V(r)$ first cuts the $x$ axis (i.e. the point at which $V(r)$ first transits from negative to positive).

- $r = r_*$, the point at which $V(r)$ has a local minimum. Notice $V(r_*) \approx 0$

It would not be difficult to systematically improve (139) in a power series expansion in $\frac{1}{Nq_n}$. However every term in this power expansion is real, so no finite order in this expansion captures the effect of physical interest to us, namely the imaginary part of $\omega$ (a consequence of the decay of the wave function into the black hole). This imaginary part is a tunneling effect and occurs at order $e^{-Nq_n}$, and so is nonperturbative from the viewpoint of the $\frac{1}{Nq_n}$ expansion. In order to capture the effect of physical importance, we proceed as follows.

At generic values of $\omega$, a solution of (120) that is normalizable at infinity (to the right of the oscillator minimum) grows rather than decaying to the left of the oscillator minimum.

---

[99]Working precisely within the theory governed by the Klein-Gordon equation (111) that the ground state energy of a harmonic oscillator around $r = r_*$ -in the potential $V(r)$ - carries energy equal to its classical value plus $\frac{1}{N}$. Consequently (111) itself is solved by choosing (see (126))

$$\omega = \omega_* + 1 + \mathcal{O}\left(\frac{1}{q_n N}\right), \tag{140}$$

where the term $\mathcal{O}(\frac{1}{q_n N})$ accounts for the effects of non harmonicity in the potential $V(r)$ around $r_*$. While (140) holds for the equation (111), but is not correct for the physical situation under study. This is because our dual D3-brane also has fermionic degrees of freedom, whose zero point energy cancels the 1 on the RHS of (140). A serious computation of $\omega - \omega_*$ would require one to account for these. In fact it would require us to use the full D3-brane action (the reduction to the particle action is presumably insufficient). As the real part of $\omega - \omega_*$ is not of great physical interest, we leave its (rather involved) computation to the interested reader, noting only that supersymmetry guarantees that this quantity vanishes exactly in the absence of the black hole.

Let us, for a moment, imagine that the potential around $r_*$ was a genuine harmonic oscillator (with no anharmonic corrections). Upon varying $\omega$, in this case, we would find that the wave function decays to the left of this minimum only when one of the eigenstates of the harmonic oscillator obeys (120). This happens when $\omega = \omega_*^q + 2n$ (where $\omega_*^q = \omega_* + 1$; the +1 accounts for the zero point energy shift) for $n = 0, 1, 2 \ldots$ (recall that the angular frequency of our harmonic oscillator equals 2). The lowest energy at which this is achieved is $\omega = \omega_*^q$. Now our actual problem includes anharmonicities, so we would have to correct $\omega$ systematically in a power series in $\frac{1}{Nq_n}$ (these are the power series corrections in (139)). To any order in this expansion we would, however, be able to compute $\omega$ - and in particular, find the (real) lowest energy solution for which - the wave function decays to the left of $r_*$. Let us continue to call this value of $\omega$, $\omega_*^q$. The solution at this energy would (by construction) match onto the WKB solution that decays (when moving away from $r_*$, towards smaller values of $r$).[100] As this solution proceeds further (to smaller values of $r$) it eventually encounters a turning point in $V(r)$ at '$r = r_1$ (see fig.8). Of course, the WKB approximation breaks down in the neighborhood of $r = r_1$, and we have to continue the solution to $r < r_1$ using the well known WKB connection formulae. These formulae, however have the property that they match the purely growing solution to the right of $r_1$[101] to an equal linear combination of incoming and outgoing modes for $r < r_1$.[102] At least generically, however, the WKB approximation holds once again to the left of $r_1$. Continuing this linear sum of ingoing and outgoing solutions to smaller $r$ using WKB, we obtain a linear combination of ingoing and outgoing solutions at the horizon, in violation of the boundary condition (122). Consequently, the solution at energy $\omega = \omega_*^q$ is not physically relevant, because it does not obey physical boundary conditions.

Let us search for a solution that does obey the right boundary conditions. As we have mentioned above, the connection formulae tell us that a purely ingoing solution to the left of $r_1$ (and so at the horizon) matches onto a linear combination of decaying and growing modes (to the right of $r_1$). The coefficients of these growing and decaying modes are equal in magnitude but differ by a relative factor of $i$. We must thus choose $\omega$ to make sure that our wave function has precisely this structure just to the right of $r_1$.

We can, indeed, produce such a wave function by changing $\omega$ away from $\omega_*^q$ by a term proportional to $ie^{-2N\beta}$ where $\beta = \int_{r_1}^{r_*} \sqrt{V} dr$. This maneuver forces the solution to the left $r_*$ to be linear combination of the decaying and growing modes, with relative coefficients[103] proportional to $ie^{-2N\beta}$. By the time this solution reaches $r_1$, the decaying piece has diminished by a factor proportional to $e^{-N\beta}$, while the growing piece has increased by a factor of $e^{N\beta}$, allowing the two modes to have equal coefficients - but differ by a phase factor $i$ -at $r = r_1$.

Note that the imaginary shift of energy discussed above is extremely small; it is of order $e^{-Nq_n}$. The reader may wonder how it is consistent for us to discuss such minute nonperturbative changes in energy without first accounting for the (much larger) perturbative shifts in energy in a power series in $1/(Nq_n)$. The reason this is consistent is that this nonperturbative shift is imaginary (in contrast all perturbative corrections to the energy are real). While the nonperturbative shift in energies that we compute is tiny, it is nonetheless the leading contribution to the imaginary shift of energies.

---

[100]This is precisely the procedure we would adopt when searching for bound states of a potential that continues to grow (on both sides) when we move away from $r_*$.

[101]Note that a solution that decays as $r$ decreases away to the left of $r_*$ is a solution that grows as $r$ increases to the right of $r_1$.

[102]We give a quick argument that this must be the case. The probability flux vanishes when evaluated on a purely decaying (or any other real solution). As the probability current is conserved, it must also vanish for $r < r_1$. This tells us that the ingoing and outgoing pieces must come with equal coefficients for $r < r_1$ so that their probability flux adds to zero

[103]We report the ratio of growing/decaying: measured moving away from $r_*$ to the left, i.e to smaller values of $r$.

In the rest of this section we implement the description of the previous paragraph to find a precise formula for the imaginary part of $\omega$.

## 5.3 Validity of the WKB approximation

In the rest of this section we will attempt to find the 'lowest' $\omega$ solution to the equation (120) in the WKB approximation.[104]

### 5.3.1 Condition for validity

Solutions to the Schrodinger equation (120) are well approximated by the WKB approximation

$$g(r) = \frac{1}{\sqrt{N}(2V(r))^{\frac{1}{4}}} \left( a_1 e^{N \int dr \sqrt{2V(r)}} + a_2 e^{-N \int dr \sqrt{2V(r)}} \right), \tag{141}$$

whenever the fractional change in the local momentum is small over one wavelength, i.e. provided $\frac{d}{kdr} \ln k \ll 1$, (here $k$ is the local wave number equal to $N\sqrt{-2V}$), i.e. provided

$$\frac{V'}{NV^{\frac{3}{2}}} \ll 1. \tag{142}$$

The factor of $\frac{1}{N}$ in (142) tells us that the WKB approximation is always valid except

- In the immediate neighborhood of turning points, i.e. locations where $V(r) = 0$.

- In regions where $V(r)$ happens to be of order $\frac{1}{N^2}$ (for whatever reason) even though $r$ does not lie in the immediate neighborhood of a turning point.

As we have explained above, we are interested in the potential $V(r)$ at $\omega \approx \omega_*$. At such values of $\omega$ we have 3 turning points: the 'harmonic oscillator turning points' in the immediate neighborhood of $r_*$, and the turning point at $r = r_1$.

### 5.3.2 Validity around $r = r_*$

Around $r_*$ the WKB approximation only holds for $|r - r_*| \gg \frac{1}{\sqrt{N}}$. In the immediate neighborhood of $r_*$ (where this condition is not met) we are required to solve the equation exactly, and the match with the WKB solution at $|r - r_*| \gg \frac{1}{\sqrt{N}}$. Fortunately, the required exact solution is easy to find because the potential is quadratic (to great accuracy) in the relevant range of $r$.

### 5.3.3 Validity around $r = r_1$

We study the question of validity of the WKB approximation near $r_1$ first at generic values of $\mu$ (at which $\omega \sim N q_n$) and then separately at anomalously low values of $\omega$.

At generic values of $\mu$, the potential can be linearized around $r = r_1$. It follows from (134) that

$$V(r) \sim q_n^2(r - r_1). \tag{143}$$

Consequently, the figure of merit (142) evaluates to

$$\frac{1}{N q_n (r - r_1)^{\frac{3}{2}}},$$

---

[104]See [33] for the WKB analysis of *Fermionic* fields in a somewhat similar potential in a similar context. [33] find that 'superradiant unstable' Fermions simply populate a Fermi Sea.

and the WKB approximation is good provided

$$|r - r_1| \gg \frac{1}{(Nq_n)^{\frac{2}{3}}} \,. \tag{144}$$

The usual approach to WKB through a turning point is to approximate the potential as linear in the immediate neighborhood of the turning point, and to match this solution to WKB. The linearized approximation around $r_1$ works provided

$$|r - r_1| \ll 1 \tag{145}$$

(see again (134)). It follows that the linearized approximation and WKB approximation have a common domain of validity (so that standard connection formulae can be applied) provided

$$\frac{1}{(Nq_n)^{\frac{2}{3}}} \ll 1 \,. \tag{146}$$

Since $N$ (and perhaps also $q_n$) is large, this condition is always obeyed at generic $\mu$

Let us now turn to anomalously small values of $\omega$. Roughly (see (136) )

$$V = \frac{\alpha^2}{(r-R)^2} + \frac{q_n^2}{(r-R)} \,,$$

where

$$\alpha^2 = \max\left( \frac{\omega^2}{N^2}, \frac{1}{N^2} \right) \,. \tag{147}$$

Note $\alpha^2 \ll 1$. The turning point occurs at

$$r_1 - R \sim \frac{\alpha^2}{q_n^2} \,,$$

and $V'$ at this point $\sim \frac{q_n^6}{\alpha^4}$, and $V''$ around this point $\sim \frac{q_n^8}{\alpha^6}$. The linearized approximation to the potential around the turning point is thus valid provided

$$|r_1 - r| \ll r_1 - R \sim \frac{\alpha^2}{q_n^2} \,. \tag{148}$$

Within the domain of this linearization,

$$V(r) \sim \frac{q_n^6}{\alpha^4}(r - r_1) \,. \tag{149}$$

The WKB figure of merit, (142), is small provided that

$$|r - r_1| \gg \left( \frac{\alpha^2}{Nq_n^3} \right)^{\frac{2}{3}} \,, \tag{150}$$

(150) and (148) can only be simultaneously valid provided

$$\alpha \gg \frac{1}{N} \,. \tag{151}$$

It follows from (147) that (151) is satisfied if and only if

$$|\omega| \gg 1 \,. \tag{152}$$

When (152) is not met, we cannot use the usual WKB method around small values of $r_1$.[105]

Recall that $\omega \sim \omega_*$ and that

$$\omega_* = N(q_n - \mu q_c) = -N q_c \left( \mu - \frac{\mu q_n}{q_c} \right). \tag{153}$$

We thus see that the condition $|\omega_*| \gg 1$ is met provided

$$|\mu - \frac{q_n}{q_c}| \gg \frac{1}{N q_c}. \tag{154}$$

When (152) fails, we can no longer use the WKB approximation for $r < r_1$ (and, in particular, in the neighborhood of the horizon). In this case other methods must be employed for $r$ in the neighborhood of $r_1$ and smaller.

Note that the values of $\mu$ that do not obey (154) - i.e. those $\mu$ for which the WKB approximation fails in the near horizon region - are of special interest, as they lie on the cusp of stability.

## 5.4 Solving around $r = r_*$

Motivated by the discussion above, we set

$$\omega = \omega_*^q + \epsilon, \tag{155}$$

where

$$\omega_*^q = \omega_* + 1 + \ldots \tag{156}$$

(see (140)). The $\ldots$ account for non harmonicities, and so can be ignored in what follows.[106]

(We will find, below, that $\epsilon$ is imaginary and of order $e^{-2\beta}$: we are uninterested in real power law corrections to $\epsilon$ but focus on the nonperturbative imaginary corrections). As we have explained above, the potential $V(r)$ is of the harmonic oscillator form in the neighborhood of $r_*$, and the Schrodinger equation at large $r$ takes the following form

$$\left( -\frac{1}{2N^2} \frac{\partial^2}{\partial r^2} + \frac{V''(r_*)}{2} (r - r_*)^2 \right) g(r) = \left( \frac{\sqrt{V''(r_*)}}{2N} + \epsilon \right) g(r). \tag{157}$$

Using $r_* = \frac{\sqrt{n_Z + n_{\bar{Z}}}}{N}$ and $V''(r_*) = 4$ (see (130)), (157) simplifies to[107]

$$\left( -\frac{1}{2N^2} \frac{\partial^2}{\partial r^2} + 2(r - r_*)^2 \right) g(r) = \left( \frac{1}{N} + \epsilon \right) g(r). \tag{158}$$

The most general solution to (157) takes the form

$$g(r) = c_1 D_{\frac{N\epsilon}{2}} \left( 2\sqrt{N}(r - r_*) \right) + c_2 D_{-\frac{N\epsilon}{2} - 1} \left( 2i\sqrt{N}(r - r_*) \right), \tag{159}$$

where $D_\nu(x)$ is the parabolic D-function. We are interested only in normalizable solutions, i.e. solutions that decay as $r \to \infty$. Now when $r - r_* \gg \frac{1}{\sqrt{N}}$, it follows from standard

---

[105]It is interesting that we recover the same result also by focusing on the extreme near horizon region. plugging (132) into (142), (and recalling that $W'(R)$ is of order unity), we find that (142) holds only if $\omega \gg 1$.

[106]Note that the $\omega$ that appears in (155) is the $\omega$ that appears in the solution to the wave equation (111). As we have emphasized in the discussion around (140), this is not the full energy of the solution: indeed at this order we expect that (in the full energy) the shift by 1 in (156) is exactly cancelled by the energy of the fermionic harmonic oscillator.

[107]As we are uninterested in real corrections to $\omega$, and only the leading term in the imaginary correction to $\omega$, it is sufficient to work at quadratic order.

formulae (for the large argument behavior of parabolic cylindrical functions) $g(r)$ in (159) is well approximated by

$$g(r) = c_1 e^{-N(r-r_*)^2} - i c_2 \frac{e^{N(r-r_*)^2}}{2\sqrt{N}(r-r_*)}. \tag{160}$$

Consequently, the requirement of normalizability forces us to set $c_2$ to zero, and $g(r)$ in (160) reduces to

$$g(r) = c_1 D_{\frac{N\epsilon}{2}}\left(2\sqrt{N}(r-r_*)\right). \tag{161}$$

In the opposite limit $r_* - r \gg \frac{1}{\sqrt{N}}$, $g(r)$ in (161) is well approximated by

$$
\begin{aligned}
g(r) &= c_1 \left( \frac{\sqrt{\pi} e^{N(r-r_*)^2}}{\sqrt{2N}(r_*-r)\Gamma\left(-\frac{1}{2}(N\epsilon)\right)} + e^{-N(r-r_*)^2} \right) \\
&= c_1 \left( \frac{-\epsilon\sqrt{N\pi}}{2\sqrt{2}(r_*-r)} e^{N(r-r_*)^2} + e^{-N(r-r_*)^2} \right),
\end{aligned}
\tag{162}
$$

where in the second step we used the fact that $\epsilon$ is small. We see that our $g(r)$ has a growing piece with coefficient $\epsilon$, in addition to the dominant decaying piece with coefficient unity. Matching (162) with the WKB wave function (141),[108] we conclude that everywhere in the region (144) together with $r - r_* \ll \sqrt{N}$, our solution is well approximated by the WKB wave function

$$g(r) = \frac{1}{(2V(r))^{1/4}} \left( A\exp\left(-\int_r^{r^*} dr' \sqrt{2V(r')}\right) + B\exp\left(\int_r^{r^*} dr' \sqrt{2V(r')}\right) \right), \tag{163}$$

with coefficients

$$A = \frac{i}{\sqrt[4]{e}}, \qquad B = -i\sqrt{\frac{\pi}{2}}\sqrt[4]{e}N^{3/2}\epsilon. \tag{164}$$

The WKB solution $g(r)$ given in (163) can be re-written as follows

$$g(r) = \frac{1}{(2V(r))^{1/4}} \left( A\exp(\int_{r_1}^r N\sqrt{2V})e^{-\beta} + B\exp(-\int_{r_1}^r N\sqrt{2V})e^{\beta} \right), \tag{165}$$

where

$$\beta = \int_{r_1}^{r^*} N\sqrt{2V}. \tag{166}$$

## 5.5 Matching across the turning point at $r_1$

We have seen above that the usual WKB methods (including use of the usual connection formulae) work around the turning point $r_1$ provided that (154) (equivalently (152)) holds. Assuming this is the case, the usual matching formulae (presented in standard quantum mechanical texts) allow us to match the WKB solution (163) with another WKB solution valid in the range (150). We find (see Appendix I for a derivation) that (163) matches (in the range (150)) to

$$g(r) \sim \frac{Ae^{-\beta}}{(2V)^{-\frac{1}{4}}} \left( \sin(\int_{r_1}^r N\sqrt{-2V} + \frac{\pi}{4}) + \frac{2B}{A}e^{2\beta}\cos(\int_{r_1}^r N\sqrt{-2V} + \frac{\pi}{4}) \right). \tag{167}$$

---

[108]In order to perform this match, we use the fact that the potential in the relevant region evaluates to $V(r) = 2(r-r_*)^2 - \frac{1}{N} - \epsilon$.

In the extreme near horizon limit, $V(r)$ simplifies to

$$V(r) = -\frac{(\omega_*^q)^2}{2N^2 W'(R)^2 (r-R)^2}\,,\tag{168}$$

yielding

$$\sqrt{-2V} = \frac{|\omega_*^q|}{N\sqrt{W'(R)}(r-R)}\,.\tag{169}$$

$g(r)$ can thus be explicitly evaluated in the near horizon limit: we find

$$g(r) = (-1)^{1/4} C(r-R)^{1/2}\left(\left(-i + \frac{2B}{A}e^{2\beta}\right)e^{i\frac{|\omega_q^*|}{\sqrt{W'(R)}}\log(r-R)} + \left(\frac{1}{2} - i\frac{B}{A}e^{2\beta}\right)e^{-i\frac{|\omega_q^*|}{\sqrt{W'(R)}}\log(r-R)}\right)$$

$$= (-1)^{1/4} C(r-R)^{1/2}\left(\left(-i + \frac{2B}{A}e^{2\beta}\right)e^{i|\omega_q^*|r_s} + \left(\frac{1}{2} - i\frac{B}{A}e^{2\beta}\right)e^{-i|\omega_q^*|r_s}\right),\tag{170}$$

where

$$C = Ae^{-\beta}\left(\frac{|\omega_*^q|}{N\sqrt{W'(R)}}\right)^{-\frac{1}{2}}.\tag{171}$$

We must now implement the boundary conditions (122). Because of the modulus in (169), it is convenient to consider the two cases $\omega_*^q > 0$ and $\omega_*^q < 0$ separately.[109]

When or $\omega_q^* > 0$, the above solution matches the regular solution at the horizon when

$$\frac{2B}{A}e^{2\beta} = i\,,$$
$$\epsilon = -\frac{i}{N^{3/2}\sqrt{2e\pi}}\exp(-2\beta)\,,\tag{172}$$

so that our final solution for $\omega$ is

$$\omega = \omega_*^q - \frac{i}{N^{3/2}\sqrt{2e\pi}}\exp(-2\beta)\,.\tag{173}$$

When $\omega_q^* < 0$, we need to set the second term in (170) to zero to obtain a regular solution at the horizon. This happens when

$$\frac{2B}{A}e^{2\beta} = -i\,,$$
$$\epsilon = \frac{i}{N^{3/2}\sqrt{2e\pi}}\exp(-2\beta)\,.\tag{174}$$

We see from (172) and (174) that the imaginary part of $\omega_*^q$ flips signs $\mu$ increases through the critical value.

## 5.6 Solution at small $\omega_*^q$

When $\omega_*^q$ is of order unity, the interval $(R, r_1)$ is extremely small. For these values of $\omega_*^q$, (142) is never obeyed - and so the WKB approximation is never valid - within the relevant range.[110] While the results of §5.4 continue to hold at such values of $\omega_*^q$, the analysis of §5.5 is no longer valid.

---

[109]Recall that the procedure we are implementing in this subsection fails for $\omega_*^q$ of order unity: the interpolation between these two cases requires further analysis, see the next subsection.

[110]However the WKB approximation continues to be valid for $r - r_1 \gg \frac{1}{N^{\frac{2}{3}}}$.

If $\omega$ is of order unity, however, then it is certainly true that

$$\frac{\omega}{N} \ll q_n, \tag{175}$$

and when this is true we can use a second approximation: $V(r)$ is well approximated by (136) for $r - R \ll 1$ (and in particular in the neighborhood of $r_1$). In this subsection we will use the approximation (136) to re-solve for the $\epsilon$ (the imaginary correction to the energy), but this time for any $\omega$ that obeys (175).

Notice that (175) and the condition for the validity of the WKB approximation (152) have a large overlap region. It follows, therefore, that the results of this subsection should reduce to those of the previous subsection in the range

$$1 \ll \omega \ll N q_n. \tag{176}$$

We will use this requirement as a consistency check on the computations of the current subsection.

The Schrodinger equation (120) with the potential (136) is easy to solve. Performing the following transformation:

$$g(r) = 2\sqrt{2b(r-R)}f(y), \tag{177}$$

with

$$y = 2\sqrt{2b(r-R)}, \tag{178}$$

yields the following differential equation in the variable $y$:

$$y^2 f''(y) + y f'(y) + \left(\frac{4(\omega_*^q)^2}{W'(R)^2} - y^2\right)f(y) = 0. \tag{179}$$

The solutions of these differential equations are Modified Bessel functions ($I_\mu(y)$ and $K_\mu(y)$) but with imaginary values: specifically with $\mu = i\nu = i\frac{2\omega}{W'}$. The most general solution to this equation takes the form

$$f = c_1 \tilde{I}_\nu(y) + c_2 \tilde{K}_\nu(y), \tag{180}$$

where [34]

$$\tilde{I}_\nu(y) = \mathrm{Re}(I_{i\nu}(y)), \qquad \tilde{K}_\nu(y) = K_{i\nu}(y), \tag{181}$$

and

$$\nu = \frac{2\omega_*^q}{W'}. \tag{182}$$

At small values of $y$ (i,e near the horizon) the solution (180) is well approximated by [34]

$$f \overset{y\to 0}{\sim} c_1 \left(\frac{\sinh(\pi\nu)}{\pi\nu}\right)^{\frac{1}{2}} \cos\left(\nu\log(y/2) - \gamma_\nu\right) - c_2 \left(\frac{\pi}{\nu\sinh(\pi\nu)}\right)^{\frac{1}{2}} \sin\left(\nu\log(y/2) - \gamma_\nu\right). \tag{183}$$

In order that $g(r)$ meet the boundary condition (122)) we are forced to choose[111]

$$\frac{c_2}{c_1} = \frac{i\sinh\pi\nu}{\pi}. \tag{184}$$

---

[111]Once we make this choice, $f(y) = c_1 e^{-i\nu\log\frac{y}{2}}$ and hence $g(r) = 2c_1\sqrt{2b(r-R)}e^{-\frac{i\omega_*^q}{W'(r)}\log b(r-R)}$.

When $r - R \gg \frac{1}{N}$, $y \gg 1$ (see (178)). It follows from the large $y$ asymptotic expansions

$$\tilde{I}_\nu(y) \approx \frac{1}{\sqrt{2\pi y}} e^y \, ,$$
$$\tilde{K}_\nu(y) \approx \sqrt{\frac{\pi}{2y}} e^{-y} \, , \tag{185}$$

that our solution ((180) with (184)), at large $y$, is well approximated by

$$f \overset{y \to \infty}{\sim} \frac{c_1}{\sqrt{2\pi y}} \left( e^y + i \sinh(\pi \nu) \, e^{-y} \right) . \tag{186}$$

It follows (using (177)) that for $r - R \gg \frac{1}{N}$, the function $g(r)$ is well approximated by

$$g(r) = \frac{c_1 \sqrt{2} b^{1/4} (r-R)^{1/4}}{\sqrt{2\pi}} \left( e^{2\sqrt{2b(r-R)}} + i \sinh\left( \frac{2\pi \omega_*^q}{W'(R)} \right) e^{-2\sqrt{2b(r-R)}} \right) . \tag{187}$$

We must now match (187) with the WKB expression (165). Plugging (136) into (165) we find, we find that the WKB wave function becomes

$$g(r) = \frac{\sqrt{N} \sqrt[4]{(r-R)}}{\sqrt[4]{2} \sqrt[4]{b}} \left( A \exp(-\beta) e^{2\sqrt{2b(r-R)} - \alpha} + B \exp(\beta) e^{\alpha - 2\sqrt{2b(r-R)}} \right) , \tag{188}$$

where

$$\alpha = \pi \sqrt{\frac{1}{4} + \frac{(\omega_*^q)^2}{W'(R)^2}} \, . \tag{189}$$

Comparing (188) and (187), we get

$$\frac{B}{A} e^{2\beta} e^{2\alpha} = i \sinh\left( \frac{2\pi \omega_*^q}{W'(R)} \right) . \tag{190}$$

(190) gives us the ratio of coefficients in the WKB wave function. We can use this to determine the imaginary part of $\omega$. Comparing (190) with (164), we find

$$\epsilon = -\frac{1}{N^{3/2}} \sqrt{\frac{2}{e\pi}} \left( i \sinh\left( \frac{2\pi \omega_*^q}{W'(R)} \right) \right) e^{-2\beta} e^{-2\alpha} \, . \tag{191}$$

(191) is our final result for the imaginary part of $\omega$.

As we have explained around (176), when $|\omega_*^q| \gg 1$, the WKB analysis of (5.5) also applies, so we should expect (191) to reduce to (174). This is indeed the case. When $\omega \gg 1$, $\alpha \to \frac{\pi |\omega_*^q|}{W'(r)}$ and so (191) reduces to

$$\epsilon = -\frac{i}{N^{3/2} \sqrt{2e\pi}} e^{-2\beta} \, , \tag{192}$$

in agreement with (174) as expected.

On the other hand, for $\omega \ll -1$,

$$\epsilon = \frac{i}{N^{3/2} \sqrt{2e\pi}} e^{-2\beta} \, , \tag{193}$$

again in agreement with (174) as expected.

## 5.7 Intuitive explanation for the flip in the sign of $\text{Im}(\omega)$ as $\mu$ exceeds $\frac{q_n}{q_c}$

In this subsection, we will give an intuitive explanation for the fact that $\epsilon$, the imaginary part of $\omega$, flips sign as $\omega_*^q$ changes from positive to negative.

Recall that any charged scalar field governed by a charged Klein-Gordon equation has a conserved 'charge' or 'probability' current defined by

$$J_\mu = -i\left(\phi^* D_\mu \phi - \phi D_\mu \phi^*\right). \tag{194}$$

On solutions studied in this paper, it is easy to convince oneself that the nontrivial part of the current conservation equation is simply $\partial_r J^r + \partial_v J^v = 0$. Working out the explicit expressions for these currents we find that the currents

$$\sqrt{g} J^v = 2\left(\frac{(A_t - f A_{\sigma_3})(n_Z - n_{\bar{Z}}) + r^4 - R^4}{W}\right)|h|^2 e^{2\text{Im}(\omega)v} - i(h^* \partial_r h - h \partial_r h^*)e^{2\text{Im}(\omega)v},$$
$$\sqrt{g} J^r = -2(\text{Re}(\omega))|h|^2 e^{2\text{Im}(\omega)v} - iW(h^* \partial_r h - h \partial_r h^*)e^{2\text{Im}(\omega)v}, \tag{195}$$

are conserved. Recall that $h(r)$ is related to the field $g(r)$ (which obeys the Schrodinger equation) by (118). Therefore

$$h^* \partial_r h - h \partial_r h^* = (g^* \partial_r g - g \partial_r g^*)e^{-2\text{Im}(\omega)r_s} + \frac{2i\text{Re}(\omega)}{W}|g|^2 e^{-2\text{Im}(\omega)r_s}. \tag{196}$$

It is easy to convince oneself that the expressions for the currents can be rewritten in terms of $g(r)$ as

$$\sqrt{g} J^v = \frac{1}{W}\left(\left(\frac{2(A_t - f A_{\sigma_3})}{W}\right)|g|^2 - i(g^* \partial_r g - g \partial_r g^*)\right)e^{2\text{Im}(\omega)v}e^{-2\text{Im}(\omega)r_s},$$
$$\sqrt{g} J^r = -i(g^* \partial_r g - g \partial_r g^*)e^{2\text{Im}(\omega)v}e^{-2\text{Im}(\omega)r_s}. \tag{197}$$

The fact that the currents (197) are conserved also follows directly from an analysis of the Schrodinger equation (120).[112]

Let us now understand how current conservation works on the solutions constructed earlier in this section. Integrating the current conservation equation we find

$$J^r(R) = \int_r^\infty \partial_v J^v. \tag{198}$$

The $v$ variation of all fields, in the solutions of interest, is $e^{-i\omega v}$. Let $\omega = \omega_*^q + i\omega_i$ where $\omega_i = -i\epsilon$ with $\epsilon$ given in (191). It follows that all bare fields behave like $e^{-i\omega_*^q v}e^{\omega_i v}$. Consequently, bilinears built out of $\phi$ and $\phi^*$ - and so $J^v$ and $J^r$ - are proportional to $e^{2\omega_i v}$. It follows that (198) can be rewritten as

$$J^r(R) = 2\omega_i \int_r^\infty J^v. \tag{199}$$

Now, on our solutions, $J^v$ is highly peaked (over length scale $\frac{1}{\alpha}$) at $r = r_*$. Let us suppose that we normalize our solution so that $\int_r^\infty J^v = 1$ at $v = 0$. It follows from (198) that

$$J^r(R) = \begin{cases} 0, & r - r_1 \gg \frac{1}{\alpha}, \\ 2\omega_i, & r - r_1 \ll \frac{1}{\alpha}. \end{cases} \tag{200}$$

---

[112]This is simplest to see in the case when $\omega$ is real. In this case $J^v$ is independent of time, so $\partial_v J^v$ vanishes trivially. Conservation of current then amounts to the fact that $\partial_r J^r$ vanishes. But the expression for $J^r$ is simply the usual expression for the probability current in quantum mechanics, and the conservation of this current in eigenstates is a familiar result from non relativistic quantum mechanics.

(note that $\omega_i$ is negative for a solution that decays into the black hole). In particular, $J^r$ is positive and constant as long as we are much to the left of the harmonic oscillator.

As $J^r$ is constant all the way from the horizon to $r_*$, we can compute $J^r$ from our solution near the horizon. Let us suppose first that $|\omega| \gg 1$ so that the WKB approximation is valid near the horizon. At the horizon $W = 0$, and $J^r$ is given by the first term on the RHS of (195). Plugging the near horizon form of $h(r)$ we find using (195)

$$\sqrt{g}J^r(R) = -8N^2 \text{sgn}(\omega_*^q)|A|^2 e^{-2\beta} W'(R) e^{2\omega_i v},$$ (201)

where $A$ is given in (164). The key point here is that $J^r$ is proportional to $\text{sgn}(\omega_*^q)$. This came about as follows. The first term on the RHS of (195) is proportional to $\omega_*^q$. However the WKB formula also involves a prefactor $\propto \frac{1}{\sqrt{V}}$. In the near horizon region (and when $\omega \gg 1$) this prefactor was $\propto \frac{1}{\sqrt{\omega^2}}$ (because the potential is $\propto \frac{\omega^2}{(r-R)^2}$). This prefactor cancels the explicit factor of $\omega$ in the numerator - up to sign.[113]

The fact that LHS of (200) is proportional to $\text{sgn}(\omega_*^q)$ explains the non analytic flip in the value of $\text{Im}(\omega)$ as a function of $\omega_*^q$ (and hence of $\mu$) within the WKB approximation (and hence for $\omega_*^q \gg 1$).

Let us now suppose that $\omega_*^q$ is of order unity. In this case the current is still given by the first term on the RHS of (195), but $|h|^2$ should be computed using (183) (as the WKB approximation fails for such values of $\omega_*^q$). We find that $J^r$ at the horizon is now given by an expression proportional to

$$\omega_*^q \left( \frac{\sinh\left(\frac{2\pi|\omega_*^q|}{W'(R)}\right)}{\frac{2|\omega_q^*|}{W'(R)}} \right) \approx \pi \frac{\omega_q^*}{W'(R)}$$ (202)

(where the approximation in (202) is valid when $\omega_q^* \ll 1$). This explains how the apparently non analytic behaviour of $\text{Im}(\omega)$ (around unity) is smoothened out when we examine its variation over over scales (in the variable $\mu$) of order $\frac{1}{Nq_c}$.

Let us end this subsection by reiterating its punchline. (153) tells us that $\omega_*$ switches sign as $\mu$ increases past its critical value, and vanishes at the critical value. This result follows from the analysis of probes with minima at large values of the radius (we restrict our discussion to such probes in this paragraph) and so is universal and robust; it applies to all black holes (including hairy black holes) independent of details. In this section we have demonstrated that the 'charge current flux' at the horizon determines the sign of $\text{Im}(\omega)$, and, moreover, that the sign of this flux is, itself determined by the sign of $\omega_*$. Putting these facts together, we conclude that it follows on general grounds (and for all black holes including hairy black holes) that $\text{Im}(\omega)$ changes sign (from stable to unstable) as $\mu$ increases past its critical value.

This discussion of this subsection also allows quick qualitative generalizations of the detailed computations presented above. Through this section we have focussed on the analysis of duals whose charges, $q_n$ and $q_c$ are both much larger than unity. Let us try to understand the generalization of these results to finite values of $q_c$. At any value of $q_c$ we would search for the lowest (real part of) $\omega$ solution to (120). Such a solution would locally approximate the ground state of the harmonic oscillator around the local minimum of the quantum potential $V(r)$ at that value of $q_c$, and would yield an approximate solution to (120) provided that $\omega$ is chosen so that $V(r)$ vanish at this local minimum, $r = r_{\min}$. As we have explained above, however, up to corrections of order $1/N$, this happens when $\omega$ equals $V_{\text{cl}}(r_{\min}) - V_{\text{cl}}(R)$. Now suppose that $\mu = 1 + \delta$ where $\delta$ is positive. At least for a range of $\delta$ around zero, we can always find a value of $q_c$ for which $V_{\text{cl}}(r_{\min}) - V_{\text{cl}}(R)$ vanish. This condition defines a critical

---

[113]The fact that $J^r$ is a finite number at the horizon can also be seen from (197).

charge

$$q_c^{\text{crit}}(\delta).\tag{203}$$

The importance of $q_c^{\text{crit}}(\delta)$ is the following. For $q < q_c^{\text{crit}}(\delta)$ our approximate solution to the equation (120) will have positive values of (the real part of) $\omega$, so the analysis of this subsection tells us that $\text{Im}(\omega)$ will be negative, and our solution will be unstable to decaying into the black hole. For $q > q_c^{\text{crit}}(\delta)$, on the other hand, our approximate solution to the equation (120) will have negative values of (the real part of) $\omega$, so $\text{Im}(\omega)$ will be positive, and the black hole will suffer from a super radiant instability.

We conclude, in other words, that dual giant solutions (at any value of the charge) that live at a local minimum of $V_{\text{cl}}(r)$ decay (by quantum tunneling) into the black hole when $V_{\text{cl}}(r_{\min}) - V_{\text{cl}}(R)$ is positive, but suffer from a super radiant instability when $V_{\text{cl}}(r_{\min}) - V_{\text{cl}}(R)$ is negative. These solutions are stable quantum mechanically only when $V_{\text{cl}}(r_{\min}) - V_{\text{cl}}(R)$ vanishes. For a black hole with $\mu = 1 + \delta$ this happens for duals with $q_c = q_c(\delta)$. $q_c(\delta)$ goes to infinity as $\delta \to 0$, and does not exist for negative values of $\delta$.

## 5.8 Summary and discussion our final result

In this section, we set out to answer the following question: at the quantum level, what is the value of $\omega$ (energy) for the lowest energy state of our dual giant graviton outside the horizon, as a function of black hole parameters and the two dual giant charges $q_n = \frac{n_Z + n_{\bar{Z}}}{N}$ and $q_c = \frac{n_Z - n_{\bar{Z}}}{N}$.

Our final answer to this question, derived in this section is

$$\omega = \omega_*^q - \frac{i}{N^{3/2}}\sqrt{\frac{2}{e\pi}}\left(\sinh\left(\frac{2\pi\omega_*^q}{W'(r)}\right)\right)e^{-2\beta}e^{-2\alpha},\tag{204}$$

where $\beta$ is listed in (166) and $\alpha$ is given in (189).

Of course, the interesting aspect of (204) is that $\omega$ has an imaginary piece. Let us first note that $\beta$ is of order $Nq_n$, so the imaginary term is of order $e^{-Nq_n}$ and so is very small. In fact it is of order $e^{-N^2}$ when $q_n$ is of order $N$. This small imaginary piece changes sign as $\omega_*$ flips sign, i.e. as $\mu$ moves from less than $\frac{q_n}{q_c}$ to greater than $\frac{q_n}{q_c}$, $\omega_*$ changes from positive to negative, so the imaginary part of $\omega_*$ changes from negative (decaying) to positive (growing), exactly as thermodynamic arguments would lead us to suspect.

## 5.9 The SO(6) representations of DDBHs

In this section, we have explained that the quantization of dual giants in black hole backgrounds produces several representations of $U(3)$. These representations are labeled by two integers, $(n_Z, n_{\bar{Z}})$. We have demonstrated that the lowest energy solutions in the representations $(n_Z, n_{\bar{Z}})$ suffer from a superradiant instability (and so are dynamically produced) in black holes with $|\mu| > \frac{n_Z + n_{\bar{Z}}}{n_Z - n_{\bar{Z}}}$. Modes with $n_{\bar{Z}} = 0$ are the first to go dynamically (as well as thermodynamically) unstable and so should dominate the final equilibrium ensemble. As $U(3)$ is a symmetry of the black hole background, all modes in the same symmetry multiplet are equivalent. As a consequence, we have produced a whole multiplet of equilibrium states.

However, the story does not end here. Recall that the dual giants in our final equilibrium configuration carry an order one fraction of the energy and charge of the black hole. For this reason, it is unnatural to quantize the ($U(3)$ group space) motion of dual giants, but to treat the $SO(6)$ group directions of the central black hole fixed to given classical values: we should really quantize both these motions simultaneously. Such a quantization is easily achieved in a manner we now describe.

Recall that black hole we study has $SO(6)$ Cartan charges $(Q,Q,Q)$. $SO(6)$ rotations of this black hole produce co-adjoint orbits of this charge, and the quantization of these orbits produces the $SO(6)$ representation with highest weights $(Q,Q,Q)$. Thus, in the absence of the dressing dual giant, the black hole (and all its $SO(6)$ orbits) actually represent a whole representation of states.[114]

We now want to quantize the manifold of black holes and the dual giants simultaneously. In order to do this we reword the quantization of dual giants in terms of co adjoint orbits. The set of dual giant configurations that carry lowest energy at any fixed (overall) $U(1)$ charge of $U(3)$ are parameterized by the $SO(6)$ highest weight vector $(n_Z, 0, 0)$[115] and all its $U(3)$ rotations (see Appendix H.1 for the embedding of $U(3)$ in $SO(6)$), and the quantization of this orbit yields the $U(3)$ representation $(n_Z, 0)$.

Now let us consider the coadjoint orbits of the vector given by the sum of highest weight vectors[116] $(Q,Q,Q)$ and $(n_Z, 0, 0)$, i.e. of $(Q+n_Z, Q, Q)$. The $U(3)$ subgroup of $SO(6)$ leaves $(Q,Q,Q)$ invariant. Consequently, the coadjoint orbit of this $U(3)$ acts only on $n_Z$, producing the manifold of classical D3-brane solutions. Subsequently, the coadjoint orbits of $SO(6)/U(3)$ simultaneously rotate both the black hole and the probe D3-brane (exactly as we want). We thus see that the coadjoint orbit of $(Q+n_Z, Q, Q)$ consists of precisely the physical system of interest. Consequently, the DDBH of interest lies in the $(Q+n_Z, Q, Q)$ representation of $SO(6)$.

The more general rule is the following. Consider a phase space of D3-brane solutions, whose quantization in the background of the $(Q,Q,Q)$ black hole produces a representation of $U(3)$ with highest weights $(n_1, n_2, n_3)$ (these are $U(3)$ highest weights, but we view them as embedded in $SO(6)$). Then the quantization of the full physical phase space - the $SO(6)/U(3)$ rotations of this system - produces the representation $(Q+n_1, Q+n_2, Q+n_3)$.

We have seen, above, that a single D3-branes of charge $n_Z$ produces a DDBH in the $SO(6)$ representation $(Q_{n_Z}, Q, Q)$. Recall, however, that DDBHs with charge $(Q,Q,Q)$ are of rank 6, and so, generically, have 3 D3-branes. As an example, consider a DDBH with 3 D3-branes, each of charge $n_Z$. Around a fixed black hole background, the quantization of any one of these branes produces the $U(3)$ representation $(n_Z, 0, 0)$. As the branes are non interacting, their joint quantization produces the representations in the Clebsh Gordon decomposition of 3 of these representations, i.e. produces the $U(3)$ representations

$$\sum_{i=0}^{n_z} \sum_{j=0}^{n_z-i} \sum_{k=0}^{2i} (2n_Z + i - j - k, n_z - i + k, j).$$

---

[114]The states produced by the quantization of the classical manifold of solutions of this paragraph can be thought of as the charged analogues of the Revolving Black Holes (RBH)s of [8]. Recall RBHs were obtained by quantizing the space of solutions obtained by acting on the black holes with conformal generators $P_\mu$: the states so obtained fill out a conformal multiplet (the connection to coadjoint quantization, in that case, was also highlighted in [8]) In a similar manner, the quantization of the manifold of solutions obtained from the action of $O(6)$ generators on a charged black hole produce an $SO(6)$ representation. Unlike RBHs, however, the $SO(6)$ representations produced here do not give us a candidate end point for the charged instability. The main difference is that $SO(6)$ representations are finite dimensional, unlike their $SO(4,2)$ counterparts. We thank S. Kundu for a discussion on this point.

[115]In other words, these representations have $n_{\bar{Z}} = 0$. This is the quantum version of the classical fact that the duals with the lowest energy by charge ratio are those that have $q_c = q_n$.

[116]We would like to thank A. Gadde for a very useful discussion on this topic.

As a consequence, the corresponding DDBHs are in the $SO(6)$ representations[117]

$$\sum_{i=0}^{n_z}\sum_{j=0}^{n_z-i}\sum_{k=0}^{2i}(Q+2n_Z+i-j-k,Q+n_z-i+k,Q+j)\,. \tag{205}$$

In particular, on setting $j = n_Z$, $i = k = 0$, we find the representation

$$(Q+n_Z,Q+n_Z,Q+n_Z)\,, \tag{206}$$

demonstrating that a black hole with the charges listed in (206) can, indeed, decay into a DDBH with three duals, each of charge $n_Z$, around a core black hole with charges $(Q,Q,Q)$.

The study of DDBHS with 3 duals of different charges proceeds similarly.

### 5.9.1 DDBHs of ranks 4 and 2

The discussion above has focussed on DDBHs of rank 6. The discussion of DDBHs of rank 4 is similar. Such DDBHs are constructed around black holes of charges $(Q,Q,Q_1)$ with $Q > Q_1$. These black holes preserve a $U(2) \times U(1)$ symmetry. Stable duals in this background are charged only under the $U(2)$. The quantization of 2 duals, each of charge $n_Z$, around such a black hole, produces DDBHs in representations

$$\sum_{i=0}^{n_Z}(Q+n_Z+i,Q+n_Z-i,Q_1)\,. \tag{207}$$

In particular, the choice $i = 0$ gives a DDBH with charges $(Q+n_z,Q+n_Z,Q_1)$.

Finally, DDBHs of rank 2 are built around black holes with charges $(Q_1,Q_2,Q_3)$. Stable duals in this background are charged only under the $U(1)$ corresponding to the largest charge $Q_1$. A dual of charge $n_z$ in this background yields the single $SO(6)$ representation with highest weights

$$(Q_1+n_Z,Q_2,Q_3)\,. \tag{208}$$

## 6 Gravitational backreaction of dual giants

In §4 and §5 we have studied the dynamics of dual giants in the probe approximation. This approximation is entirely sufficient for dual giants that carry charge of order $N$, as the gravitational backreaction of such duals is of order $\mathcal{O}((1/N))$ and so is negligible. When the duals carry charges of order $N^2$ however (as they do in generic DDBHs), we cannot ignore the gravitational back reaction of the probe: at the very least it must be taken into account while computing the boundary stress energy tensor for the solution.

Even though the gravitational backreaction cannot be ignored, in the situations of interest to this paper this backreaction can be computed very simply. The chief simplification is the following: the back reaction only needs to be studied at linear order. Nonlinearities are never important (they are always suppressed by inverse powers of $1/N$). The reason for this is the

---

[117]Here $i$ is the number of boxes of the second irrep in the first row when taking tensor product of it with the first irrep, hence after this tensor product, we get

$$\sum_{i=0}^{n_Z}(n_Z+i,n_Z-i,0)\,.$$

Then we take tensor product of each of these with the third irrep. Here, $j$ is the number of boxes in the third row (which has to be less than $n_z-i$ since no two boxes from third irrep can be antisymmetrized), and $k$ is the number of boxes from third irrep in second row.

following. The dual giant in a DDBH has a large back reaction only if it is very big. Locally (an order unity distance away from the brane) this point does not matter: the Newton's constant times the energy density of brane is of order $1/N$, so the backreaction on the metric is also of order $1/N$. Once we go out to distances large compared to the radial size, $\sqrt{N}$, of the brane, we start to 'see' its full charge and energy density. By this point, however, we are already far in the normalizable tail of the solution. Though the contribution of the dual giant to the coefficient $C$ of this normalizable tail is non negligible, the tail itself, is $\propto \frac{C}{r^4}$ (relative to leading order) and so is extremely small compared to unity (recall $r^4 \gg N^2$).

For the reasons described above, in this section we evaluate the gravitational backreaction of the dual giants in DDBH solutions in the linearized approximation. Actually, over most of this section we actually evaluate the linearized gravitational back reaction of dual giants propagating in pure $AdS_5 \times S^5$: the modification of the solution to include the central black hole is relatively simple, and is performed at the end of this section.

## 6.1 Gravitational solution for dual giants in $AdS_5 \times S^5$ in LLMs coordinates

The supergravity solution including the first (linearized) backreaction to a single dual giant, rotating in the $Z$ plane of the $S^5$ in $AdS_5 \times S^5$ may be obtained by specializing the (very general) solutions of Lin, Lunin and Maldacena (LLM) [23] (for a review, see Appendix J) to a simple special case. In the LLM construction, we shade in the $x_1 - x_2$ plane with one disk of radius $\rho_0 = \sqrt{\frac{N-1}{N}}$ centered at the origin and a second disk of radius $\rho_1 = \sqrt{\frac{1}{N}}$ centered at a distance $d$ away from the origin.[118] Inserting this shading into the LLM construction gives the backreacted metric corresponding to a single dual giant that carries a $Z$ plane $SO(6)$ charge equal to $d^2 - 1$.

The LLM construction gives the fully backreacted metric for this configuration. The back-reaction of the D3-brane is small (and so is approximately linear) when the distance from the brane is $r \gg g^{\frac{1}{4}}\sqrt{\alpha'}$ (see Appendix A for conventions). At smaller distances, the gravitational response to the D-brane is large, and naively leads to a space curved on length scale $g^{\frac{1}{4}}\sqrt{\alpha'}$. Of course spacetimes curved on these extremely small length scales are not well described by supergravity (or even by classical string theory). Consequently, the LLM solution is not reliable where nonlinear effects are significant. Instead we expect that the dynamics of this highly curved region (in the small $g$ limit) is that of a single D3-brane localized on this manifold. In other words, we expect the accurate description of our background to be given by the probe brane of the previous section, together with the linearized gravitational response to this brane. The linearized gravitational response, in turn, is obtained by linearizing the LLM solution described above in the strength of the probe brane source, i.e. in $\frac{1}{N}$. This linearization is simple to implement (see Appendix J for details), and yields the following metric.

$$
\begin{aligned}
ds^2 &= l_{N-1}^2 \left( ds_{(0)}^2 + ds_{(1)}^2 \right), \\
F^{(5)} &= F_{(0)}^{(5)} + F_{(1)}^{(5)}.
\end{aligned}
\tag{209}
$$

Here

$$
ds_{(0)}^2 = \left( ds_{AdS_5}^2 + ds_{S^5}^2 \right),
\tag{210}
$$

and

$$
(l_{N-1})^4 = 4\pi g_s (N-1)(\alpha')^2.
\tag{211}
$$

---

[118]Note that in this normalization a disk of radius 1 unit in the $x_1 - x_2$ plane would correspond to $AdS_5 \times S^5$ with $N$ units of 5-form flux

$ds^2_{AdS_5}$ and $ds^2_{S^5}$ are, respectively, the metric on a unit AdS$_5$ and $S^5$[119]

$$ds^2_{AdS_5} = -(1+r^2)dt^2 + \frac{dr^2}{1+r^2} + r^2 d\Omega_3^2,$$
$$ds^2_{S^5} = \sin^2\theta \, d\tilde\Omega_3^2 + \cos^2\theta d\phi^2 + d\theta^2. \tag{212}$$

$F^{(5)}_{(0)}$ is the round 5-form appropriate to the $AdS_5 \times S^5$ solution with $N-1$ units of flux, i.e.

$$\frac{F^{(0)}_{(5)}}{(l_{N-1})^4} = -4\epsilon_5 + 4\tilde\epsilon_5, \tag{213}$$

where $\epsilon_5$ is the volume form on the unit $AdS_5$ and $\tilde\epsilon_5$ is the volume form on the unit $S^5$[120]

The linearized correction to the pure $AdS_5 \times S^5$ metric is given by

$$ds^2_{(1)} = \rho_1^2 \Bigg[ -\frac{(r^2 + \sin^2\theta)^2}{2x^4} r^2 d\Omega_3^2 + \frac{(r^2 + \sin^2\theta)^2}{2x^4} \sin^2\theta d\tilde\Omega_3^2 \tag{214}$$
$$+ \left( \frac{(r^2 + \cos^2\theta)^2 - 4(1+r^2)\cos^2\theta - 1 + 4d\sqrt{1+r^2}\cos\theta\cos(\phi - t)}{2x^4} \right)(1+r^2)dt^2$$
$$+ \left( \frac{r^4 - \sin^4\theta}{2x^4} \right)\frac{dr^2}{1+r^2} + \left( \frac{r^4 - \sin^4\theta}{2x^4} \right)d\theta^2$$
$$+ \left( \frac{(r^2 + \cos^2\theta)^2 - 4(1+r^2)\cos^2\theta - 1 + 4d\sqrt{1+r^2}\cos\theta\cos(\phi - t)}{2x^4} \right)\cos^2\theta \, d\phi^2$$
$$- \frac{2d\sin(\phi - t)}{x^4}\left(dt(1+r^2) - \cos^2\theta d\phi\right)d\left(\sqrt{1+r^2}\cos\theta\right)$$
$$+ 2d\phi dt \frac{\cos\theta\left(2(1+r^2)\cos\theta - d\sqrt{1+r^2}(1+r^2+\cos^2\theta)\cos(\phi - t)\right)}{x^4} \Bigg] + O(\rho_1^4),$$

where

$$x^2 = \rho_0^2(r^2 + \cos^2\theta) + d^2 - 2d\rho_0\sqrt{1+r^2}\cos\theta\cos(\phi - t). \tag{215}$$

Recall that $\rho_0$ and $\rho_1$ are the constants defined on the first paragraph of this subsection.

Similarly, the first correction to the five-form field strength takes the form[121]

$$F^{(5)}_{(1)} = 4\left(F \wedge d\Omega_3 + \tilde F \wedge d\tilde\Omega_3\right), \tag{216}$$

---

[119]The relationship of these coordinates on the $S^5$ to those we used in section 2 is as follows. $\phi_3$ in section 2 is $\phi$ here. $l_3$ in section 2 is $\cos\theta$ here. And $l_1$, $l_2$, $\phi_1$, $\phi_2$ of section 2 can be taken to be coordinates on the $S^3$ here.
[120]In the language of subsection 2, $\tilde\epsilon_5 = \epsilon^{(4)}(\mathbb{CP}^2) \wedge (d\Psi + \Theta - A)$ (see (8)).
[121]Note that in our conventions, the five-form field strength is 4 times the field strength in conventions of [23].

where

$$F_{tr} = \frac{2(d^2-1)r\sin^4\theta\left(r^2+\sin^2\theta\right)}{2Nx^6},$$

$$F_{t\phi} = \frac{d\sqrt{r^2+1}\sin^3\theta\sin 2\theta\left(\sin^2\theta+r^2\right)^2\sin(t-\phi)}{2Nx^6},$$

$$F_{t\theta} = \frac{\sqrt{1+r^2}\sin^3\theta}{8Nx^6}\Big[8\sqrt{r^2+1}\cos\theta\left(\left(d^2\left(r^2+2\right)+2r^2+1\right)\cos\theta+d^2\cos^2\theta\cos(2t-2\phi)\right)$$
$$-4d\cos(t-\phi)\left(\left(d^2+3r^2+2\right)\cos 2\theta+d^2+2r^4+7r^2+4\right)\Big],$$

$$F_{r\theta} = \frac{dr\sin^3\theta\sin(t-\phi)(-\left(d^2+3r^2+2\right)\cos 2\theta-d^2+4d\sqrt{r^2+1}\cos^3\theta\cos(t-\phi)+2r^4+r^2)}{2N\sqrt{r^2+1}x^6},$$

$$F_{r\phi} = \frac{2r\sin^4\theta\cos\theta\left(r^2+\sin^2\theta\right)\left(d\left(\cos 2\theta+2r^2+3\right)\cos(t-\phi)-4\sqrt{r^2+1}\cos\theta\right)}{4N\sqrt{r^2+1}x^6},$$

$$F_{\theta\phi} = \frac{\sin^3\theta\cos\theta}{4Nx^6}\Big[2\cos^2\theta\left(2d^2\left(2\left(r^2+1\right)\cos^2(t-\phi)+2r^2+1\right)\right.$$
$$+\sin^2\theta\left(4d^2+3r^4+2r^2-1\right)+2\left(2r^2+1\right)\sin^4\theta\big)$$
$$+2d\sqrt{r^2+1}\cos(t-\phi)\cos\theta\left(2\cos^2\theta\left(2\cos 2\theta-5\left(r^2+1\right)\right)\right.$$
$$-\left(2\left(d^2-r^4\right)+\sin^2\theta\left(-\cos 2\theta+2r^2+1\right)\right)\big)$$
$$-\frac{1}{4}\left(r^2+1\right)\left(16r^4-8r^2\cos 2\theta+\left(r^2+1\right)\cos 4\theta+7r^2-1\right.$$
$$+4\cos^4\theta\left(-\cos 2\theta+2r^2+2\right)\big)\Big], \tag{217}$$

The components of $\tilde{F}_{\mu\nu}$ are given by

$$\tilde{F}_{tr} = \frac{r^3}{32Nx^6}\Big[\left(48d^2\left(r^2+1\right)-16r^4+17\right)\cos 2\theta-\cos 6\theta+2\left(6r^2+5\right)\cos 4\theta$$
$$+2\left(-8\left(4d^2+3\right)r^4-6\left(4d^2+1\right)r^2\right.$$
$$+2d\cos\theta\left(\sqrt{r^2+1}\cos(t-\phi)\left(-8d^2+\cos 4\theta-8\left(2r^2+3\right)\cos 2\theta\right.\right.$$
$$+24\left(r^4+r^2\right)-1\right)+8d\left(r^2+1\right)\cos\theta\cos(2t-2\phi)\big)+8d^2-16r^6+3\big)\Big],$$

$$\tilde{F}_{t\theta} = -\frac{4r^4\sin\theta\left(r^2+\sin^2\theta\right)\left(4\left(r^2+1\right)\cos\theta-d\sqrt{r^2+1}\left(\cos 2\theta+2r^2+3\right)\cos(t-\phi)\right)}{8Nx^6},$$

$$\tilde{F}_{t\phi} = -\frac{dr^4\sqrt{r^2+1}\cos\theta\left(\sin^2\theta+r^2\right)^2\sin(t-\phi)}{Nx^6},$$

$$\tilde{F}_{r\theta} = \frac{-dr^3\sin\theta\sin(t-\phi)}{8N\sqrt{r^2+1}x^6}\Big[8d\left(r^2+1\right)\left(d-2\left(r^2+1\right)^{1/2}\cos\theta\cos(t-\phi)\right)$$
$$-\cos 4\theta+4\left(3r^2+2\right)\cos 2\theta+4r^2+1\Big],$$

$$\tilde{F}_{r\phi} = -\frac{r^3\cos\theta}{8Nx^6}\Bigg[\frac{\cos(t-\phi)\left(d\left(8d^2\left(r^2+1\right)+\cos 4\theta+4\left(3r^2+4\right)\cos 2\theta+4r^2+7\right)\right)}{\sqrt{r^2+1}}$$
$$-4\cos\theta\left(\left(d^2+2\right)\cos 2\theta-2d^2\left(r^2+1\right)\cos(2t-2\phi)+3d^2\right)\Bigg],$$

$$\tilde{F}_{\theta\phi} = \frac{2\left(d^2-1\right)r^4\sin(2\theta)\left(-\cos(2\theta)+2r^2+1\right)}{8Nx^6}. \tag{218}$$

The linearized metric $ds^2_{(1)}$ and $F^{(1)}_{(5)}$, presented above are singular on the four dimensional surface given by $r = \sqrt{d^2-1}$, $\theta = 0$, $\phi = t$. This is precisely the world volume of the probe

brane (see the previous section) once we identify $d^2 - 1 = q_n$). Away from this singular manifold, we have explicitly checked (on Mathematica) that the solution presented above obeys the equations

$$
-\frac{1}{2}\nabla_\alpha\nabla^\alpha h_{\mu\nu} - \frac{1}{2}\nabla_\mu\nabla_\nu h^\alpha{}_\alpha + \frac{1}{2}\nabla_\alpha\nabla_\nu h^\alpha_\mu + \frac{1}{2}\nabla_\alpha\nabla_\mu h^\alpha_\nu - h^{\alpha\beta}R_{\mu\alpha\nu\beta} + \frac{1}{2}R_{\nu\alpha}h^\alpha_\mu + \frac{1}{2}R_{\mu\alpha}h^\alpha_\nu
$$

$$
= \left(2(F^{(5)}_{(1)})_{\mu\rho\sigma\delta\kappa}(F^{(5)}_{(0)})^{\rho\sigma\delta\kappa}_\nu - 4h^{\alpha\rho}(F^{(5)}_{(0)})_{\mu\rho\sigma\delta\kappa}(F^{(5)}_{(0)})^{\sigma\delta\kappa}_{\nu\alpha}\right),
\tag{219}
$$

$$
F^{(5)}_{(1)} = \star F^{(5)}_{(1)} + \frac{1}{2}h^\alpha_\alpha \star F^{(5)}_{(0)} - 5\sqrt{-g}\epsilon_{\mu_1\mu_2\cdots\mu_{10}}h^{\mu_6\nu_6}g^{\mu_7\nu_7}\cdots g^{\mu_{10}\nu_{10}}(F^{(5)}_{(0)})_{\nu_6\cdots\nu_{10}}dx^{\mu_1}\wedge\cdots dx^{\mu_5}.
$$

The nature of this singularity can be understood by zooming into the neighborhood of $r = \sqrt{d^2-1}$, $\theta = 0$, $\phi = t$. To study the neighbourhood of this point we define coordinates as follows.

In a small neighborhood of the location of the D-brane, the metric on $AdS_5 \times S^5$ takes the form

$$
ds^2 = l^2_{N-1}\left(\frac{dr^2}{d^2} - dt^2 \times d^2 + (d^2-1)d\alpha_i^2 + dy_j^2 + d\phi^2\right),
\tag{220}
$$

where we have parameterized points on $\tilde{\Omega}_3$ using a unit vector $\hat{n}_j$ in an embedding $R^4$ and defined

$$
\theta\hat{n}^j = y^j, \qquad j = 1\ldots 4,
$$

We have also used the symbols $\alpha_i$, $i = 1\ldots 3$ to denote a set of orthonormal coordinates in the tangent space of $\Omega_3$ around the point of interest. It may be checked that the coordinate change

$$
\begin{aligned}
l_{N-1}\theta\hat{n}^j &= y^j, && j = 1\ldots 4, \\
l_{N-1}(r - \sqrt{d^2-1}) &= d \times y^5, \\
l_{N-1}\alpha_i &= \frac{x_i}{\sqrt{d^2-1}}, && i = 1\ldots 3, \\
l_{N-1}(\phi - t) &= \frac{\sqrt{d^2-1}}{d}y^6, \\
l_{N-1}x^0 &= \frac{d^2 t - \phi}{\sqrt{d^2-1}},
\end{aligned}
\tag{221}
$$

turns the metric (220) into

$$
ds^2 = dx_\mu dx^\mu + dy_i dy^i, \qquad \mu = 0\ldots 3, \qquad i = 1\ldots 6.
\tag{222}
$$

Note that our locally Minkowskian coordinates $x^\mu$ are all tangent to the brane, while the locally Euclidean coordinates $y^i$ are all transverse to the brane. To leading order, it is easily verified that the quantity $x^2$ defined in (215) reduces to

$$
x^2 = (d^2-1)\frac{(y_i y^i)}{l^2_{N-1}}.
\tag{223}
$$

Plugging (221) and (223) into the metric (214), and the field strength (217) (218), we find that these reduce, at leading order (in deviation from the brane) to (A.12), (A.13) and (A.10) respectively. When we go very near the brane, our solution reduces to the solution of an infinite brane in flat space, as expected on physical grounds.

In particular the non-zero components of the field strength in this limit and the coordinates (221) are the following:

$$
F_{y^i x^0 x^1 x^2 x^3} = -16\pi\alpha' g_s^2 \frac{y^i}{(y^j y_j)^3}.
\tag{224}
$$

Similarly the metric backreaction from the D-brane written in (214) takes the following form:

$$h_{\mu\nu} = -2\pi\alpha' g_s^2 \frac{1}{(y_j y^j)^2}\eta_{\mu\nu},$$

$$h_{ij} = 2\pi\alpha' g_s^2 \frac{1}{(y_j y^j)^2}\delta_{ij}. \tag{225}$$

Hence we see that the backreaction of the brane on field strength (224) and metric (225) are in agreement with the backreaction expected from a flat D3-brane (compare with (A.8) and (A.12) respectively).

It follows, in other words, that our solution obeys the linearized Einstein's equation and Gauss' law for the field strength with the stress tensor as in (A.11), i.e.

$$G_{AB} = \frac{1}{2}\left(\partial^C\partial_A h_{BC} + \partial^C\partial_B h_{AC} - \Box h_{AB} - \partial_A\partial_B h - \partial^C\partial^D h_{CD}\eta_{AB} + \Box h\eta_{AB}\right) = 8\pi G_N T_{AB},$$

$$\nabla_i F^{i\mu\nu\rho\sigma} = -16\pi^4 g_s\alpha'^2 \epsilon^{\mu\nu\rho\sigma}\delta^6(\vec{y}). \tag{226}$$

Where in the first equation $T_{AB}$ (we have denoted all 10d components by capital alphabets $A, B$) is only along the worldvolume directions as is the same as in (A.11). Also note that the second equation is in agreement with (A.7).

## 6.2 The metric in the extreme large $r$ limit

Starting with the metric in LLM coordinates, it is easy to check that the metric correction takes the following form in the extreme large $r$ limit (we have retained all terms of the same order as the large $r$ reduction of the background $AdS$ metric, $ds_{(0)}^2$)

$$\frac{ds_{(1)}^2}{l_{N-1}^2} \approx \left(\frac{r^2}{2}\right)\frac{dt^2}{N} + \left(\frac{1}{2r^2}\right)\frac{dr^2}{N} - \frac{r^2}{2N}d\Omega_3^2 + \frac{1}{2N}d\Omega_5^2. \tag{227}$$

Recall that the full metric took the form $ds^2 = ds_{(0)}^2 + ds_{(1)}^2$. Always working only to first subleading order in $\frac{1}{N}$, the coordinate transformation

$$r = r' + \frac{r'}{2N}, \tag{228}$$

changes $ds_{(0)}^2$ by a term of order $\frac{1}{N}$. This change effectively shifts $ds_{(1)}^2$. Accounting for this shift (and replacing $r'$ with $r$ for simplicity) we find that $ds_{(1)}^2$ becomes

$$\frac{ds_{(1)}^2}{l_{N-1}^2} \approx \frac{1}{2N}\left(-r^2 dt^2 + \left(\frac{dr^2}{r^2}\right) + r^2 d\Omega_3^2 + d\Omega_5^2\right)$$

$$= \frac{1}{2N}(ds_{AdS_5}^2 + d\Omega_5^2), \tag{229}$$

where $ds_{AdS_5}^2$ is the large $r$ (or Poincare patch) metric on a unit radius $AdS_5$.

$$ds^2 = ds_{(0)}^2 + ds_{(1)}^2 = l_{N-1}^2\left(1 + \frac{1}{2N}\right)\left(ds_{Ads_5}^2 + d\Omega_5^2\right) = l_N^2\left(ds_{Ads_5}^2 + d\Omega_5^2\right), \tag{230}$$

where we have used

$$\frac{l_N^2}{l_{N-1}^2} = 1 + \frac{1}{2N}.$$

It follows, in other words, that corrected metric asymptotes (at large $r$) to $AdS_5 \times S^5$ with radius $l_N$, as expected on physical grounds.

## 6.3 Asymptotic gauge field and charge

As we have mentioned above, the leading order (in the large $r$ expansion) correction to the metric can be absorbed into a renormalization of the radius of $AdS_5 \times S^5$. After removing this constant piece, rest of the correction metric decays at infinity (in comparison with the leading $AdS_5 \times S^5$ metric). After performing the additional coordinate change

$$
\begin{aligned}
r &= r' + \frac{1}{N}\left(d\cos\theta'\cos(\phi'-t') - \frac{2\cos^2\theta\left(1-3d^2\cos^2(\phi'-t')\right)+\left(d^2-1\right)}{2r'}\right), \\
\phi &= \phi' - \frac{d\sin(\phi'-t')}{r'\cos\theta' N}, \\
\theta &= \theta' - \frac{d\sin\theta'\cos(\phi'-t')}{Nr'},
\end{aligned}
\tag{231}
$$

(and once again, for simplicity, dropping all primes in the resultant formulae) we find that the metric correction decays like $\mathcal{O}(1/r^2)$ (in comparison to the background $AdS_5 \times S^5$ metric). Keeping all terms to relative order $\mathcal{O}(1/r^2)$ we find

$$
\begin{aligned}
\frac{ds^2}{l_N^2} \approx ds^2_{AdS_5 \times S^5} + \frac{1}{N}\Bigg[ &-\left(2\cos^2\theta\left(1-3d^2\cos^2(\phi-t)\right)+\left(d^2-1\right)\right)dt^2 \\
&+ \frac{4\cos^2\theta\left(1-2d^2\cos^2(\phi-t)\right)}{r^2}d\phi\,dt \\
&+ \left(\frac{\sin^2\theta\left(\cos^2\theta\left(6d^2\cos^2(\phi-t)-1\right)-d^2+\sin^2\theta\right)}{r^2}\right)d\tilde{\Omega}_3^2 \\
&+ \left(\frac{\cos^2\theta\left(6d^2\cos^2(\phi-t)-1\right)-d^2}{r^4}\right)dr^2 \\
&+ \left(\cos^2\theta\left(1-6d^2\cos^2(\phi-t)\right)+d^2-\sin^2\theta\right)d\Omega_3^2 \\
&+ \left(\frac{2d\cos\theta\cos(\phi-t)}{r} + \frac{\cos^2\theta\left(6d^2\cos^2(\phi-t)-1\right)-d^2}{r^2}\right)d\theta^2 \\
&- \left(\frac{d^2\sin(2\theta)\sin(2t-2\phi)}{r^2}\right)d\theta\,dt\Bigg].
\end{aligned}
\tag{232}
$$

Those terms in the metric (232) that are proportional to $dx^\mu dx^\nu$ (here $\mu,\nu$ are $AdS_5$ indices) can be removed from the metric by further coordinate changes of the form (231) (the first 'physical' data in this part of the metric is at relative order $\mathcal{O}(1/r^4)$: these terms encode the boundary stress tensor, see below). At this order, some terms in the metric of the form $dx^\mu dx^\alpha$ (here $\alpha$ is a coordinate on the $S^5$) can also be removed by a coordinate transformation. However terms of the form $(\zeta^\alpha dx^\alpha)$ (where $\zeta^\alpha$ is one of the 15 killing vectors on $S^5$) are physical at this order: the coefficient of such terms compute the expectation value of the $SO(6)$ gauge field.

In the metric (232) we can extract the expectation values of the $SO(6)$ gauge field - corresponding to the $S^5$ killing vector $\zeta$ - using the formula

$$
A_\mu^\zeta = -3\frac{\int d\Omega_5\,\zeta^\alpha g_{\alpha\mu}}{\int d\Omega_5}.
\tag{233}
$$

The gauge field in (233) is normalized to ensure that its charges (obtained from Gaussian integrals) are $N^2$ times charges of the dual field theory.

Using (233), we find that the gauge field in the metric (232) is purely in the $SO(6)$ $\partial_\phi$ direction (i.e. the gauge field is in the $U(1)$ that rotates one of the three planes in the $C^3$ into which we can imagine embedding $S^5$). We find that the value of this gauge field is

$$A = \frac{2(d^2 - 1)}{N} \frac{dt}{r^2}. \tag{234}$$

The charge contained in this gauge field is easily extracted (by performing a Gaussian integral at infinity and multiplying by $N^2$): we find

$$Q = N(d^2 - 1). \tag{235}$$

This is precisely the charge predicted from the probe analysis (recall that $d^2 - 1 = r_*^2$ where $r_*$ is the $AdS_5$ radial location of the probe).

### 6.3.1 The boundary stress tensor

The boundary stress tensor of our solution is encoded in the five dimensional $AdS_5$ metric. This metric is obtained by projecting those terms that are proportional to $dx^\mu dx^\nu$ (here $\mu$ and $\nu$ are coordinates on $AdS_5$) to the space of $SO(6)$ singlets. This projection is implemented by integrating the components of $g_{\mu\nu}$ over $S^5$ (using the Haar measure) and then dividing by the volume of the $S^5$. The metric so obtained can then be power expanded in $\frac{1}{r}$. Keeping terms in metric up to fractional subleading order $\frac{1}{r^4}$ we find

$$ds^2 = \left( -(1 + r^2) + \frac{6d^2 + 6r^4 + 2r^2 - 7}{6Nr^2} \right) dt^2 + \left( \frac{r^4 - r^2 + 1}{r^6} - \frac{r^2 - 1}{3Nr^6} \right) dr^2$$
$$+ \left( r^2 - \frac{-2d^2 + 6r^4 + 2r^2 + 1}{6Nr^2} \right) d\Omega_3^2 \tag{236}$$

(this metric is reported in the original coordinates, i.e. before we have performed the coordinate transformations (228) and (231)). We now read off the boundary stress tensor dual to this metric following the procedure of [35]. We cut the metric (236) off at some large value of $r$. The unit normal vector to our constant $r$ cut off surface is given by

$$n^\mu = \left( 0, \frac{1}{\sqrt{g_{rr}}}, 0, 0, 0 \right). \tag{237}$$

The extrinsic curvature of the constant $r$ slice surface is easily computed using the formula

$$\Theta^\mu_{\ \nu} = -\frac{1}{2} \left( \nabla^\mu n_\nu + \nabla_\nu n^\mu \right). \tag{238}$$

The stress tensor is then given by the formula[122]

$$T^\mu_{\ \nu} = \frac{N^2 r^4}{4\pi^2} \left( \Theta^\mu_{\ \nu} - \Theta \gamma^\mu_{\ \nu} - 3\gamma^\mu_{\ \nu} + \frac{1}{2} G^\mu_{\ \nu} \right), \tag{239}$$

where the last two terms are the counter terms added to remove the divergences in the stress tensor in any asymptotically $AdS_5$ metric [30, 35] (here $\gamma^\mu_{\ \nu}$ and $G^\mu_{\ \nu}$ are the induced metric at fixed $r$ and the Einstein tensor respectively). The above stress tensor exactly matches

---

[122]The extra factor of $r^4$ in the definition of $T^\mu_{\ \nu}$ will cancel the factors of $r$ coming from volume and $n_\mu$ in the definition of energy.

the one computed in [30]. Performing the necessary computations (recall our metric is $ds^2 = ds^2_{(0)} + ds^2_{(1)}$) we find

$$
\begin{aligned}
T^0_0 &= \frac{(N-1)^2}{4\pi^2}\left(\frac{3}{8}\left(1+\frac{2}{N}\right) + \frac{2(d^2-1)}{N}\right) \approx \frac{N^2}{4\pi^2}\left(\frac{3}{8} + \frac{2(d^2-1)}{N}\right), \\
T^i_j &= -\frac{\delta^j_i}{3}\frac{(N-1)^2}{4\pi^2}\left(\frac{3}{8}\left(1+\frac{2}{N}\right) + \frac{2(d^2-1)}{N}\right) \approx -\frac{\delta^j_i}{3}\frac{N^2}{4\pi^2}\left(\frac{3}{8} + \frac{2(d^2-1)}{N}\right)
\end{aligned}
\tag{240}
$$

(the two terms on the RHS of each of the equations above differ only at order unity, i.e. fractional subleading order $\frac{1}{N^2}$).

We see that our giant graviton corrects the pure AdS boundary stress tensor in a very simple manner. Note that the correction carries positive energy but negative pressure, with relative coefficients of the form that ensures the boundary stress tensor is traceless, as expected on general grounds for a boundary CFT

The total energy of our solution is obtained by integrating the stress tensor over the boundary $S^3$:

$$
E = \int d^3x \sqrt{\gamma}\, T^0_0 = \frac{N^2}{2}\left(\frac{3}{8}\left(1+\frac{2}{N}\right) + \frac{2(d^2-1)}{N}\right).
\tag{241}
$$

The excess energy of our solution (above that of the vacuum $AdS_5$) equals $N(d^2-1)$ as predicted by the probe analysis.

## 6.4 The 5-form field strength at large $r$

In the new coordinates (i.e. after having performed the coordinate transformations (228) and (231)) we find $F$ and $\tilde{F}$ reduce, in the large $r$ limit, to

$$
\delta \tilde{F} = -\frac{\sin^3 \theta \cos \theta}{N} d\theta \wedge d\phi\,,
\tag{242}
$$

and

$$
\delta F = \frac{r'^3}{N} dt \wedge dr'\,,
\tag{243}
$$

with corrections at fractional order $1/r$. It follows that, at large $r$, the total 5-form field strength of our solution equals[123]

$$
F_{(5)} = l^4_{N-1}\left(1 + \frac{1}{N}\right)(-4\epsilon_5 + 4\tilde{\epsilon}_5) \approx l^4_N(-4\epsilon_5 + 4\tilde{\epsilon}_5).
\tag{244}
$$

Consequently, the probe brane causes the 5-form field strength to jump from $N-1$ (at small $r$) to $N$ (at large $r$) as expected on physical grounds.

## 6.5 Expectation value of Tr($Z^n$)

The metric (and five form) corrections presented above clearly do not lie within 5d gauged supergravity. As a consequence, infinite classes of single trace operators - including operators whose dual fields do not lie within the gauge supergravity truncation - have nonzero one point

---

[123]Note that the correction to the five form field strength is

$$
\delta F_{(5)} = \frac{l^4_{N-1}}{N} 4(F \wedge d\Omega_3 + \tilde{F} \wedge d\tilde{\Omega}_3) = \frac{l^4_{N-1}}{N}(-4\epsilon_5 + 4\tilde{\epsilon}_5).
$$

functions within our solution.[124] As an illustration of this point, in this section, we extract the expectation values (in the solution of interest) of the operators $\text{Tr}(Z^n)$ for all values of $n$.

Consider the components of the five form field strength, whose legs all lie along the five $S^5$ directions. This part of the field strength can be written as

$$F_5 = \sqrt{g'}\tilde{\epsilon}_5 \chi \, . \tag{245}$$

Here $g'$ is the determinant of restriction of the metric to the five $S^5$ directions, and $\chi$ is a scalar function.

Let us now examine how $\chi$ transforms under coordinate changes of the form $x' = x + \frac{1}{N}h(x)$. Keeping track only of terms of order $1/N$, such coordinate transformations change the field strength only because of the action of the gauge transformation on the background $F$. In the background, the only nonzero components of $F$ were those with all legs in either the $S^5$ or the $AdS_5$ directions. Using the equation

$$dx^\mu = dx'^\mu + \frac{1}{N}\frac{\partial x^\mu}{\partial x'^\nu}dx'^\nu \, , \tag{246}$$

we see that this background field strength changes under coordinate transformations. At leading order in $1/N$, however, the change (due to coordinate changes) in the part of the field strength retained in (245) comes purely from those terms in (246) in which $\mu$ and $\nu$ both lie in the $S^5$ directions. But this change is exactly compensated for by the change, under coordinate transformations, of $\sqrt{g'}$. To the order in $1/N$ under study, consequently, $\chi$ defined in (245) is coordinate invariant.

It was demonstrated in [36] that on any solution of the linearized equations of motion, the scalar $\chi$ can be decomposed as

$$\chi = \sum_k (-k)(k+4)\left(-s^I + t^I\right)Y^I \, , \tag{247}$$

where $I$ is an 'internal' index for spherical harmonics with $k$ boxes in the first row of the Young Tableaux, $s^I$ is the field dual to the various chiral primary operators built out of $k$ scalars, and $t^I$ is the field dual to a particular scalar decedent (of dimension $k + 2$) of the $k^{th}$ chiral primary field.

As the fields $s^I$ are dual to operators of dimension $k$, they will decay at large $r$ like

$$s^I = -\frac{\frac{\alpha^I}{k}}{r^k} \, . \tag{248}$$

The coefficient $\alpha^I$ is the one point function (see equation 10 of [37]) of the chiral primary operator, chosen so that the index $I$ corresponds to an orthonormalized basis in the space of spherical harmonics of angular momentum $k$, and is normalized to have the two point function listed in 4.5 of [36] with $w^I = 1$).

Using the explicit solution presented in this paper, it is not too difficult to read off the coefficient $\alpha^I$ for the $S^5$ spherical harmonic proportional to $\cos^k \theta e^{ik\phi}$ (see (K.9) for the normalized version of this spherical harmonic).[125] The computation is presented in Appendix K: we find

$$\alpha^I = \frac{4i^k}{N(k+4)}\left(\frac{(k^2-1)}{\sqrt{2(k+1)(k+2)}}\right)(d)^k \, . \tag{249}$$

---

[124]The state dual to our supergravity solution is expected to be obtained (in radial quantization) by acting on the vacuum with linear combinations of the operators $\prod_m \text{Tr}(Z^m)^{n_m}$. As a consequence, one should expect that an operator $O$ with nonzero three point function $\left\langle \text{Tr}\bar{Z}^{m'} O \text{Tr}Z^m \right\rangle$ - for some $m$ and $m'$ - will, generically, have nonzero expectation value in this state. Of course, an example of such an operator is $\text{Tr}(Z^a)$.

[125]This field is dual to the suitably normalized version of the operator $\text{Tr}(Z^k)$.

A striking feature of this answer is the factor $d^k$, the exponential dependence of the expectation value on $k$. This behavior can be intuitively understood as follows. Consider a matrix $Z$, one of whose eigenvalues is taken to be large (of order $d$). It follows that $\mathrm{Tr}Z^k$ computed on such a matrix equals $d^k$, exactly as seen in (249).

## 6.6 Solution in an alternative coordinate system

Consider the metric and Field strength listed in 6.1. Let us now assume that $d \gg 1$ (as is the case for DDBH solutions). In this subsection we focus on the 'center of $AdS$' i.e. $r$ of order unity (this is where the black hole will eventually sit in the DDBH solution). At these values of $r$, we ask how the metric scales with $d$. Expanding the correction in a power series expansion in $\frac{1}{d}$ (at fixed $r$), we find that the metric correction (214) starts at order $\frac{1}{d^3}$; this leading order piece takes the explicit form

$$
\begin{aligned}
ds^2_{(1)} = \frac{1}{d^3 N}\Bigg[ &(2\left(1+r^2\right)^{1/2}\cos\theta\cos(t-\phi)) ((1+r^2)dt^2 + \cos^2\theta d\phi^2) \\
&+ \left(\frac{2r\cos\theta\sin(t-\phi)}{\sqrt{r^2+1}}dr - 2\left((r^2+1)^{1/2}\sin\theta\sin(t-\phi)\right)d\theta\right)((1+r^2)dt - \cos^2\theta d\phi) \\
&- 2\left(\sqrt{r^2+1}\cos\theta\left(\cos^2\theta + r^2 + 1\right)\cos(t-\phi)\right)dt d\phi\Bigg] + \mathcal{O}\left(\frac{1}{d^4 N}\right).
\end{aligned}
\tag{250}
$$

Similarly, the 5-form field strength appears to start out at order $O\left(\frac{1}{d^3}\right)$ and takes the form

$$
\begin{aligned}
F_{t\theta} = F_{\theta\phi} &= -\frac{\sqrt{r^2+1}\sin^3\theta\cos^2\theta\cos(t-\phi)}{d^3 N}, & F_{r\theta} &= -\frac{r\sin^3\theta\cos^2\theta\sin(t-\phi)}{d^3 N\sqrt{r^2+1}}, \\
\tilde{F}_{tr} = \tilde{F}_{r\phi} &= -\frac{r^3\sqrt{r^2+1}\cos\theta\cos(t-\phi)}{d^3 N}, & \tilde{F}_{r\theta} &= -\frac{r^3\sqrt{r^2+1}\sin\theta\sin(t-\phi)}{d^3 N}.
\end{aligned}
\tag{251}
$$

It might thus, at first, seem that the correction metric starts at order $\frac{1}{d^3}$ (at values of $r$ of order unity). However sample computations of invariants (like $R_{\mu\nu\alpha\beta}R^{\mu\nu\alpha\beta}$ and $F_{\mu\nu\alpha\beta\rho}F^{\mu\nu\alpha\beta\rho}$) on the full metric (background plus fluctuation) and the full 5-form field strength (background plus fluctuation) reveal that these invariants deviate from those in pure $AdS_5 \times S^5$ by a term of order $\frac{1}{d^4}$. Indeed the terms in (250) and (251) are 'pure gauge'. Always working only to order $1/N$ (i.e. within the linearized theory), we find that the metric fluctuation in (250) and the field strength fluctuation in (251) are both removed by the coordinate change

$$
\begin{aligned}
\phi' &= \phi - \frac{\sqrt{r^2+1}\cos\theta\sin(t-\phi)}{x^3 N}, \\
t' &= t - \frac{\sqrt{r^2+1}\cos\theta\sin(t-\phi)}{x^3 N}.
\end{aligned}
\tag{252}
$$

In the new coordinate system (whose coordinates $\phi'$ and $t'$ we rename as $\phi$ and $t$, for sim-

plicity), we find that the metric correction takes the form,

$$
\begin{aligned}
ds^2_{(1)} = \left(\frac{1}{2Nx^4}\right)\Bigg[ & (r^2+\sin^2\theta)^2\left(r^2 d\Omega_3^2 + \sin^2\theta\, d\tilde\Omega_3^2\right) + (r^4-\sin^4\theta)\left(\frac{dr^2}{1+r^2}+d\theta^2\right) \\
& + \Bigg( (r^2+\cos^2\theta)^2 - 4(1+r^2)\cos^2\theta - 1 + 4d\sqrt{1+r^2}\cos\theta\cos(\phi-t) \\
& + \frac{4\sqrt{1+r^2}\cos\theta}{x}\left(3d\sqrt{1+r^2}\cos\theta\sin^2(t-\phi)-x^2\cos(t-\phi)\right)\Bigg)\left((1+r^2)dt^2+\cos^2\theta\,d\phi^2\right) \\
& + \Bigg(4\left(2(1+r^2)\cos\theta-d\sqrt{1+r^2}(1+r^2+\cos^2\theta)\cos(\phi-t)\right)+ \\
& + \frac{2\sqrt{1+r^2}(3+2r^2+\cos2\theta)}{x}\left(-3d\sqrt{1+r^2}\cos\theta\sin^2(t-\phi)+x^2\cos(t-\phi)\right)\Bigg)\cos\theta\,d\phi\,dt \\
& + \frac{12}{x}\sqrt{1+r^2}\cos\theta\sin(\phi-t)\left(dt(1+r^2)-\cos^2\theta\,d\phi\right)(r\,dr-\sin\theta\cos\theta\,d\theta) \\
& + \sin(\phi-t)\left(dt(1+r^2)-\cos^2\theta\,d\phi\right)\left(\frac{r\cos\theta\,dr}{\sqrt{1+r^2}}-\sqrt{1+r^2}\sin\theta\,d\theta\right)\left(4(x-d)\right. \\
& + 12\frac{d}{x}\sqrt{1+r^2}\cos\theta\cos(t-\phi)\Bigg)\Bigg],
\end{aligned}
\tag{253}
$$

and the field strength,

$$
F_{tr} = \frac{2\left(d^2-1\right)r\sin^4\theta\left(r^2+\sin^2\theta\right)}{2Nx^6},
\tag{254}
$$

$$
F_{t\phi} = \frac{d\sqrt{r^2+1}\sin^3\theta\sin2\theta\left(\sin^2\theta+r^2\right)^2\sin(t-\phi)}{2Nx^6},
$$

$$
\begin{aligned}
F_{t\theta} = \frac{\sin^3\theta}{2Nx^6}\Bigg[ & 2\left(r^2+1\right)\left(d^2\left(r^2+2\right)+2r^2+1\right)\cos\theta + 2\cos^2\theta\sqrt{(r^2+1)}x^3\cos(t-\phi) \\
& + d\left(r^2+1\right)\cos^3\theta\left((2d+3x)\cos(2t-2\phi)-3x\right) \\
& - d\sqrt{r^2+1}\cos(t-\phi)\left(\left(d^2+3r^2+2\right)\cos(2\theta)+d^2+2r^4+7r^2+4\right)\Bigg],
\end{aligned}
$$

$$
\begin{aligned}
F_{r\theta} = \frac{r\sin^3\theta\sin(t-\phi)}{2N\sqrt{r^2+1}x^6}\Bigg[ & d\left(\sin^2\theta+r^2\right)^2 + 6dx\cos^3\theta)\sqrt{(r^2+1)}\cos(t-\phi) \\
& + 2x\cos^2\theta\left(-dx-3r^2+x^2-3\right)\Bigg],
\end{aligned}
$$

$$
F_{r\phi} = \frac{2r\sin^4\theta\cos\theta\left(r^2+\sin^2\theta\right)\left(d\left(\cos2\theta+2r^2+3\right)\cos(t-\phi)-4\sqrt{r^2+1}\cos\theta\right)}{4N\sqrt{r^2+1}x^6},
$$

$$
\begin{aligned}
F_{\theta\phi} = \frac{\sin^3\theta\cos\theta}{8Nx^6}\Bigg[ & 8\cos^2\theta\left(\left(2r^2+1\right)\left(d^2+\sin^4\theta\right)+2d^2\left(r^2+1\right)\cos^2(t-\phi)\right. \\
& \left. -3d\left(r^2+1\right)x\sin^2(t-\phi)\right) \\
& + \sqrt{r^2+1}\cos\theta\cos(t-\phi)\left(-d\left(8d^2-8r^4+4r^2+3\right)-d\cos4\theta\right. \\
& \left. +4d\left(r^2+1\right)\cos(2\theta)+8x^3\right) \\
& - 2\left(\left(d^2+r^4+r^2\right)\cos(4\theta)-d^2+4r^6+5r^4+r^2-2\left(r^4+r^2\right)\cos(2\theta)\right) \\
& + 8d\sqrt{r^2+1}\cos^3\theta\left(2\cos2\theta-5\left(r^2+1\right)\right)\cos(t-\phi) \\
& - 8\left(r^2+1\right)\cos^4\theta\left(\cos2\theta-2\left(r^2+1\right)\right)\Bigg].
\end{aligned}
\tag{255}
$$

The components of $\tilde{F}_{\mu\nu}$ are given by

$$
\begin{aligned}
\tilde{F}_{tr} = \frac{r^3}{8Nx^6}\bigg[ &-2\left(\cos 2\theta - 2r^2 - 1\right)\left(\left(d^2 + r^4\right)\cos 2\theta - d^2\left(4r^2 + 1\right) - r^4\right) \\
&+ \cos\theta\left(\sqrt{r^2 + 1}\cos(t - \phi)\left(d\left(-8d^2 + 24\left(r^4 + r^2\right) - 1\right)\right.\right. \\
&\left.+ d\left(\cos 4\theta - 8\left(2r^2 + 3\right)\cos 2\theta\right) + 8x^3\right) \\
&+ r^2\cos 3\theta\right) + \cos^2\theta\left(2\left(2d^2 + \left(r^2 + 2\right)^2\right)\cos 2\theta + 4d\left(r^2 + 1\right)(2d + 3x)\cos(2t - 2\phi) \\
&+ 12d\left(d - \left(r^2 + 1\right)x\right) - 2\cos 4\theta - 8r^6 - 22r^4 - 13r^2 - 6\right) \\
&+ 4\cos^4\theta\left(\cos 2\theta + r^4 + r^2 + 1\right)\bigg],
\end{aligned}
$$

$$
\tilde{F}_{t\theta} = -\frac{4r^4\sin\theta\left(r^2 + \sin^2\theta\right)\left(4\left(r^2 + 1\right)\cos\theta - d\sqrt{r^2 + 1}\left(\cos 2\theta + 2r^2 + 3\right)\cos(t - \phi)\right)}{8Nx^6},
$$

$$
\tilde{F}_{t\phi} = -\frac{dr^4\sqrt{r^2 + 1}\cos\theta\left(\sin^2\theta + r^2\right)^2\sin(t - \phi)}{Nx^6},
$$

$$
\begin{aligned}
\tilde{F}_{r\theta} = -\frac{r^3\sin\theta\sin(t - \phi)}{8N\sqrt{r^2 + 1}x^6}\bigg[ &8\left(r^2 + 1\right)\left(d^3 - x\left(d^2 + r^2 - 1\right)\right) + 4dr^2 + d \\
&- d\left(8\left(r^2 + 1\right)^{3/2}\cos\theta\left(2d + x\right)\cos(t - \phi) + \cos 4\theta\right) \\
&+ 4\cos 2\theta\left(d\left(3r^2 + 2\right) + 2\left(r^2 + 1\right)x\right)\bigg],
\end{aligned}
$$

$$
\begin{aligned}
\tilde{F}_{r\phi} = \frac{r^3\cos\theta}{8Nx^6}\bigg[ &2\left(7d^2 + 2\right)\cos\theta + 2\left(d^2 + 2\right)\cos 3\theta + 8d^2\left(r^2 + 1\right)\cos\theta\cos(2t - 2\phi) \\
&- \frac{d\cos(t - \phi)\left(8d^2\left(r^2 + 1\right) + \cos 4\theta + 4\left(3r^2 + 4\right)\cos 2\theta + 4r^2 + 7\right)}{\sqrt{r^2 + 1}} \\
&- 24d\left(r^2 + 1\right)x\cos\theta\sin^2(t - \phi) + 8\sqrt{r^2 + 1}x^3\cos(t - \phi)\bigg],
\end{aligned}
$$

$$
\tilde{F}_{\theta\phi} = \frac{2\left(d^2 - 1\right)r^4\sin 2\theta\left(-\cos 2\theta + 2r^2 + 1\right)}{8Nx^6}. \tag{256}
$$

It may be checked that the expansion of each of (253), (255) and (256) starts out at $\mathcal{O}\left(\frac{1}{d^4}\right)$ in a power series expansion at fixed $r$.

## 6.7 Approximate solution for a black hole surrounded by a large dual giant

Given the discussion presented earlier in this section, it follows that the bulk solution that approximately describes a black hole surrounded by a dual giant, takes the form (209), with $F_{(0)}^{(5)}$ given as in (213), $F_{(1)}^{(5)}$ given by the sum of terms of the form (255) and (256) (one for each dual giant), $ds_{(0)}^2$ given in (1) and $ds_{(1)}^2$ given by the sum of terms in (253) (one for each dual giant). In other words, the supergravity solution that captures the dual giant 'surrounding' a black hole is approximately given by the metric of the dual giant with the pure $AdS_5 \times S^5$ metric in (210) replaced by the black hole metric reviewed earlier in this paper.[126]

---

[126] We have chosen to perform the replacement in the coordinates of §6.6 rather than §6.1 because the metric in the coordinates of §6.6 deviates from pure AdS (at $r$ of order unity) only at order $d^4$. As a consequence, the replacement of $AdS_5 \times S^5$ by a black hole metric results in smaller error, i.e. will a smaller RHS of Einstein's equations.

## 6.8 Estimate of the errors in this solution

In this section we have presented bulk solutions for DDBHs that are correct in the large $N$ limit. Recall that DDBHs are solutions that consist of two distinct sets of constituents:

- A core black hole.

- One, two or three surrounding dual giant gravitons.

DDBHs solutions are simple for two reasons.

- First because the two different sets of components are very well separated away from each other.

- Second, because the number of duals is of order unity, so the back reactions of these duals can be handled perturbatively.

Neither of these points are exact: the components of a DDBH are not infinitely separated, and the backreaction of the probe D3-branes is not exactly linear. Each of these points introduces errors into the solutions presented in this section.

We refer to errors arising from the finiteness of separation between duals and the core black hole as 'irreducible'. These errors are governed by a small parameter $\epsilon^{\mathrm{irred}}$ which we will estimate below. $\epsilon^{\mathrm{irred}}$ parameterizes the extent to which a DDBH can no longer be regarded as a non interacting mix of two systems. It also governs the deviation of thermodynamics of DDBHs from those of the non interacting model of §3.1. While it would be possible (and interesting, see §7 for further discussion) to perform a matched asymptotic expansion that systematically improve DDBH solutions as a power series $\epsilon^{\mathrm{irred}}$, it is harder to imagine working exactly in $\epsilon^{\mathrm{irred}}$[127] and so completely eliminating these errors.

Errors arising from the backreaction of D3-branes are paremeterized by a second small parameter $\epsilon^{\mathrm{red}}$, again estimated below. These errors are less serious. The underlying reason for this is that our system is supersymmetric[128] when $\epsilon^{\mathrm{irred}} = 0$, and so $\epsilon^{\mathrm{red}}$ parameterizes nonlinear supersymmetric corrections to a linearized (supersymmetric) solution.

In the case that the DDBH has only one dual giant, the errors parameterized by $\epsilon^{\mathrm{red}}$ can be exactly corrected for very simply: we simply have to use the full nonlinearly backreacted LLM solution (rather than the linearized version we have developed in this paper). When the DDBH has more than one dual (e.g. in the rank 3 case) accounting for this backreaction exactly is a bit harder, one would first have to find the nonlinear solution corresponding to 3 duals with charges in different $SO(6)$ directions in pure AdS (e.g. using analysis of [38]). Irrespective of the details, the supersymmetry of these solutions guarantees that they do not modify the thermodynamics of the non interacting model when $\epsilon^{\mathrm{irred}} = 0$.

### 6.8.1 Estimate of reducible errors

Reducible errors are those that arise from the fact that we have dealt with the branes in DDBH solutions in the linearized approximation. In the solutions of this section, we have retained the linearized backreaction of branes on the background metric; this is of order $\mathcal{O}(1/N)$. The first nonlinearity - which we have ignored - is of order $\mathcal{O}(1/N^2)$. This gives us the estimate

$$\epsilon^{\mathrm{red}} = \frac{1}{N^2} \, , \tag{257}$$

for the reducible errors in our solutions.

---

[127]Except, perhaps, in a supersymmetric context. See §7 for relevant discussion.

[128]This follows because the multi dual configuration that appears in DDBH solutions are configurations which (in pure $AdS$) preserve a common 4 ($1/8^{th}$ of total) supersymmetries, and so exactly obey $E = Q_1 + Q_2 + Q_3$.

### 6.8.2 Estimate of irreducible errors

The second source of error - one that is more irreducible - arises from interference between the black hole and the D-brane probe. This correction may be estimated as follows. Let $\delta_{BH}(r)$ represent an estimate of the fractional deviation of the black hole metric away from the metric of $AdS_5 \times S^5$. Let $\delta_{Brane}(r)$ represent an estimate of the metric correction (210) (or field strength correction (216)) in units of the background $AdS_5 \times S^5$ metric (or background uniform field strength). Very roughly $\delta_{BH}(r)$ is of order unity when $r$ is of order unity, but is of order $\frac{1}{r^4}$ at larger values of $r$. On the other hand, $\delta_{Brane}(r)$ is of order $\frac{1}{N}$ near the brane, of order $\frac{r^4}{Nd^4}$ when $1 \ll r \ll d$, and of order $\frac{1}{d^4}$ when $r$ is of order unity.

We expect that the solution described §6.7 will receive fractional corrections of order $\delta_{BH}(r)\delta_{Brane}(r)$. This product is of order $\frac{1}{Nd^4}$ for all $r$ less than or of order $d$.

In this paper we are most interested values of $d$ of order $\sqrt{N}$. At these values, $\delta_{BH}(r)\delta_{Brane}(r)$ evaluates to a number of order $\frac{1}{N^3}$, yielding the estimate

$$\epsilon^{\text{irred}} = \frac{1}{N^3}, \tag{258}$$

for the irreducible errors to the solution presented in this paper.

## 7 Discussion

In this paper we have argued that the end point of super radiant instability of $\mathcal{N} = 4$ SYM theory is a Dual Dressed Black Hole (DDBH). DDBHs consist of a central black hole surrounded by one, two or three very large dual giant gravitons. The duals have radius of order $\sqrt{N}$, and so are effectively non interacting with the central black hole. The backreaction to these dual giants is significant only at distances of order $\sqrt{N}$ away from the probe brane where it is effectively linear and easily determined. For this reason, the bulk solutions for DDBHs are easy to construct approximately (with errors governed by inverse powers of $N$) and are presented in §6.

The entropically dominant DDBH solution hosts a few (up to 3) dual giants, each of which lives very far from the central black hole. At the level of classical supergravity, there presumably also exist solutions corresponding describing several - let us say $N\zeta$ - coincident duals, located at the radial coordinate $r \approx \frac{\sqrt{h}}{\sqrt{\zeta}}$. at least for $\zeta \ll 1$ such solutions[129] (whose duals giants carry total charge $\approx (N\zeta) \times \frac{Nh}{\zeta} \sim N^2 h$, independent of $\zeta$), can be thought of as a one parameter set of solutions, describing dual giants carrying charge $N^2 h$ surrounding the 'same' central black hole. The solutions of §6 of this paper can be thought of as one extreme end of this line of solutions, with $\zeta$ taking the smallest allowed value $\zeta = \frac{1}{N}$. It should be possible to generalize the gravitational solutions of §6 to larger values of $\zeta$ at for $\zeta \ll 1$.[130] This would be an interesting and informative exercise.

At finite values of $\zeta$, an explicit analytic construction of these solutions may be most feasible in supersymmetric situations. As we have explained in §4.7, Gutowski-Reall black holes

---

[129]In the neighbourhood of the branes, these solutions should resemble the corresponding nonlinear LLM solutions (one sourced by a finite sized blob around $A$ in the $y = 0$ plane depicted in Fig 16), and so should be completely smooth - and presumably horizon free - solutions of supergravity without the need for a D3 brane source. At radial coordinates of order unity - much smaller than $\frac{1}{\sqrt{\zeta}}$ when $\zeta$ is small - they would resemble the usual black hole solutions of §2. The interactions between these components should scale like $\frac{1}{\zeta^3}$.

[130]In this regime of parameters, it should be possible to construct these solutions via a matched asymptotic expansion (using the fact that the solutions resemble known nonlinear LLM solutions at large $r$, and known black holes at $r$ of order unity).

admit exactly supersymmetric probe dual solutions. It thus seems very likely that the 'ζ' generalizations of these solutions will also be 1/16 BPS. The exact construction of the relevant SUSY black holes (perhaps using a generalization of the analysis of [38]) would be a fascinating exercise, one that could provide an explicit nonlinear construction of a 5 parameter family of SUSY black holes, settling a long standing question about the existence of such black holes.[131]

The constructions presented in this paper lead to a natural quantitative conjecture for the phase diagram of $\mathcal{N} = 4$ Yang Mills theory in the microcanonical ensemble. In §3 we have described the resultant phase diagram in detail in two cases; first when $J_1 = J_2 = 0$, and second when $Q_1 = Q_2 = Q_3 = Q$ and $J_1 = J_2 = J$, always ignoring Gregory Laflamme phase transitions and the resultant 10d black hole phases. It would be useful to generalize the analysis of §3 to account for 10d black hole phases, and also to account for the full 6 parameter charge space of $\mathcal{N} = 4$ Yang Mills theory.

The phase presented in Fig 4 has an interesting feature. It predicts a nonzero entropy all the way down to the BPS plane,[132] and so gives a quantitative prediction for SUSY cohomology in $\mathcal{N} = 4$ Yang Mills theory at large $N$ (at least in the large $\lambda$, but plausible also at every finite value of $\lambda$). There has been recent progress in identifying black hole cohomologies [39–45]. It would be interesting to find the black hole cohomologies corresponding to this supersymmetric solution. See the upcoming paper [22] for details.

In this paper we have constructed DDBH solutions within the supergravity approximation, i.e. in the dual to $\mathcal{N} = 4$ Yang Mills theory at very strong coupling. As the key element that went into the construction of DDBHs (namely separation of components in the radial direction) are robust to small corrections, it seems very likely that such phases will continue to exist, and to dominate the ensemble, at large but finite $\lambda$.[133] It is even possible that DDBH phases continue to exist all the way down to weak coupling. The authors of [47] demonstrated that dual giant gravitons can be thought of as arising from the quantization of the 'Coulomb Branch' of $\mathcal{N} = 4$ Yang Mills (these configurations are viewed as configurations on $S^3$, and so quantized in radial quantization). Since $\mathcal{N} = 4$ Yang Mills theory has a Coulomb branch at all values of the coupling, dual giants should continue to exist even at weak coupling. From this point of view, moreover, dual giants at large charge are associated with configurations in which the scalar fields of $\mathcal{N} = 4$ Yang Mills theory develop a very large eigenvalue (see [47, 48]). This large 'vev' plausibly leads to decoupling between this eigenvalue and the remaining theory even at weak coupling.[134] It may thus be practically possible to construct these phases at weak coupling (this may require the summation of infinite sequences of self energy type diagrams). Such an investigation would be fascinating, and could give an independent field theory window onto DDBHs and their role in the phase diagram of $\mathcal{N} = 4$ Yang Mills.

In this paper we have focussed on the study of a single theory, namely the bulk dual to $\mathcal{N} = 4$ Yang Mills theory. However, it seems very likely that similar phases exist in the bulk dual to, for instance, the theory of $M_2$ and $M_5$ branes, and perhaps (more generally) any theory with a nontrivial Coulomb branch [47]. It would also be interesting to investigate this point further.

In §5, we have explicitly demonstrated that black holes with $\mu > 1$ are dynamically unstable to the emission of large dual giant gravitons. However, the time scale for the direct

---

[131]The construction of this paper has already established the existence of such configurations at the probe level, see below for further discussion.

[132]In particular, the supersymmetric phase to the right of the black dot in Fig. 4 consists of a Gutowski-Reall black hole dressed by a SUSY probe dual giant. The final solution carries a nonzero entropy (that of the Gutowski-Reall black hole at its centre.

[133]See [46] for a study of $\alpha'$ corrections to the pure Gutowski-Reall black hole.

[134]In the strongly coupled theory this decoupling is a consequence of the large separation between these two components in the radial direction.

tunneling of such a black hole to its final DDBH is of order $e^{N^2}$ (see §5). It seems almost certain that the passage of unstable black holes with $\mu > 1$ to the end state DDBH, will not be direct, but will, instead proceed via intermediate configurations. One class of such intermediate configurations consists of several dual giants of a charges of order $N$ rather than a single dual giant carrying a charge of order $N^2$. For example, we have seen above (see the discussion around (203)) that black holes with chemical potentials $\mu = 1 + \delta$ are unstable to the emission of duals with charges $Q$ when

$$Q > N q_c(\delta). \tag{259}$$

As the time scale for the decay into duals of charge $Q$ is of order $e^Q$, we see that duals of charge just above $N q_c(\delta)$ will be the first to be created. If $\delta$ is positive and of order unity, $q_c(\delta)$ is also of order unity. Consequently, the amplitude for such a black hole to decay into duals with charge just greater than $N q_c$, is of order $e^{-N q_c(\delta)}$ (note this is much larger than $e^{-N^2}$). At the very least, therefore, we expect the decay of a black hole with $\mu > 1$ into DDBHs, to proceed via the production of several duals of charge of order $N$, and their subsequent merger (perhaps via $D3$ brane 'pants diagrams') leading to a genuine DDBH.

While a time scale of order $e^N$ is much smaller than $e^{N^2}$, it is still very long. Although we do not have a clear expectation here, it is also possible that the decay process involves even more rapidly formed intermediate states involving Kaluza Klein gravitons. As motivation to consider this possibility, recall that the authors of the paper [8] encountered a similar question when studying the end point of black holes in $AdS$ space that suffered from rotational superradiant instabilities. In that situation, the mode of a scalar field (dual to an operator of dimension $\Delta$) with angular momentum $l$, was superradiant unstable when $\omega > \frac{\Delta + l}{l}$. On the other hand, the decay rate into a mode of angular momentum $l$ was of order $e^{-al}$. Consequently, while the modes that first went unstable (and dominated the ensemble) have very large values of $l$, the time scale for the formation of these modes was enormous. The authors of [8] proposed that superradiant unstable black holes, with any given value of $\omega > 1$, first decay into the mode with the smallest value of $l$ that happens to be unstable (because the time scale for this decay is relatively short) and only gradually spread into modes of larger $l$, approaching the final equilibrium configurations at very large time. It is natural to wonder whether a similar scenario plays out for with the operators $\text{Tr}(Z^n)$ (with $n$ of order unity, but increasingly large) playing the role of operators with large angular momentum (in the discussion above).[135] One way to check this possibility would be to linearize the equations of IIB supergravity around black hole backgrounds (a formidable, but doable computation), and then employ the WKB analysis of [49] on the resulting equations.[136]

Finally, we remind the reader that the DDBH phase, constructed in this paper, replaces the Gubser type 'hairy black hole' phase (which we have shown to be unstable). Hairy black hole phases have been studied over 15 years, and their dynamics was carefully investigated. In particular, it has been demonstrated (see e.g. [51–53]) that the long distance dynamics of hairy black holes is governed by the equations of two fluid superfluid hydrodynamics. Now hydrodynamics usually emerges in a large charge and energy limit in which a black hole can,

---

[135] As modes with large $n$ do not physically separate from the black hole (as was the case, both for modes of large $l$ in [8], as well as for the dual giants of large charge studied in this paper), it is not clear that this will happen. Rough estimates (performed modeling the dual to $\text{Tr}(Z^n)$ as a minimally coupled scalar with mass and charge equal to $n$) suggest that it does not happen. It is nonetheless interesting to investigate this possibility more definitively.

[136] Such computations would allow one to compute $\mu_n$, the chemical potential at which the $n^{th}$ mode first goes unstable. If it turns out that $\mu_n$ decreases with $n$, approaching unity from above, then that would suggest that charged instabilities undergo a cascade to higher Kalutza Klein number, similar to the cascade of their rotating counterparts to higher angular momentum. We have performed a back of the envelope version of such a computation, modeling the $n^{th}$ mode as a charged minimally coupled scalar with $m = Q = n$. Our rough estimate suggests that this such a minimally coupled scalar does not go unstable at large $n$ and $\mu$ of order unity. It is, however, possible that nonminimal couplings play an important role in this analysis (as in the recent paper [50] and change this conclusion).

patch wise, be well approximated by a black brane [54]. DDBH phases behave rather weirdly in this limit. As we have explained in §3.6, they separate into two parts; an uncharged black brane which contains all the energy, and a charged dual giant (D brane) which contains all the charge. In the strict scaling limit, the charged brane moves into the deep UV; infinitely separated from the black hole (this appears to be the case even at finite $N$; equivalently even for finite $\zeta$ analogues of DDBH solutions). Consequently, long distance dynamics in the black brane scaling limit of §3.6 is is simply governed by the usual equations of uncharged fluid dynamics (see [55]) together with fully decoupled dynamics D3 branes located at $r = \infty$. The DDBHs phase thus seems to be an unfamiliar new phase of matter - one in which the gauge group is effectively Higgsed down to $U(N-1)$ or $U(N-2)$ or $U(N-3)$, the Higgs vev goes to infinity (resulting of absolute decoupling of the $U(1)$ factors from the non abeliean dynamics). The unbroken non abelian sector carries all the energy, while the $U(1)$ sectors carry all the charge. It would certainly be interesting to understand this better from a field theory viewpoint (see [56–58] for related work).

## Acknowledgments

We would like to thank A. Arabi Ardehali, K. Bajaj, F. Benini, N. Bobev, A. Grassi, W. A. Hamdan, G. Horowitz, V. Kumar, S. Kundu, F. Larsen, G. Mandal, D. Marolf, P. Mitra, J. Mukherjee, S. Murthy, K. Papadodimas, O. Parrikar, A. Rahaman, K. Sharma, A. Strominger, S. Trivedi, A. Zhiboedov and especially A. Gadde for very useful discussions. We would also like to thank O. Aharony, S. Hartnoll, O. Henriksoon, G. Horowitz, S. Kundu, J. Minahan, S. Murthy, H. Ooguri, H. Reall, J. Santos and A. Sen for comments on the manuscript. S.M. and C.P would like to thank ICTP Trieste for hospitality while this work was being carried out.

**Funding information**    The work of S.C. was supported by a KIAS Individual Grant PG081602 at Korea Institute for Advanced Study, and World Premier International Research Center Initiative (WPI), MEXT, Japan. The work of D.J., S.M., C.P. and V.K. was supported by the J C Bose Fellowship JCB/2019/000052, and the Infosys Endowment for the study of the Quantum Structure of Spacetime. The work of S.K. and E.L. was supported in part by the National Research Foundation of Korea (NRF) Grant 2021R1A2C2012350. V.K. was supported in part by the U.S. Department of Energy under grant DE-SC0007859. V.K. would like to thank the Leinweber Center for Theoretical Physics for support. The work of E.L. was supported by Basic Science Research Program through the National Research Foundation of Korea (NRF) funded by the Ministry of Education RS-2024-00405516. C.P was supported in part by grant NSF PHY-2309135 to the Kavli Institute for Theoretical Physics (KITP). D.J, V.K, S.M, and C.P would all also like to acknowledge our debt to the people of India for their steady support for the study of the basic sciences.

## A    Notations and conventions

The Bosonic part of the action for Type IIB Supergravity takes the form[137]

$$S = \frac{1}{16\pi(2\pi^2)^3 g_s^2 \alpha'^4} \sqrt{-g} \left( R - \frac{1}{2(5!)} F_5^2 - \frac{H^2}{2(3!)} - \partial_\mu \phi \partial^\mu \phi \right), \tag{A.2}$$

---

[137]We will sometimes use the notation

$$l_p^8 = g_s^2 \alpha'^4. \tag{A.1}$$

where $F_5$ is the RR five-form field whose components in terms of four-form potential can be written as:[138]

$$(F_5)_{\mu_1\mu_2\mu_3\mu_4\mu_5} = \partial_{\mu_1} C_{\mu_2\mu_3\mu_4\mu_5} + \partial_{\mu_2} C_{\mu_3\mu_4\mu_5\mu_1} + 3 \text{ other.} \tag{A.4}$$

We work in the normalization for the five-form field such that

$$\int_{S^5} *F_5 = (2\pi)^4 N g_s \alpha'^2 \,, \tag{A.5}$$

where the integral is taken on an $S^5$ surrounding $N$ D3-branes.

The probe action for a single D3-brane is given by

$$S = \frac{1}{(2\pi)^3\alpha'^2 g_s}\left(-\int d^4 y \sqrt{-\det h_{ab}} + \frac{1}{4!}\int \epsilon^{abcd} C_{\mu_1\mu_2\mu_3\mu_4} \frac{dx^{\mu_1}}{dy^a}\frac{dx^{\mu_2}}{dy^b}\frac{dx^{\mu_3}}{dy^c}\frac{dx^{\mu_4}}{dy^d} d^4 y\right). \tag{A.6}$$

If we place $N$ branes at the origin of the transverse $R^6$, the equation of motion that follows from the variation of (A.2) and (A.6) is

$$\frac{1}{16\pi(2\pi^2)^3 g_s^2\alpha'^4}\nabla_\mu F^{\mu\mu_1\mu_2\mu_3\mu_4} = -\frac{N}{(2\pi)^3\alpha'^2 g_s}\delta^6(\vec{r})\epsilon^{\mu_1...\mu_4}$$

$$\implies \frac{1}{16\pi^4 g_s\alpha'^2}\nabla_\mu F^{\mu\mu_1\mu_2\mu_3\mu_4} = -N\delta^6(\vec{r})\epsilon^{\mu_1...\mu_4}\,. \tag{A.7}$$

The solution to this equation is given as follows. Let us set

$$F_{\mu_1\mu_2\mu_3\mu_4 r} = A(r)\epsilon_{\mu_1\mu_2\mu_3\mu_4 r}\,, \tag{A.8}$$

where $\mu_1\ldots\mu_4$ are Cartesian coordinates on the brane world volume.

The equation of motion (A.7) then takes the following form

$$\frac{1}{(2\pi)^4 g_s\alpha'^2}\frac{1}{r^5}\partial_r\left(r^5 F^{r\mu_1\mu_2\mu_3\mu_4}\right) = -N\frac{\delta(r)}{\pi^3 r^5}\epsilon^{\mu_1...\mu_4}\,, \tag{A.9}$$

where $\pi^3$ is the volume of the unit $S^5$. It therefore follows that, $A(r) = \frac{A}{r^5}$. To determine the constant $A$, we multiply (A.9) by $r^5$ and then integrate. Now. comparing the integral of the field strength on the $S^5$ surrounding the brane with (A.5), we find

$$A = -N\frac{(2\pi)^4 g_s\alpha'^2}{\pi^3}\,, \qquad F = -A\frac{\epsilon_{\mu_1\mu_2\mu_3\mu_4 r}}{r^5}\,. \tag{A.10}$$

It is easy to check that (A.10) obeys (A.5) with $N = 1$, as we expect (the factors of $\pi^3$ cancel with a factor of $\sqrt{g_5}$, the unit metric on the sphere, when evaluating (A.5)).

The backreaction of a probe D3-brane on the metric may also be computed as follows. Let us suppose that our brane spans an $R^{3,1}$ with metric $\eta^{\mu\nu}$, but sits at the origin of the transverse $R^4$. The stress tensor of the brane (that follows by varying the action (A.6) w.r.t. the background metric: see (A.15)) is given by

$$T^{\mu\nu} = -\frac{1}{(2\pi)^3\alpha'^2 g_s}\eta^{\mu\nu}\delta^6(\vec{r})\,. \tag{A.11}$$

---

[138]We note for future use that the bulk stress tensor associated to second term in action (A.2) can be obtained as follows:

$$\begin{aligned} T_F^{\mu\nu} &= \frac{2}{\sqrt{-g}}\frac{\delta S_F}{\delta g_{\mu\nu}} \\ &= \frac{-1}{16\pi(2\pi^2)^3 g_s^2\alpha'^4 5!}\left(\frac{1}{2}g^{\mu\nu}F_5^2 - 5F^{\mu\nu_1\nu_2\nu_3\nu_4}F^\nu{}_{\nu_1\nu_2\nu_3\nu_4}\right). \end{aligned} \tag{A.3}$$

In a convenient choice of coordinates, our back reacted metric takes the form

$$ds_{(1)}^2 = \phi(r)\big(dx^\mu dx_\mu - dy^i dy_i\big) \tag{A.12}$$

(here $i = 1\ldots6$ and $y_i y_i = r^2$).

It is now easy to see that the linearized Einstein's equations around flat space are satisfied if the field $\phi(r)$ satisfies

$$4\partial_i\partial^i\phi = 8\pi G_N \frac{1}{(2\pi)^3\alpha'^2 g_s}\delta^6(\vec{r}). \tag{A.13}$$

Solving the equation above we get the following expression for $\phi$:

$$\phi = \frac{2\pi\alpha'^2 g_s}{r^4}. \tag{A.14}$$

Note that since we have solved the linearized equations, this approximation is no longer valid when $F^2$ is of order $R$, i.e. when $g_s^2(\alpha')^4/r^8$ is of order unity, i.e. when $r = r_0 \sim \sqrt{\alpha}(g_s N)^{\frac{1}{4}}$.

We can also understand that $N$ D-branes carry tension of order $\frac{N}{g_s}$ as follows. The energy contained in the 5-form field strength is of order

$$\frac{1}{\alpha'^4 g_s^2}\int F^2 \sim N^2 \int \frac{dr\, r^5}{r^{10}} \sim N^2 \frac{1}{r_0^4} \sim \frac{N}{g_s\alpha'^2}.$$

This is of the same order as $N$ times the tension of a single D-brane (see the coefficient of the first term in (A.6)).

## A.1 Stress tensor

The probe brane described in (A.6) carries a stress tensor given by

$$\begin{aligned}
T_D^{\mu\nu}(x) &= \frac{2}{\sqrt{-g}}\frac{\delta S}{\delta g_{\mu\nu}} \\
&= \frac{-1}{(2\pi)^3\alpha'^2 g_s}\int d^4y\,\sqrt{-h}h^{ab}\frac{\partial x_0^\mu(y)}{\partial y^a}\frac{\partial x_0^\nu(y)}{\partial y^b}\frac{\delta^{10}(x^\mu - x_0^\mu(y))}{\sqrt{-g(x)}},
\end{aligned} \tag{A.15}$$

where we have used the formula $\delta h_{ab} = \partial_a X^\mu\partial_b X^\nu\delta g_{\mu\nu}$, and the $\delta$ function is normalized so that $\int\delta = 1$: note that $\frac{\delta^{10}}{\sqrt{-g}}$ is the covariantly normalized $\delta$ function.

The stress tensor is tangent to the world volume of the brane. This is intuitively clear from the expression (A.15), and can be seen more carefully as follows. Let $n_\mu$ is a vector normal to the worldvolume of the brane. Then it can be written as

$$n_\mu = k(x)\frac{\partial f(x)}{\partial x^\mu}, \tag{A.16}$$

where $f$ is a function which is a function which is constant on the worldvolume of the brane and $k(x)$ is an arbitrary function.[139] The contraction of the stress tensor with such a vector becomes

$$\begin{aligned}
n_\mu T_D^{\mu\nu} &= \frac{-1}{(2\pi)^3\alpha'^2 g_s}\int d^4y\,\sqrt{-h}h^{\alpha\beta}k(x)\frac{\partial f(x)}{\partial x^\mu}\frac{\partial x^\mu}{\partial y^\alpha}\frac{\partial x^\nu}{\partial y^\beta}\frac{\delta^{10}(x^\mu - x_0^\mu(y))}{\sqrt{-g}} \\
&= \frac{-1}{(2\pi)^3\alpha'^2 g_s}\int d^4y\,\sqrt{-h}h^{\alpha\beta}k(x)\frac{\partial f(x(y))}{\partial y^\alpha}\frac{\partial x^\nu}{\partial y^\beta}\frac{\delta^{10}(x^\mu - x_0^\mu(y))}{\sqrt{-g}} = 0,
\end{aligned} \tag{A.17}$$

where in the last line we have used the fact that $f$ is constant on the worldvolume. Hence we have shown that the stress tensor is completely tangent to the worldvolume.

---

[139]There is a six dimensional space of such functions.

## A.2  Equation of motion

The equation of motion of the brane can be obtained by taking the variation of the action (A.6) with respect to the brane embedding $x_0^\mu(y)$. We put $F_{ab} = 0$. Then, the variation of the first term gives

$$
\begin{aligned}
\frac{-\delta \int d^4y \sqrt{-h}}{\delta x_0^\alpha(y')} &= -\int d^4y \frac{\delta \sqrt{-h}}{\delta h_{ab}} \frac{\delta h_{ab}}{\delta x_0^\alpha(y')} \\
&= -\frac{1}{2} \int d^4y \sqrt{-h} h^{ab} \frac{\delta \left( \frac{\partial x_0^\mu}{\partial y^a} \frac{\partial x_0^\nu}{\partial y^b} g_{\mu\nu} \right)}{\delta x_o^\alpha(y')} \\
&= -\frac{\sqrt{-h} h^{ab}}{2} \frac{\partial x_0^\mu}{\partial y^a} \frac{\partial x_0^\nu}{\partial y^b} \frac{\partial g_{\mu\nu}}{\partial x_0^\alpha} + \frac{\partial}{\partial y^b} \left( \sqrt{-h} h^{ab} g_{\alpha\nu} \frac{\partial x_0^\nu}{\partial y^a} \right) \\
&= -\frac{\sqrt{-h} h^{ab}}{2} \frac{\partial x_0^\mu}{\partial y^a} \frac{\partial x_0^\nu}{\partial y^b} \frac{\partial g_{\mu\nu}}{\partial x_0^\alpha} + \sqrt{-h} h^{ab} \left( \hat\nabla_b g_{\alpha\nu} \right) \left( \frac{\partial x_0^\nu}{\partial y^a} \right) \\
&\quad + \sqrt{-h} h^{ab} \left( g_{\alpha\nu} \right) \hat\nabla_b \left( \frac{\partial x_0^\nu}{\partial y^a} \right) \\
&= -\frac{\sqrt{-h} h^{ab}}{2} \frac{\partial x_0^\mu}{\partial y^a} \frac{\partial x_0^\nu}{\partial y^b} \frac{\partial g_{\mu\nu}}{\partial x_0^\alpha} + \sqrt{-h} h^{ab} \frac{\partial g_{\alpha\nu}}{\partial x_0^\mu} \frac{\partial x_0^\mu}{\partial y^b} \frac{\partial x_0^\nu}{\partial y^a} \\
&\quad + \sqrt{-h} h^{ab} g_{\alpha\nu} \hat\nabla_b \left( \frac{\partial x_0^\nu}{\partial y^a} \right) \\
&= \sqrt{-h} h^{ab} g_{\alpha\nu} \left( \hat\nabla_b \left( \frac{\partial x_0^\nu}{\partial y^a} \right) + \frac{\partial x_0^\mu}{\partial y^a} \frac{\partial x_0^\beta}{\partial y^b} \Gamma^\nu_{\mu\beta} \right).
\end{aligned} \tag{A.18}
$$

In going from the second to the third line we have both differentiated $g_{\mu\nu}$ (giving the first term on the third line) and varied the partial derivatives of $x$ w.r.t. $y$: this gives the second term on the RHS of the third line after an integration by parts). In going from the third to the fourth line we have used the formula $\partial_a \left( \sqrt{-h} h^{ab} A_b \right) = \sqrt{-h} \nabla_b A^b$, where $A^b$ is any vector field on the submanifold. In going from the fourth to the fifth line we have used the usual formula for the spacetime Christoffel connection).

The variation of the second term gives

$$
\begin{aligned}
\frac{1}{4!} \frac{\delta}{\delta x_0^\alpha(y')} & \int d^4y \, \epsilon^{a_1 a_2 a_3 a_4} C_{\mu_1 \mu_2 \mu_3 \mu_4} \frac{dx_0^{\mu_1}}{dy^{a_1}} \frac{dx_0^{\mu_2}}{dy^{a_2}} \frac{dx_0^{\mu_3}}{dy^{a_3}} \frac{dx_0^{\mu_4}}{dy^{a_4}} \\
&= \frac{1}{4!} \epsilon^{a_1 a_2 a_3 a_4} \left( \frac{\partial C_{\mu_1 \mu_2 \mu_3 \mu_4}}{\partial x_0^\alpha} \frac{dx_0^{\mu_1}}{dy^{a_1}} \frac{dx_0^{\mu_2}}{dy^{a_2}} \frac{dx_0^{\mu_3}}{dy^{a_3}} \frac{dx_0^{\mu_4}}{dy^{a_4}} - 4 \frac{\partial}{\partial y_{a_1}} \left( C_{\alpha \mu_2 \mu_3 \mu_4} \frac{dx_0^{\mu_2}}{dy^{a_2}} \frac{dx_0^{\mu_3}}{dy^{a_3}} \frac{dx_0^{\mu_4}}{dy^{a_4}} \right) \right) \\
&= \frac{1}{4!} \epsilon^{a_1 a_2 a_3 a_4} \left( \frac{\partial C_{\mu_1 \mu_2 \mu_3 \mu_4}}{\partial x_0^\alpha} \frac{dx_0^{\mu_1}}{dy^{a_1}} \frac{dx_0^{\mu_2}}{dy^{a_2}} \frac{dx_0^{\mu_3}}{dy^{a_3}} \frac{dx_0^{\mu_4}}{dy^{a_4}} - 4 \frac{\partial}{\partial y_{a_1}} \left( C_{\alpha \mu_2 \mu_3 \mu_4} \frac{dx_0^{\mu_2}}{dy^{a_2}} \frac{dx_0^{\mu_3}}{dy^{a_3}} \frac{dx_0^{\mu_4}}{dy^{a_4}} \right) \right) \\
&= \frac{1}{4!} \epsilon^{a_1 a_2 a_3 a_4} \left( \frac{\partial C_{\mu_1 \mu_2 \mu_3 \mu_4}}{\partial x_0^\alpha} \frac{dx_0^{\mu_1}}{dy^{a_1}} \frac{dx_0^{\mu_2}}{dy^{a_2}} \frac{dx_0^{\mu_3}}{dy^{a_3}} \frac{dx_0^{\mu_4}}{dy^{a_4}} - 4 \frac{\partial C_{\alpha \mu_2 \mu_3 \mu_4}}{\partial x_0^{\mu_1}} \frac{dx_0^{\mu_1}}{dy^{a_1}} \frac{dx_0^{\mu_2}}{dy^{a_2}} \frac{dx_0^{\mu_3}}{dy^{a_3}} \frac{dx_0^{\mu_4}}{dy^{a_4}} \right) \\
&= \frac{1}{4!} \epsilon^{a_1 a_2 a_3 a_4} F_{\alpha \mu_1 \mu_2 \mu_3 \mu_4} \frac{dx_0^{\mu_1}}{dy^{a_1}} \frac{dx_0^{\mu_2}}{dy^{a_2}} \frac{dx_0^{\mu_3}}{dy^{a_3}} \frac{dx_0^{\mu_4}}{dy^{a_4}}.
\end{aligned} \tag{A.19}
$$

Combining (A.18) and (A.19), we get the following equation of motion of the brane:

$$
\sqrt{-h} h^{ab} g_{\alpha\nu} \left( \hat\nabla_b \left( \frac{\partial x_0^\nu}{\partial y^a} \right) + \frac{\partial x_0^\mu}{\partial y^a} \frac{\partial x_0^\beta}{\partial y^b} \Gamma^\nu_{\mu\beta} \right) + \frac{1}{4!} \epsilon^{a_1 a_2 a_3 a_4} F_{\alpha \mu_1 \mu_2 \mu_3 \mu_4} \frac{dx_0^{\mu_1}}{dy^{a_1}} \frac{dx_0^{\mu_2}}{dy^{a_2}} \frac{dx_0^{\mu_3}}{dy^{a_3}} \frac{dx_0^{\mu_4}}{dy^{a_4}} = 0. \tag{A.20}
$$

## A.3 Conservation of stress tensor

We can also derive the brane equation of motion from the conservation of the stress tensor. Let us first apply the covariant derivative on the brane stress tensor given in (A.15).

$$
\begin{aligned}
\nabla_\mu T_D^{\mu\nu}(x) &= \partial_\mu T_D^{\mu\nu}(x) + \Gamma_{\mu\alpha}^\nu T^{\mu\alpha} + \Gamma_{\mu\alpha}^\mu T^{\alpha\nu} \\
&= \frac{-1}{(2\pi)^3\alpha'^2 g_s}\left[\int d^4y\sqrt{-h}h^{ab}\frac{\partial x_0^\mu}{\partial y^a}\frac{\partial x_0^\nu}{\partial y^b}\frac{\partial}{\partial x^\mu}\left(\frac{\delta^{10}(x^\mu-x_0^\mu(y))}{\sqrt{-g(x)}}\right)\right. \\
&\quad\left.+\sqrt{-h}h^{ab}\left(\frac{\partial x_0^\alpha}{\partial y^a}\frac{\partial x_0^\mu}{\partial y^b}\Gamma_{\mu\alpha}^\nu+\frac{\partial x_0^\alpha}{\partial y^a}\frac{\partial x_0^\nu}{\partial y^b}\Gamma_{\mu\alpha}^\mu\right)\left(\frac{\delta^{10}(x^\mu-x_0^\mu(y))}{\sqrt{-g(x)}}\right)\right] \\
&= \frac{-1}{(2\pi)^3\alpha'^2 g_s}\left[\int d^4y\frac{1}{\sqrt{-g(x)}}\frac{\partial}{\partial y^a}\left(\sqrt{-h}h^{ab}\frac{\partial x_0^\nu}{\partial y^b}\right)\delta^{10}(x^\mu-x_0^\mu(y))\right. \\
&\quad+\sqrt{-h}h^{ab}\delta^{10}(x^\mu-x_0^\mu(y))\left(\frac{\partial x_0^\mu}{\partial y^a}\frac{\partial x_0^\nu}{\partial y^b}\frac{\partial}{\partial x^\mu}\left(\frac{1}{\sqrt{-g(x)}}\right)\right. \\
&\quad\left.\left.+\left(\frac{\partial x_0^\alpha}{\partial y^a}\frac{\partial x_0^\mu}{\partial y^b}\Gamma_{\mu\alpha}^\nu+\frac{\partial x_0^\alpha}{\partial y^a}\frac{\partial x_0^\nu}{\partial y^b}\Gamma_{\mu\alpha}^\mu\right)\right)\right] \\
&= \frac{-1}{(2\pi)^3\alpha'^2 g_s}\frac{1}{\sqrt{-g(x)}} \\
&\quad\times\left[\int d^4y\,\delta^{10}(x^\mu-x_0^\mu(y))\sqrt{-h(y)}h^{ab}\left(\hat\nabla_a\frac{\partial x_0^\nu}{\partial y^b}+\frac{\partial x_0^\beta}{\partial y^a}\frac{\partial x_0^\mu}{\partial y^b}\Gamma_{\mu\beta}^\nu\right)\right],
\end{aligned}
\tag{A.21}
$$

where, in going from the second to the third line we interchanged the derivative of the delta function with respect to $x^\mu$ to a derivative with respect to $x_0^\mu$ with an additional negative sign and then used the chain rule to combine with $\frac{\partial x_0^\mu}{\partial y^a}$ to get $-\frac{\partial}{\partial y^a}(\delta^{10}(x^\mu-x_0^\mu(y)))$. Further, we performed an integration by parts with respect to $y^a$ to obtain the third line (the additional negative sign from integration from parts cancels the previous negative sign). In going from third equality to fourth, we have used the fact that

$$
\frac{\partial g}{\partial x^\mu} = 2g\,\Gamma_{\alpha\mu}^\alpha.
$$

Next consider the covariant derivative of the term in the field strength stress tensor which signifies the interaction energy between the brane and the background field. This term is as follows:

$$
T_{int}^{\mu\nu} = \frac{-1}{16\pi(2\pi^2)^3 g_s^2\alpha'^4 5!}\left(g^{\mu\nu}F.\tilde F - 5F^{\mu\nu_1\nu_2\nu_3\nu_4}\tilde F^\nu{}_{\nu_1\nu_2\nu_3\nu_4} - 5\tilde F^{\mu\nu_1\nu_2\nu_3\nu_4}F^\nu{}_{\nu_1\nu_2\nu_3\nu_4}\right),
\tag{A.22}
$$

where $F$ is the background field strength and $\tilde F$ is the field strength due to the brane. Now taking the covariant derivative,

$$
\begin{aligned}
\nabla_\mu T_{int}^{\mu\nu}(x) &= \frac{-1}{16\pi(2\pi^2)^3 g_s^2\alpha'^4 5!}\left(g^{\mu\nu}\partial_\mu(F.\tilde F) - 5F^{\mu\nu_1\nu_2\nu_3\nu_4}\nabla_\mu\tilde F^\nu{}_{\nu_1\nu_2\nu_3\nu_4}\right. \\
&\quad\left.-5\nabla_\mu\tilde F^{\mu\nu_1\nu_2\nu_3\nu_4}F^\nu{}_{\nu_1\nu_2\nu_3\nu_4} - 5\tilde F^{\mu\nu_1\nu_2\nu_3\nu_4}\nabla_\mu F^\nu{}_{\nu_1\nu_2\nu_3\nu_4}\right) \\
&= \frac{1}{16\pi(2\pi^2)^3 g_s^2\alpha'^4 5!}5\nabla_\mu\tilde F^{\mu\nu_1\nu_2\nu_3\nu_4}F^\nu{}_{\nu_1\nu_2\nu_3\nu_4} \\
&= \frac{-1}{(2\pi)^3\alpha'^2 g_s}\frac{1}{4!\sqrt{-g}}\int d^4y\,\delta^{10}(x^\mu-x_0^\mu(y))\epsilon^{a_1a_2a_3a_4}\frac{dx_0^{\mu_1}}{dy^{a_1}}\frac{dx_0^{\mu_2}}{dy^{a_2}}\frac{dx_0^{\mu_3}}{dy^{a_3}}\frac{dx_0^{\mu_4}}{dy^{a_4}}F^\nu{}_{\mu_1\mu_2\mu_3\mu_4}.
\end{aligned}
\tag{A.23}
$$

In going from first line to second, we note that first term cancels with the second and Here we have used the equations of motion of background field $F$ and perturbation $\tilde{F}$,

$$\nabla_\mu F^{\mu \nu_1 \nu_2 \nu_3 \nu_4} = 0, \tag{A.24}$$

$$\frac{1}{16\pi(2\pi^2)^3 g_s^2 \alpha'^4} \nabla_\mu \tilde{F}^{\mu\mu_1\mu_2\mu_3\mu_4} = -\frac{1}{(2\pi)^3 \alpha'^2 g_s} \int d^4 y \frac{\delta^{10}(x^\mu - x_0^\mu(y))}{\sqrt{-g}} \epsilon^{a_1 a_2 a_3 a_4} \frac{dx_0^{\mu_1}}{dy^{a_1}} \frac{dx_0^{\mu_2}}{dy^{a_2}} \frac{dx_0^{\mu_3}}{dy^{a_3}} \frac{dx_0^{\mu_4}}{dy^{a_4}}.$$

Combining (A.21) and (A.23), we get

$$\nabla_\mu \left( T_D^{\mu\nu} + T_F^{\mu\nu} \right) = 0 \tag{A.25}$$

$$\implies \sqrt{-h(y)} h^{ab} \left( \hat{\nabla}_a \frac{\partial x_0^\nu}{\partial y^b} + \frac{\partial x_0^\beta}{\partial y^a} \frac{\partial x_0^\mu}{\partial y^b} \Gamma_{\mu\beta}^\nu \right) + \frac{1}{4!} \epsilon^{a_1 a_2 a_3 a_4} \frac{dx_0^{\mu_1}}{dy^{a_1}} \frac{dx_0^{\mu_2}}{dy^{a_2}} \frac{dx_0^{\mu_3}}{dy^{a_3}} \frac{dx_0^{\mu_4}}{dy^{a_4}} F^\nu{}_{\mu_1\mu_2\mu_3\mu_4} = 0,$$

which is equivalent to (A.20).

# B  Details of the black hole solution

## B.1  The four-form potential for the 5-form flux

In this subsection, we check that the four-form potential in (26) correctly reproduces the five-form field strength in (24).

To show this we use the following results which can all be obtained simply by using the definitions of $C_V, \Theta, J$ in (27) and (7).

$$dC_V = -4\epsilon^{(5)},$$
$$d\left(\Theta \wedge (-*_5 F + A \wedge F)\right) = -2J \wedge *_5 F + 2J \wedge A \wedge F,$$
$$d\left(l_2^2 d(l_3^2) \wedge d\phi_1 \wedge d\phi_2 \wedge d\phi_3\right) = -2J \wedge J \wedge (d\Psi + \Theta) = -4 *_{10} \epsilon^{(5)} - 2J \wedge J \wedge A, \tag{B.1}$$
$$d(-A \wedge (d\Psi + \Theta) \wedge J) = -F \wedge (d\Psi + \Theta) \wedge J + 2A \wedge J \wedge J.$$

It follows from the bulk Maxwell-Chern-Simons equation of motion that the expression $(-*_5 F + A \wedge F)$, that appears in (26), is a closed 3 form, i.e. that

$$d(-*_5 F + A \wedge F) = 0. \tag{B.2}$$

It follows, in particular, that the integral of this 3-form over any 3-manifold that surrounds the black hole world line evaluates to the same number - the electric charge of the black hole - independent of the details of this 3-manifold. On the black hole spacetime (9), it is easily checked that this 3-form evaluates to

$$(-*_5 F + A \wedge F) = \frac{2\pi^2 Q}{4} \sigma_1 \wedge \sigma_2 \wedge \sigma_3. \tag{B.3}$$

Consequently, (26) can be rewritten as

$$\frac{C}{R_{AdS_5}^4} = C_V + \left(\frac{\pi^2 Q}{4}\right) \Theta \wedge \sigma_1 \wedge \sigma_2 \wedge \sigma_3 + l_2^2 d(l_3^2) \wedge d\phi_1 \wedge d\phi_2 \wedge d\phi_3 + A \wedge (d\Psi + \Theta) \wedge J. \tag{B.4}$$

While (26) solves (25), this solution is certainly not unique. The gauge transformation $C \to C + d\chi$ for any $\chi$ (here $\chi$ is any 3 form) gives an equally good solution to this equation. As this addition changes the Lagrangian by a total derivative, it does not modify the equations of motion that follow from this action. However, such a modification can change the

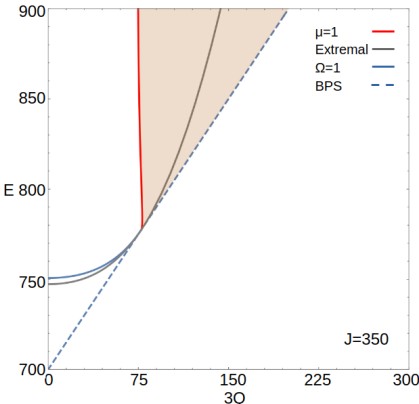

Figure 9: The $\mu = 1$ curve (red) is shown for a fixed $J = 350$ in $E$ $3Q$ plane. The orange region represents the DDBH phase. The slope of the curve is negative, indicating that the black hole can become unstable under the condensation of hair as energy increases. As the energy increases further (beyond what is shown in the figure), the slope of the curve becomes positive, though it remains greater than 1.

value of conjugate momenta - and hence of Noether Charges including the energy- by additive constants. The correct value of Noether Charges (i.e. the value that we find from the field contribution to energy and other charges once we account for backreaction) is obtained only if we choose $C$ so that its integral over the horizon is a vanishing one-form.[140] Our solution (B.4) does not obey this criterion, but is gauge equivalent to a solution of that does meet this criterion. Using

$$\sigma_1 \wedge \sigma_2 \wedge \sigma_3 = d\left(\cos\theta_a d\psi \wedge d\phi\right), \tag{B.5}$$

and so

$$\Theta \wedge \sigma_1 \wedge \sigma_2 \wedge \sigma_3 = -d((\Theta \wedge (\cos\theta_a d\psi \wedge d\phi)) + J \wedge (\cos\theta_a d\psi \wedge d\phi), \tag{B.6}$$

we find a second solution to (25)

$$\frac{C}{R_{AdS_5}^4} = C_V + \left(\frac{\pi^2 Q}{4}\right)\cos\theta_a J \wedge d\psi \wedge d\phi + l_2^2 d(l_3^2) \wedge d\phi_1 \wedge d\phi_2 \wedge d\phi_3 + A \wedge (d\Psi + \Theta) \wedge J, \tag{B.7}$$

which has the property that it integrates to zero on the horizon.

## C $\mu = 1$ curve at large $J$

In this Appendix, we show that when the angular momentum becomes sufficiently large, the $\mu = 1$ curve exhibits interesting behavior, as was not seen for small $J$ (see Fig. 4). Specifically, there is an interval of the $\mu = 1$ line in the $E$ $Q$ plane where the slope of the line becomes negative (see Fig. 9).

---

[140]An analogy might help here. up to a proportionality constant, the interaction energy between two charges in (usual) electromagnetizm $\int \vec{E}_1 . \vec{E}_2 = \int \vec{\nabla}\phi_1 . \vec{\nabla}\phi_2$, where $\vec{E}_1 = -\vec{\nabla}\phi_1$ is the Electric field sourced by the first charge and $\vec{E}_2 = -\vec{\nabla}\phi_2$ is the Electric field sourced by the second charge. Integrating by parts turns this expression into $-\int \phi_2 \vec{\nabla}^2 \phi_1 + \int_B \phi_2 \vec{\nabla}\phi_1$, where the second integral is taken over the boundary of spacetime (at infinity). This is proportional to $q_1 \phi_2$ (where $\phi_2$ is evaluated at the location of the charge $q_1$ if and only if the integral at infinity vanishes, i.e. if $\phi_2$ vanishes at infinity. It follows that the interaction energy between the two charges is correctly given by the contribution to the probe action $q_1 \int \phi_2$ only when $\phi_2$ is chosen to vanish at infinity. The analogue of this condition, in the current situation, is to demand that $C$ vanishes when integrated over the black hole horizon.

We should note that the phase diagram of the DDBH is unaffected since the slope remains greater than 1 as long as it is positive. However, it is intriguing that a black hole can become unstable and transition to the DDBH phase as energy increases. Naively, one would expect the black hole to become more stable as energy increases, since entropy increases when other charges are held constant. At the moment, we don't have any specific explanation to account for this phenomenon. It would be interesting to investigate why this happens.

# D   Phase diagrams for black holes with one or two non-zero charges and $J_1 = J_2 = 0$

In this appendix, we discuss the micro-canonical phase diagrams of $\mathcal{N} = 4$ SYM in the special cases where we only have either one or two of the $Q_i \neq 0$. In section 3.4, we reported the phase diagrams for the generic case where $Q_1, Q_2$ and $Q_3$ are all non-zero (see Figure 1). In what follows, we will be using the same convention as in section 3.4 where we considered the energy direction to be 'vertical'. We will also be referring to the three charges as $Q_i$, $Q_j$ and $Q_k$ where $i, j, k$ can take the values $1, 2, 3$ and $i \neq j \neq k$.

## D.1   Two non-zero charges

In the case when one of the charges (say $Q_k$) is put to zero, the charge space becomes 3-dimensional, parameterized by the two charges $(Q_i, Q_j)$ and the energy $E$. There are three surfaces which are of interest for us,

- the $\mu_i = 1$ surface $S_i$ when $Q_i > Q_j$,

- the $\mu_j = 1$ surface $S_j$ when $Q_j > Q_i$,

- the unitarity plane - $E = Q_i + Q_j$ where, $Q_i$ and $Q_j$ are the non-zero charges.

Let us fix the charges $Q_i$ and $Q_j$ to some non-zero value and construct the phase diagram as we vary energy. Let us also assume that $Q_i > Q_j$. The other case also has the same phase diagram but with $i$ replaced by $j$, i.e. the phase diagram is symmetric about the $Q_i = Q_j$ plane. At large values of energy, the vacuum black hole is the dominant phase, and as we lower the energy we first encounter $\mu_i = 1$ surface $S_1$. As we lower the energy further we transition in to the Rank-2 DDBH phase where the core black hole has $\mu_i = 1$ (lies on $S_i$) and the D-brane carries the rest of the $Q_i$ charge (and energy $E = Q_i$). The $i^{th}$ Rank-2 DDBH phase therefore is bounded above by the surface $S_i$. The surfaces $S_i$ and $S_j$ intersect on the $Q_i = Q_j$ plane on a curve $S_{ij}$, on which the vacuum black holes satisfy $\mu_i = \mu_j = 1$. One can construct a two parameter set of Rank-4 DDBH from every point on $S_{ij}$ by varying the amount of charge carried by the two dual giant gravitons. The Rank-2 DDBH phase is also bounded 'below' by the Rank-4 DDBH. The Rank-4 DDBH phase continues to exist all the way until the energy reaches the unitarity bound. The extent of the Rank-2 DDBH phase depends on the difference between the two charges $Q_i - Q_j$ and goes to zero when the charges become equal $Q_i = Q_j$.

When $Q_i = Q_j$ (and $Q_k = 0$), as we lower the energy starting from the vacuum black hole phase, we directly transition to the Rank-4 DDBH which continues to exist until the unitarity bound. The phase diagram will now have three cases: (i) $Q_i > Q_j$, (ii) $Q_i = Q_j$, and (iii) $Q_i < Q_j$ (see Fig 10). See Fig. 11 for the phase diagram at $Q_3 = 0$ and $\frac{Q_2}{N^2} = 1$, and Fig. 12 for the phase diagram at $Q_3 = 0$ and $\frac{Q_2}{N^2} = 3$ (the phase diagram, presented as a function of $\frac{E}{N^2}$ and $\frac{Q_1}{N^2}$, has a small qualitative difference on these two charge slices, as will be explained in the next subsection).

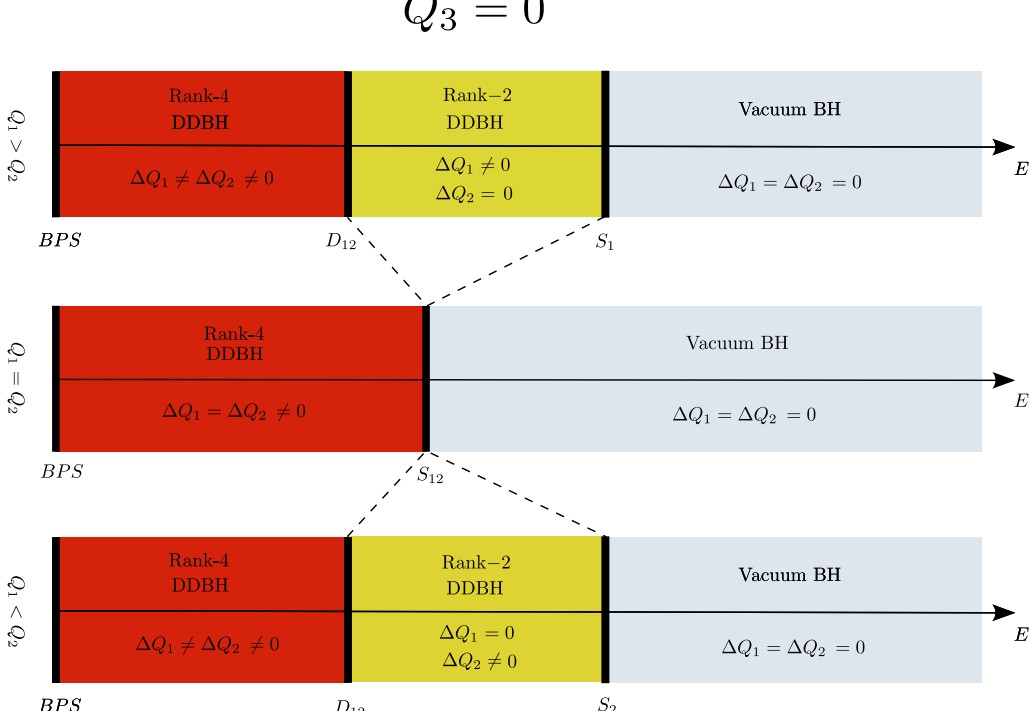

Figure 10: The phase diagram of $\mathcal{N} = 4$ Super Yang Mills theory at vanishing angular momentum and one vanishing charge (say $Q_3$). In the above figure, we depict various phases we encounter as we vary energy at given non-zero charges $Q_1$ and $Q_2$. At high energies (far right of the figure), the vacuum black hole phase dominates for all the three cases. When the non-zero charges $Q_1$ and $Q_2$ are unequal, we encounter both Rank-2 DDBH and Rank-4 DDBH phases as we reduce the energy from the vacuum BH phase. When $Q_1 = Q_2$, the vacuum BH phase transitions directly into Rank-4 DDBH as we lower the energy.

## D.2 One non-zero charge

Now let us turn off another charge $Q_j = 0$ (also $Q_k = 0$) and keep only one $Q_i \neq 0$. The charge space in this case will be two-dimensional - parameterized by the energy $E$ and the charge $Q_i$ (see Fig. 14). The curves of interest in this case are

- the $\mu_i = 1$ curve,

- the unitarity line $E = Q_i$.

Here we find some differences from previous cases. To start with, vacuum black holes exist all the way down to the BPS line. Next, the limiting temperature of black holes (as they approach the BPS line) everywhere equals $\frac{1}{\pi}$ (rather than zero, as one would naively expect) [7]. Finally, (and most important for us), in [7], it was noted that the $\mu_i = 1$ curve in this case does not start from the origin (as was true in the previous cases). The $\mu_i = 1$ curve starts from the point in the charge space $(E^c, Q^c) = (1, 1)$ (see Fig 14). This suggests that the phase diagram when two charges are turned off has the following two cases (see Fig 13),

- $Q_i < Q^c$: As we lower the energy starting from a vacuum black hole phase, we will hit the unitarity bound first and therefore do not undergo any phase transition into a DDBH. In this case we stay in the 'vacuum black hole' phase at all energies.

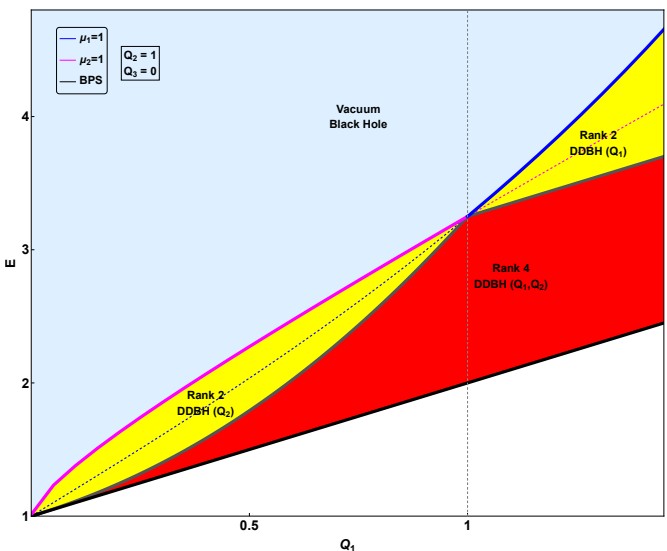

Figure 11: The cross-section of the phase diagram for $\frac{Q_2}{N^2} = 1$ and $\frac{Q_3}{N^2} = 0$. Notice that the value of $Q_2$ is chosen to be $Q^c$ where the Rank 2 DDBH phase carrying $Q_2$ charge stops existing at $Q_1 = 0$. (see Fig 13 and Fig 14).

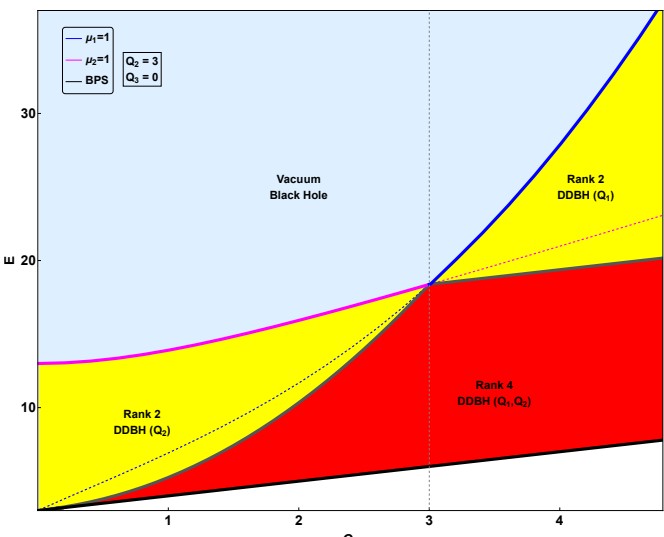

Figure 12: The cross-section of the phase diagram for $\frac{Q_2}{N^2} = 3(> Q^c)$ and $\frac{Q_3}{N^2} = 0$. The Rank 2 DDBH carrying $Q_2$ charge continues to exist at $Q_1 = 0$.

- $Q_i > Q^c$: As we lower the energy starting from a vacuum black hole phase, we will encounter $S_i$ first. Below the $S_i$ curve, the $i^{th}$ Rank-2 DDBH phase dominates the mi-

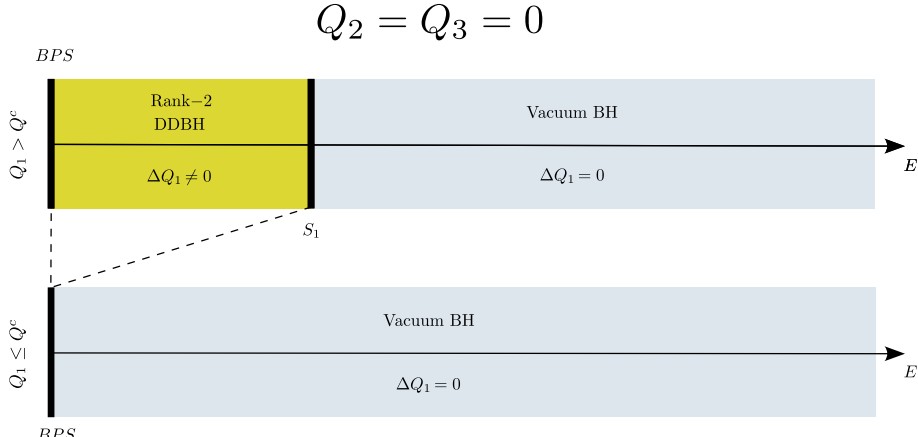

Figure 13: The phase diagram of $\mathcal{N} = 4$ Super Yang Mills with one non-zero charge $Q_1$. Until a critical value $Q_1 = Q^c$, the vacuum BH phase dominates at all energies. When, $Q_1 > Q^c$, the vacuum BH phase transitions to a Rank-2 DDBH phase as we lower energy. See Fig 14 for a 2-dimensional plot of the same phase diagram.

crocanonical ensemble.[141] The upper boundary of the $i^{th}$ Rank-2 phase is therefore the $S_i$ curve (restriction of the three dimensional $S_i$ surface to $Q_j = Q_k = 0$) and the lower boundary is the unitarity plane, where the black hole becomes zero size and the solution reduces to a dual giant graviton in empty $AdS_5 \times S^5$.

## E  The effective particle action for a dual giant graviton

In this Appendix, we demonstrate that the effective action (66) reduces to the action (67) when evaluated on brane configurations that preserve $SU(2)_R \times U(1)_L$. As we have mentioned in the main text, on such configurations, the D3-brane behaves like a point particle propagating in the remaining 6+1 dimensions.

Let us first deal with the DBI part of the action in (66), leaving the Wess-Zumino part for later. The DBI action reduces to the action for a particle propagating in an effective seven dimensional metric. The position dependent effective mass of this particle is given by $\frac{NV_3}{2\pi^2}$. Using $V_3 = 2\pi^2 \left( 2r^2 \sqrt{b^2 + A_3^2} \right)$,[142] we see that this effective mass equals $2Nr^2 \sqrt{b^2 + A_3^2}$. Consequently, the DBI part of (66) reduces to

$$S_{\text{DBI}} = 2N \int r^2 \sqrt{b^2 + A_3^2} \sqrt{-\tilde{g}_{\mu\nu} dx^\mu dx^\nu}, \tag{E.1}$$

where $\tilde{g}_{\mu\nu} dx^\mu dx^\nu$ is the line element on the effective 6+1 dimensional metric that is determined as follows. The 10 dimensional line element given in (1), can be written in the form

$$ds^2 = g_{\mu\nu} dx^\mu dx^\nu + g_{ab} dy^a dy^b + 2g_{a\mu} dy^a dx^\mu, \tag{E.2}$$

where $y^a$ are the coordinates $\theta, \phi, \psi$ that parameterize the three dimensional squashed sphere, and $x^\mu$ the remaining seven coordinates (e.g. $r$, $t$, $\phi_1$, $\phi_2$, $\phi_3$ and $\alpha$ and $\beta$, the

---

[141]This phase has a core black hole with $\mu_i = 1$ and a dual giant graviton carrying only $Q_i$ charge.

[142]This formula is obtained as follows. The restriction of the 10 dimensional metric onto the space spanned by $d\theta$, $d\phi$ and $d\psi$ (equivalently by $\sigma_1, \sigma_2, \sigma_3$) is given by $ds_3^2 = \frac{r^2}{4}(\sigma_1^2 + \sigma_2^2) + (b^2 + A_{\sigma_3}^2)\sigma_3^2)$. Our formula for $V_3$ follows once we recall that the space with metric $\frac{\sigma_1^2 + \sigma_2^2 + \sigma_3^2}{4}$ is the unit 3 sphere with volume $2\pi^2$.

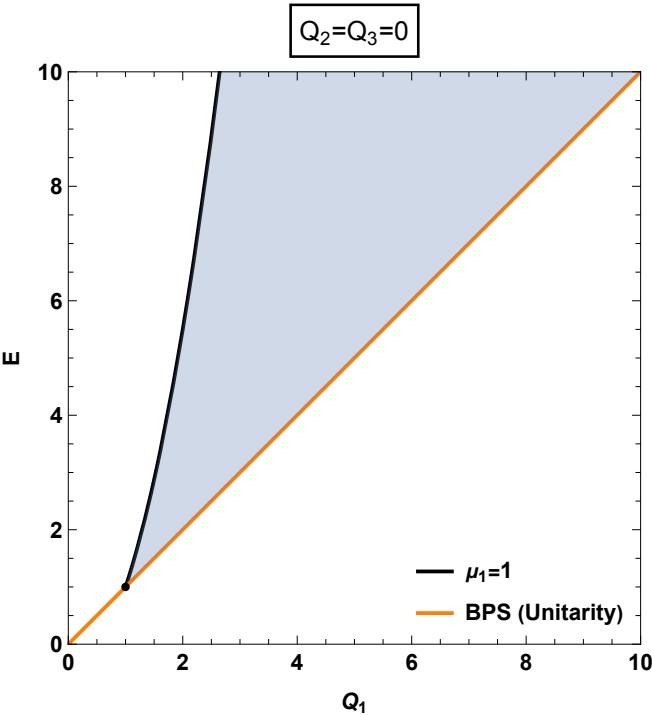

Figure 14: The phase diagram of the $\mathcal{N} = 4$ Super Yang Mills in the special case when $Q_2 = Q_3 = 0$ and $J_1 = J_2 = 0$. The Rank-2 DDBH phase is the dominant phase in the shaded region. It is bounded above by the $\mu_1 = 1$ curve and below by the unitarity bound. Vacuum black holes exist everywhere above the orange line, but are unstable in the shaded blue region. Note that the $\mu_1 = 1$ curve does not meet the BPS curve at origin, but rather they meet at non-zero charge $Q^c = 1$ (in $N^2$ units) [7].

two variables that parameter). Our effective particle propagates on the space $g_{\mu\nu}$ projected orthogonally to the squashed three sphere, i.e. the space with effective metric

$$ds_7^2 = \left(g_{\mu\nu} - g_{\mu a} g_{\nu b} g_3^{ab}\right) dx^\mu dx^\nu = \tilde{g}_{\mu\nu} dx^\mu dx^\nu. \tag{E.3}$$

Explicitly we find

$$
\begin{aligned}
ds_7^2 = \tilde{g}_{\mu\nu} dx^\mu dx^\nu &= \left(-\frac{r^2 W}{4b^2} + A_t^2 + b^2 f^2\right) dt^2 + \frac{dr^2}{W} + d\alpha^2 + \cos^2 \alpha d\beta^2 \\
&+ \sum_{i=1}^{3} \left(l_i^2 d\phi_i^2 - 2A_t l_i^2 d\phi_i dt\right) - \frac{\left(dt(A_t A_{\sigma_3} + b^2 f) - A_{\sigma_3} \sum_i l_i^2 d\phi_i\right)^2}{b^2 + A_{\sigma_3}^2} \\
&= -\frac{r^2 W}{4b^2} dt^2 + \frac{dr^2}{W} + ds^2(\mathbb{CP}^2) + \frac{b^2}{b^2 + A_{\sigma_3}^2} \left(d\Psi + \Theta + (f A_{\sigma_3} - A_t) dt\right)^2,
\end{aligned}
\tag{E.4}
$$

where the gauge fields $A_t$ and $A_{\sigma_3}$ were listed in (9).

The equations (E.1) and (E.4) may be confirmed by evaluating the induced metric on the D-brane and evaluating its volume. We can choose to use $\theta$, $\phi$, $\psi$ and and $\tau$ as world volume coordinates. As the D-brane wraps the squashed $S^3$, the remaining six spacetime coordinates are independent of $\theta$ $\phi$ and $\psi$, but are arbitrary functions of time. In other words the world volume of the D3-brane is parameterized by the 6 functions

$$l_i(\tau), \qquad r(\tau), \qquad \phi_{1,2,3}(\tau). \tag{E.5}$$

The induced metric on the D3-brane is easily computed in terms of these 6 functions: we find

$$ds^2_{\text{ind}} = \left( g_{tt} + g_{rr}\dot{r}^2 + + \sum_{i=1}^{2}(2l_i\dot{l}_i) + \sum_{i=1}^{3}\left(g_{\phi_i\phi_i}\dot{\phi}_i^2 - 2g_{t\phi_i}\dot{\phi}_i\right) \right)dt^2$$
$$+ \left( g_{t\sigma_3} + \sum_{i=1}^{3}g_{\sigma_3\phi_i}\dot{\phi}_i \right)dt\sigma_3 + \sum_{i=1}^{3}g_{\sigma_i\sigma_i}\sigma_i^2 \,. \tag{E.6}$$

In order to deal with the cross term proportional to $dt\sigma_3$ 'complete the square' to obtain

$$ds^2_{\text{ind}} = \Delta dt^2 + (b^2 + A^2_{\sigma_3})\left( \sigma_3 + \frac{\gamma}{b^2 + A^2_{\sigma_3}}dt \right)^2 + \frac{r^2}{4}\left(\sigma_1^2 + \sigma_2^2\right), \tag{E.7}$$

where

$$\Delta = \tilde{g}_{\mu\nu}\frac{dx^\mu}{dt}\frac{dx^\nu}{dt} = -\frac{r^2W}{4b^2} + A_t^2 + b^2f^2 + \frac{\dot{r}^2}{W} + \sum_{i=1}^{2}(2l_i\dot{l}_i) + \sum_{i=1}^{3}\left(l_i^2\dot{\phi}_i^2 - 2A_tl_i^2\dot{\phi}_i\right) - \frac{\gamma^2}{q^2}, \tag{E.8}$$

and

$$q^2 = b^2 + A^2_{\sigma_3}, \qquad \text{and} \qquad \gamma = A_tA_{\sigma_3} + b^2f - l_i^2A_{\sigma_3}\dot{\phi}_i \,. \tag{E.9}$$

Let us now turn to the Wess-Zumino part of the action. The second, third and fourth terms in (26) each have two or more legs on the $S^5$ and so evaluate to zero when integrated over the squashed sphere. Hence we are left with just the first term $C_V$ given by

$$C_V = \left( \frac{r^4 - R^4}{8} \right)dt \wedge \sigma_1 \wedge \sigma_2 \wedge \sigma_3 \tag{E.10}$$

(we have used (B.3) to simplify the second term in (26)).

All the terms in the bracket in (E.10) are independent of $\theta$, $\phi$, and $\psi$, so the integral over these three coordinates can simply be performed. We find

$$S_{WZ} = 8N \int \left( \frac{r^4 - R^4}{8} \right)dt \,. \tag{E.11}$$

In summary, the full probe action for the D3-brane is given by

$$S = S_{DBI} + S_{WZ}$$
$$= 2N \int r^2\sqrt{b^2 + A_3^2}\sqrt{-\tilde{g}_{\mu\nu}dx^\mu dx^\nu} + 8N \int \left[ \left( \frac{r^4 - R^4}{8} \right)dt \right]. \tag{E.12}$$

# F  Constraints on probe motion from $U(3)$ invariance

In this appendix, we write the most general orbits on the squashed $S^5$ which preserves $U(3)$ symmetry along with the general form of the $U(3)$ charge matrix of these orbits.

## F.1  Motion on the 'squashed $S^5$'

After taking $N$ out common overall, (67) may be thought of as the action of a particle of unit mass and 'charge' propagating in the metric $2m\tilde{g}_{\mu\nu}$, in the presence of an effective gauge field $C_t = (r^4 - R^4)$. Note that this effective metric and gauge field both enjoy invariance under $U(3)$ symmetry (since the effective metric only depends on metric on $\mathbb{CP}^2$, $\Theta + d\Psi$ and $J$ which are $U(3)$ invariant as discussed in appendix H.2). In this subsection, we explore the (powerful) constraints that $U(3)$ invariance imposes on the motion of our effective particle.

### F.1.1   Using $SO(6)$ symmetry to constrain particle motion on $S^5$

As a warm up that will prove relevant, let us first recall how $SO(6)$ invariance can be used to characterize particle motion on a round $S^5$. Geodesics on a round sphere traverse great circles, which may be characterized as follows. If we view the $S^5$ as centered about the origin of $R^6$, a great circle is simply the intersection of the $S^5$ with any (flat) two plane in $R^6$ that passes through the origin. Flat two planes that intersect the origin are completely characterized by their two dimensional volume form. Let the coordinates on $R^6$ be $(x_1, y_1, x_2, y_2, x_3, y_3)$(at a later point we will group these coordinates into three complex combinations $z_i = x_i + iy_i$, $i = 1 \ldots 3$). A sample great circle is obtained from the intersection of $S^5$ with the plane $x_2 = y_2 = x_3 = y_3 = 0$, i.e. the plane with volume form $dx_1 \wedge dy_1$. All other two planes through the origin may be obtained by the action of $SO(6)$ on this sample two plane. As a consequence, the space of inequivalent two planes (hence great circles) is in one to one correspondence with $SO(6)/(SO(4) \times SO(2))$ and so is 8 dimensional. Note that the volume forms associated with two planes are antisymmetric matrices, i.e. adjoint elements of $SO(6)$. These $SO(6)$ adjoint elements may be identified (up to scaling) with the $SO(6)$ charge of the associated particle motions.[143] Note that the corresponding $SO(6)$ charge matrices are all of rank 2.[144]

In addition to the 8 parameters above, the phase space for particle motion on $S^5$ has two additional coordinates. The first of these is how fast the particle is moving on its given great circle, i.e. the magnitude - not just direction - of its $SO(6)$ charge. The second of these is the initial location of the particle on the great circle. It follows that the full phase space of particle motion on $S^5$ is 10 dimensional, as, of course, is clear on general grounds.

### F.1.2   $U(3)$ charges for geodesics on $S^5$

In the previous subsection we have explained that, up to $SO(6)$ rotations and time translations, geodesics on $S^5$ are characterized by the magnitude of charge.[145] Distinct geodesics with the same charge eigenvalue can be transformed into each other by $SO(6)$ rotations and time translations.

Recall that $SO(6)$ has a $U(3)$ subgroup (see Appendix H.2 for a reminder). It is interesting to analyze how the phase space of motion on an $S^5$ decomposes into orbits of this $U(3)$ subgroup. This question will be of relevance to us below, as we will be interested in the motion on a squashed $S^5$ (rather than a round $S^5$), and the squashing breaks the $SO(6)$ symmetry of the $S^5$ down to $U(3)$.

While all two planes can be rotated into each other via an $SO(6)$ transformation, all two planes cannot be rotated into each other using only $U(3)$ rotations. This is most easily understood by working with the complex combination of real coordinates, $z_i = x_i + iy_i$ in $R^6$. We have explained above that all two planes are $SO(6)$ equivalent to $dz_1 \wedge d\bar{z}_1$.[146] Let us now

---

[143]However the set of two forms associated with planes do not form a vector space: the sum of two of these two forms is not, by itself, a two form associated with a plane.

[144]This follows because the corresponding $SO(6)$ charge matrices are $SO(6)$ rotations of the sample $SO(6)$ charge, which, in turn, is clearly of rank 2.

[145]i.e. the value of the two nonzero eigenvalue (these are plus minus of each other) of the corresponding $SO(6)$ charge.

[146]Note that $dx_1 \wedge dy_1 \propto dz_1 \wedge d\bar{z}_2$.

rotate this plane by the $SO(6)$ rotation[147]

$$z_1 \to \cos \zeta z_1 + i \sin \zeta \bar{z}_2, \qquad z_2 \to \cos \zeta z_2 - i \sin \zeta \bar{z}_1. \tag{F.2}$$

Note this rotation does not lie in $U(3)$ as it mixes $z_1$ with $\bar{z}_2$. It is easily verified that under this rotation

$$dz_1 \wedge d\bar{z}_1 \to \cos^2 \zeta dz_1 \wedge d\bar{z}_1 - \sin^2 \zeta dz_2 \wedge d\bar{z}_2 - i \sin \zeta \cos \zeta (dz_1 \wedge dz_2 + d\bar{z}_1 \wedge d\bar{z}_2). \tag{F.3}$$

Recall that the adjoint of $SO(6)$ decomposes into the $9 + 3 + \bar{3}$ of $U(3)$, where the 9 is the $U(3)$ adjoint. The adjoint $U(3)$ charge of the rotated plane is obtained by restricting the charge on the RHS of (F.3) to terms of the form $dz^a \wedge d\bar{z}^b$ (see Appendix H.2). From (F.3), it follows that this adjoint charge is given by

$$\cos^2 \zeta dz_1 \wedge d\bar{z}_1 - \sin^2 \zeta dz_2 \wedge d\bar{z}_2, \tag{F.4}$$

corresponding to the $U(3)$ adjoint element

$$\begin{pmatrix} \cos^2 \zeta & 0 & 0 \\ 0 & -\sin^2 \zeta & 0 \\ 0 & 0 & 0 \end{pmatrix}. \tag{F.5}$$

Note that (F.5) is of rank 2, demonstrating that our new plane is $U(3)$ inequivalent to the original, unrotated plane $dz_1 \wedge d\bar{z}_1$ (whose charge was of rank 1).

We can now perform a $U(3)$ rotation on the plane on the RHS of (F.3). Off diagonal $U(3)$ rotations clearly change (F.5) yielding a new solution. Let us now examine the three diagonal $U(3)$ transformations. The transformation $z_3 \to e^{i\alpha} z_3$ clearly leaves (F.3) invariant. The same is true of the transformation $z_1 \to e^{i\alpha} z_1, \quad z_2 \to e^{-i\alpha} z_2$. These transformations both act trivially on the full two plane (hence they leave the geodesic unchanged). In contrast, the phase rotation $z_1 \to e^{i\alpha} z_1, \quad z_2 \to e^{i\alpha} z_2$ changes (F.3) to

$$dz_1 \wedge d\bar{z}_1 \to \cos^2 \zeta dz_1 \wedge d\bar{z}_1 - \sin^2 \zeta dz_2 \wedge d\bar{z}_2 - i \sin \zeta \cos \zeta (e^{2i\alpha} dz_1 \wedge dz_2 + e^{-2i\alpha} d\bar{z}_1 \wedge d\bar{z}_2). \tag{F.6}$$

In other words this transformation changes the two plane,[148] even though it leaves its $U(3)$ adjoint charge invariant. Thus the action of $U(3)$ on the plane with charges listed in (F.3) produces a $U(3)/(U(1) \times U(1))$ or 7 parameter set of inequivalent planes.[149] Taking the union over all values of $\zeta$, gives the full 8 parameter set of two planes.

### F.1.3 Equivalence classes on a squashed $S^5$

In the previous subsubsection, we decomposed the adjoint of $SO(6)$ into the $9, 3, \bar{3}$ of $U(3)$. In the case of motion on a squashed $S^5$, while the 9 (the $U(3)$ adjoint) continues to be a conserved charge, the 3 and the $\bar{3}$ are no longer conserved. In this case, solutions whose $U(3)$ adjoints can be rotated into each other are symmetry related, but solutions corresponding

---

[147]In terms of $R^6$ coordinates, this rotation is given by

$$\begin{aligned} x_1 &\to \cos \zeta x_1 + \sin \zeta y_2, & y_1 &\to \cos \zeta y_1 + \sin \zeta x_2, \\ x_2 &\to \cos \zeta x_2 - \sin \zeta y_1, & y_2 &\to \cos \zeta y_2 - \sin \zeta x_1. \end{aligned} \tag{F.1}$$

In other words it consists of a simultaneous rotation, by angle $\zeta$, in the $(x_1 y_2)$ and $(x_2 y_1)$ planes.

[148]Except in the special case $\zeta = 0$. When $\zeta = 0$, this transformation leaves the full $SO(6)$ charge - hence the full geodesic - unchanged.

[149]Thus our planes only have a $U(3)/(U(1) \times U(1) \times U(1))$, i.e. a 6 parameter set of inequivalent $U(3)$ charges.

$U(3)$ charges with different eigenvalues (i.e. charges with different values of $\zeta$ in (F.5)) are symmetry inequivalent.

Of course the full phase space for motion on a squashed $S^5$ is 10 dimensional. It follows from the discussion of the previous paragraph that these 10 coordinates consist of the initial location of the particle on the geodesic, the two eigenvalues of the $U(3)$ charge matrix,[150] plus the 7 parameter equivalence class of solutions with the same $U(3)$ eigenvalues.

In addition to the generic solutions, we also have a special class of solutions with $\zeta = 0$. In this case the $U(3)$ charge is of rank 1. In this case phase space is parameterized by the initial position of the particle, the value of the $U(3)$ eigenvalue, and the four parameters on the equivalence class $U(3)/U(2) \times U(1)$. It follows that this special class of solutions is 6 dimensional. These special solutions will turn out to play a key role in this paper.

# G $\quad \kappa$ symmetry analysis

In this section, we check that the embedding of the dual giant around a Gutowski-Reall black hole described in subsection 4.7 is indeed supersymmetric at the probe level using the kappa symmetry analysis.

We mostly follow Appendix C of [32]. In their notation, the AdS part of the metric of GR black hole is given by

$$ds^2_{(5)} = -f_b^2(dt_b - \omega)^2 + \frac{1}{f_b}\left[\frac{\xi}{F(\xi)}d\xi^2 + \frac{F}{\xi}(d\Phi + \eta\Psi_b)^2 + \xi\left(\frac{d\eta^2}{1-\eta^2} + (1-\eta^2)d\Psi_b^2\right)\right], \quad \text{(G.1)}$$

where $\omega = \frac{1}{3f_b}\left(\frac{F'(\xi)}{2\xi} - 1\right)$.[151] The $S^5$ part of the metric is taken to be same as in (1). The gauge field in these coordinates is given by

$$A = -f_b dt_b - \frac{3a^2 f_b}{8(1-a)^2\xi}(d\Phi + \eta\Psi_b). \quad \text{(G.2)}$$

The coordinates used above are related to the coordinates used in this paper as follows:

$$t_b = t, \qquad \eta = \cos\theta_a, \qquad \Phi = \phi_a - 2t, \qquad \Psi_b = \psi, \qquad \xi = \frac{r^2}{4}\left(1 - \frac{R_0^2}{r^2}\right). \quad \text{(G.3)}$$

The functions $f_b, F$ and the parameter $a$ in (G.1) and (G.2) are given by

$$f_b = 1 - \frac{R_0^2}{r^2}, \qquad F = \frac{1}{16}(r^2 - R_0^2)^2(1 + 2R_0^2 + r^2), \qquad a = j. \quad \text{(G.4)}$$

The ten-dimensional Vielbein chosen in [32](see equations C.41 and C.57 of that paper) in

---

[150]Recall that the most general $U(3)$ charge matrix equals (F.5) times an arbitrary scaling factor, that keeps track of how fast our particle is moving on its geodesic.

[151]Note that we have substituted $\alpha = \frac{3}{2}$ and $\tilde{m} = 0$ in eqn C.42 of [32].

these coordinates are given by[152]

$$e^0 = f_b(dt - \omega), \qquad\qquad e^1 = -\frac{1}{\sqrt{f_b}}\sqrt{\frac{\xi}{F(\xi)}}d\xi,$$

$$e^2 = \frac{1}{\sqrt{f_b}}\sqrt{\frac{F}{\xi}}(d\Phi + \eta d\Psi_b), \qquad\qquad e^3 = \sqrt{\frac{\xi}{f_b(1-\eta^2)}}d\eta$$

$$e^4 = \sqrt{\frac{\xi}{f_b}(1-\eta^2)}d\Psi_b, \qquad\qquad e^5 = d\alpha, \tag{G.5}$$

$$e^6 = \sin\alpha\cos\alpha\left(\cos^2\beta(d\phi_1 - d\phi_2) + d\phi_2 - d\phi_3\right), \qquad e^7 = \sin\alpha d\beta,$$

$$e^8 = \sin\alpha\sin\beta\cos\beta d(\phi_2 - \phi_1), \qquad\qquad e^9 = \sum_i l_i^2 d\phi_i - A.$$

The supersymmetry of the probe D-brane solution can be checked by doing $\kappa-$symmetry analysis. The supersymmetry is preserved in the presence of a D-brane in a Gutowski-Reall black hole background (which is a 1/16 BPS solution) when the background Killing spinor satisfies the following $\kappa-$symmetry condition,

$$\Gamma\epsilon = \epsilon, \tag{G.6}$$

where the Gamma matrix $\Gamma$ is the completely antisymmetric combination of the pull back of the 10 dimensional Gamma matrices on the worldvolume of the D-brane,

$$\Gamma = \frac{1}{4!}\frac{\epsilon^{\alpha_1\alpha_2\alpha_3\alpha_4}}{\sqrt{-h}}\frac{\partial X^{\mu_1}}{\partial\sigma^{\alpha_1}}\frac{\partial X^{\mu_2}}{\partial\sigma^{\alpha_2}}\frac{\partial X^{\mu_3}}{\partial\sigma^{\alpha_3}}\frac{\partial X^{\mu_4}}{\partial\sigma^{\alpha_4}}e_{\mu_1}^{M_1}e_{\mu_2}^{M_2}e_{\mu_3}^{M_3}e_{\mu_4}^{M_4}\Gamma_{M_1 M_2 M_3 M_4}. \tag{G.7}$$

Here, $\Gamma_{M_1 M_2 M_3 M_4}$ is the 4-antisymmetric combination of the flat gamma matrices in 10 dimensions.

Now we choose the following coordinates on the worldvolume of D-brane:

$$\sigma^0 = t, \qquad \sigma^i = y^i, \tag{G.8}$$

where $y^i$ are the coordinates along $S^3$ in $AdS_5$. The embedding of the D-brane is as follows(recall from subsection 4.7 that the D-brane has velocity one along the $\phi$ direction):[153]

$$(r, \phi_1, \phi_2, \phi_3, l_1, l_2, l_3) = (r_0, t, t, t, 1, 0, 0). \tag{G.9}$$

The background Killing spinor is given in [32] (Eqns C.64 and C.65),

$$\epsilon = e^{\frac{i}{2}(\phi - 2t + \phi_1 + \phi_2 + \phi_3)}\sqrt{f}\epsilon_0, \tag{G.10}$$

where the constant spinor $\epsilon_0$ satisfies the following projection conditions,

$$\Gamma^{09}\epsilon_0 = \epsilon_0, \qquad \Gamma^{12}\epsilon_0 = -i\epsilon_0, \qquad \Gamma^{34}\epsilon_0 = \Gamma^{56}\epsilon_0 = \Gamma^{78}\epsilon_0 = i\epsilon_0. \tag{G.11}$$

Out of the five projections listed above only four of them are independent since our Killing spinor also satisfies a chirality condition in 10 dimensions,

$$\Gamma^{11}\epsilon = -\epsilon. \tag{G.12}$$

---

[152]Note that $\alpha, \beta$ are just the reparameterizations of $l_1, l_2, l_3$ defined by

$$l_1 = \cos\alpha, \qquad l_2 = \sin\alpha\cos\beta, \qquad l_3 = \sin\alpha\sin\beta.$$

[153]In fact it turns out brane is supersymmetric with any choice of constant $l_1, l_2, l_3$. This is related to the fact that with the embedding (G.9), the only non-zero contribution to $\Gamma$ in (G.7) comes from the Vielbein $e^0, e^2, e^3, e^4, e^9$ and all of them are independent of $l_i$ when $\dot{\phi}_1 = \dot{\phi}_2 = \dot{\phi}_3$.

Now, we need to verify that the background Killing spinor (G.10) satisfies the $\kappa-$symmetry equation (G.6). We substitute the vielbeins (G.5) and the D-brane embedding (G.8) and (G.9) into (G.7), after which the $\kappa-$symmetry condition simplifies to,

$$\left(\Gamma^{0349} + \frac{R\sqrt{1+R^2+2R_0^2}}{R^2+R_0^2}(\Gamma^{0234} - \Gamma^{2349})\right)\epsilon_0 = \epsilon_0. \tag{G.13}$$

Using the conditions $(\Gamma^0)^2 = -1$ and $\Gamma^{09}\epsilon_0 = \epsilon_0$, we can verify that,

$$(\Gamma^0 + \Gamma^9)\epsilon_0 = 0. \tag{G.14}$$

Therefore the above condition reduces to,

$$\Gamma^{0349}\epsilon_0 = \epsilon_0, \tag{G.15}$$

which is clearly satisfied by the spinor as seen from the projections (G.11).

# H  $U(3)$ and $SO(6)$.

## H.1  Embedding of $U(3)$ inside $SO(6)$

$SO(6)$ has a $U(3)$ subalgebra. This may be seen as follows. Consider the following general Hermitian matrix (element of the algebra of $U(3)$)

$$\begin{pmatrix} a_{11} & a_{12} & a_{13} \\ (a_{12})^* & a_{22} & a_{23} \\ (a_{13})^* & (a_{23})^* & a_{33} \end{pmatrix}, \tag{H.1}$$

where the diagonal components are real. $i$ times the $U(3)$ generator (H.1) can be embedded inside the space of $6 \times 6$ antisymmetric matrices (i.e. the generator space of $SO(6)$) via the replacements $1 \to \sigma_2$, $i \to i\mathbb{I}$. Note that this replacement rule preserves the algebra of $(1, i)$.[154] Physically this embedding of $U(3)$ generators can be thought of as follows. Let the basic column vector on which $SO(6)$ acts be denoted by $(x_1, y_1, x_2, y_2, x_3, y_3)$. Define $z_i = x_i + iy_i$. The $U(3)$ defined above acts by $z_i \to U_i^j z_j$ (and with $\bar{z}$ transforming like the complex conjugate).

Completely explicitly, we find that $SO(6)$ generator ($6 \times 6$ antisymmetric matrix) corresponding to (H.1) is given by

$$\begin{pmatrix} a_{11}(i\sigma_2) & \Re(a_{12})(i\sigma_2) - \Im(a_{12})\mathbb{I}_2 & \Re(a_{13})(i\sigma_2) - \Im(a_{13})\mathbb{I}_2 \\ \Re(a_{12})(i\sigma_2) + \Im(a_{12})\mathbb{I}_2 & a_{22}(i\sigma_2) & \Re(a_{23})(i\sigma_2) - \Im(a_{23})\mathbb{I}_2 \\ \Re(a_{13})(i\sigma_2) + \Im(a_{13})\mathbb{I}_2 & \Re(a_{23})(i\sigma_2) + \Im(a_{23})\mathbb{I}_2 & a_{33}(i\sigma_2) \end{pmatrix}, \tag{H.2}$$

where $\sigma_2 = \begin{pmatrix} 0 & -i \\ i & 0 \end{pmatrix}$.

We can also go in the reverse direction. The space of $6 \times 6$ antisymmetric matrices can, of course, be viewed as a space of vectors that transform in the irreducible adjoint representation of $SO(6)$. Given the embedding (H.2), the same space can also be viewed as a space of vectors that transform in the direct sum of the adjoint, 3 and $\bar{3}$ representations[155] of $U(3)$

---

[154]Namely the fact that these numbers commute with each other, and respectively square to unity and minus unity.

[155]This may be seen as follows. The adjoint representation of $SO(6)$ may be thought of as the linear space of forms of the form $dx^i \wedge dx^j$. Let us now change basis from $x^i$ $(i = 1 \ldots 6)$ to $z^a$, $a = 1 \ldots 3$, $\bar{z}^a$, $a = 1 \ldots 3$. The space of forms is now spanned by $dz^a \wedge d\bar{z}^b$, $dz^a \wedge dz^b$ and $d\bar{z}^a \wedge d\bar{z}^b$. These basis forms, respectively, span the representations of $U(3)$ that we have called the 9, the $\bar{3}$ and 3.

(note that these are, respectively, 9 and 3 and 3 dimensional). The three representations are distinguished by their charge under the overall $U(1)$ of $U(3)$. While the 9 is neutral under this $U(1)$, the 3 and the $\bar{3}$ are, respectively, of charge $-2$ and 2. This fact may be used to separate an $SO(6)$ adjoint element into the 9, 3 and $\bar{3}$ (in practical terms this separation is most easily accomplished if we rewrite the adjoint element as a form in $z, \bar{z}$ basis).[156]

## H.2   Action of $U(3)$ on the squashed $S^5$

In this subsection, we write the Killing vectors corresponding to the $U(3)$ symmetry in $S^5$. $U(3)$ may be thought of as the group that acts on the triple column coordinates $\mathbf{z} = (z_1, z_2, z_3)$ with a unitary matrix $U$ as

$$\mathbf{z}' = U\mathbf{z}.$$

The unitarity of $U$ ensures that $\bar{\mathbf{z}}\mathbf{z} = I$ ($I$ is the $3 \times 3$ identity matrix). Infinitesimally, $U = I + iB$ where the matrix $B$ is Hermitian. The killing vector field $iV_B$ corresponding to the matrix $B$ is

$$V_B = \partial_{z_i} B_i{}^j z_j - \partial_{z_i^*}(B^*)_i{}^j z_j^* = B_i{}^j \left( z_i \partial_{z_j} - z^*{}_j \partial_{z_i^*} \right) \tag{H.5}$$

(we have used the Hermiticity of $B$). It is easily verified that this vector field annihilates $\bar{\mathbf{z}}\mathbf{z}$. The coordinates $z_i$ are related to the direction cosines and $\phi_i$ via

$$z_i = l_i e^{i\phi_i}, \tag{H.6}$$

from which it follows that[157]

$$\partial_{z_i} = -\frac{i}{z_i} \partial_{\phi_i} + \frac{l_i}{z_i} \partial_{l_i}. \tag{H.7}$$

Inserting (H.7) into (H.5), we finally obtain an explicit formula for the vector fields that generate $U(3)$ in terms of the coordinates of (1)

$$B_i{}^j e^{i(\phi_i - \phi_j)} \left( l_i \frac{\partial}{\partial l_j} - l_j \frac{\partial}{\partial l_i} - i \frac{l_i}{l_j} \frac{\partial}{\partial \phi_j} - i \frac{l_j}{l_i} \frac{\partial}{\partial \phi_i} \right). \tag{H.8}$$

$z_i$ are related to the coordinates presented above via $z_i = l_i e^{i\phi_i}$.

It is now easy to check that $\Theta + d\Psi$, $J$ and the metric on $\mathbb{CP}^2$ are invariant under $U(3)$ by taking the Lie-derivative of these objects along the $U(3)$ vector fields in (H.8).

---

[156]Another way to achieve this separation is as follows. The representation space has a natural inner product given by $\langle A_2 | A_1 \rangle = \mathrm{Tr}(A_2^\dagger A_1)$. A convenient orthonormal basis in this space includes 3 diagonal $U(3)$ elements is given by

$$u_{11} = \frac{1}{\sqrt{2}} \begin{pmatrix} i\sigma_2 & 0 & 0 \\ 0 & 0 & 0 \\ 0 & 0 & 0 \end{pmatrix}, \tag{H.3}$$

(and similar) as well as the three pairs of off diagonal elements given by

$$u_{12,r} = \frac{1}{2} \begin{pmatrix} 0 & -\mathbb{I}_2 & 0 \\ \mathbb{I}_2 & 0 & 0 \\ 0 & 0 & 0 \end{pmatrix}, \qquad u_{12,c} = \frac{1}{2} \begin{pmatrix} 0 & i\sigma_2 & 0 \\ i\sigma_2 & 0 & 0 \\ 0 & 0 & 0 \end{pmatrix}, \tag{H.4}$$

(and similar), together with any conveniently chosen basis in the space 6 dimensional symmetric space. Given an arbitrary $6 \times 6$ antisymmetric matrix $O$, we can identify its various $U(3)$ (i.e. the adjoint) elements by computing the trace

$$\mathrm{Tr}(OB^\dagger),$$

where $B$ are the various orthonormal elements described above.

[157]The RHS of (H.7) does not annihilate $\sum_i l_i^2$, and so is a vector field on all of the embedding $R^6$ rather than the sphere. This defect is remedied in the combination of $\partial_{z_i}$ presented in (H.5).

### H.3   Decomposition of $S^5$ spherical harmonics into representations of $U(3)$

$SO(6)$ spherical harmonics can be thought of as trace removed polynomials of $X^\mu$, $\mu = 1\ldots 6$. We can change basis from $X^\mu$ to $Z_i$ and $\bar{Z}^i$ ($i = 1\ldots 3$). Here $Z_i$ are $U(3)$ fundamentals, $\bar{Z}_i$ are $U(3)$ antifundamentals and the trace removal condition tells us that all contractions of $Z$ with $\bar{Z}$ must be removed.

$SO(6)$ spherical harmonics, consisting of degree $n$ trace removed polynomials, transform in the representation with $n$ boxes in the first row of the Young Tableaux, and no boxes in any other row. Moving to the $Z\bar{Z}$ basis, label polynomials by their degree $n_Z$ in $Z_i$, and their degree $n_{\bar{Z}}$ in $\bar{Z}$, with the constraint $n_Z + n_{\bar{Z}} = n$. Such polynomials transform in the $SU(3)$ Young Tableaux with $n_Z$ columns of length 1, and $n_{\bar{Z}}$ columns of length 2, and with $U(1)$ charge $n_Z - n_{\bar{Z}}$.[158] Denoting these representations by the symbol $(n_Z, n_{\bar{Z}})$, we have, in summary[159]

$$n_{SO_6} = \sum_{n_Z=0}^{n} (n_Z, n - n_Z). \tag{H.9}$$

Let us now study the Laplacian on $S^5$. As explained in Appendix K we have

$$-\nabla^2 \phi_n = n(n+4)\phi_n \tag{H.10}$$

(where $\phi_n$ is any spherical harmonic in the $n_{SO_6}$ representation). Now the metric on $S^5$ can be written as

$$ds_{S^5}^2 = ds_{CP^2}^2 + (d\Psi + \Theta)^2, \tag{H.11}$$

where $ds_{CP^2}^2$ is the metric on $CP^2$ and $\Theta$ is the one-form, whose field strength is the $U(3)$ invariant Kahler form on $CP^2$. Let us suppose that our spherical harmonic is chosen so that it transforms in the $(n_Z, n_{\bar{Z}})$ representation. If we make this choice, it follows that

$$\phi_n = e^{i(n_Z - n_{\bar{Z}})\Psi} \phi_{n_Z, n_{\bar{Z}}}.$$

Since the $S^5$ metric is a metric of the Kaluza-Klein form, and so it follows from the usual formulae of KK theory that

$$-\nabla_{S^5}^2 = (-\nabla_{CP^2,(n_Z-n_{\bar{Z}})}^2 + (n_Z - n_{\bar{Z}})^2)\phi_{n_Z, n_{\bar{Z}}}, \tag{H.12}$$

where $\nabla_{CP^2,m}^2$ is the Laplacian for a charged particle of unit charge, propagating in a $CP^2$ with magnetic field equal to $m$ times the Kahler form. $\nabla_{CP^2,m}^2$ can be thought of, roughly, as the $CP^2$ analogue of the action of the Laplacian on Monopole Spherical Harmonics with $m$ units of monopole charge.

As $-\nabla_{S^5}^2 = (n_Z + n_{\bar{Z}})(n_Z + n_{\bar{Z}} + 4)$, it follows that

$$-\nabla_{CP^2,(n_Z-n_{\bar{Z}})}^2 \phi_{n_Z, n_{\bar{Z}}} = (n_Z + n_{\bar{Z}})(n_Z + n_{\bar{Z}} + 4) - (n_Z - n_{\bar{Z}})^2$$
$$= 4(n_Z n_{\bar{Z}} + n_Z + n_{\bar{Z}}). \tag{H.13}$$

---

[158]The polynomials of $Z$ transform in the $U(3)$ representation that carry $n_Z$ columns of length 1 and $U(1)$ charge $n_Z$. The polynomials of $Z$ transform in the $U(3)$ representation that carry $n_{\bar{Z}}$ columns of length 2 and $U(1)$ charge $-n_{\bar{Z}}$. As we are not allowed to contract the $Z$ and the $\bar{Z}$, the final representation is given simply by concatenating these Young Tableaux.

[159]at least roughly, the RHS of (H.9) can also be understood as coming from the quantization of the classical phase space described in §F.1.2. Recall that this phase space is parameterized by all $U(3)$ charges given by $Q$ times the $U(3)$ matrix in (F.5) and all their $U(3)$ coadjoint orbits. $Q$ is the total number of boxes of the $SO(6)$ representation, equal here to $n$. Now the quantization of any given charge and its coadjoint orbits gives a representation labelled by the given $U(3)$ highest weights. The factors of $\cos^2$ and $-\sin^2$ in (F.5) tell us that we should distribute $n$ into eigenvalues of rotations in the two planes, such that the sum (or difference- keeping signs) of eigenvalues equals $n$. This is precisely what we do in the representations in (H.9). The $n_Z$ $Z$s are each assigned a given weight (lets say $(1,0,0)$. The $\bar{Z}$s must then be assigned a different highest weight (if we assigned the same weight, that would correspond to contraction), lets say $(0,-1,0)$. The representations in question, thus have the highest weights $(n_Z, -n_{\bar{Z}}, 0)$, in perfect agreement with classical expectations.

## I   Matching WKB solutions

In this Appendix, we match the WKB solution for the wavefunction of the dual giant computed in §5 across the turning point. The WKB solution $g(r)$ given in (163) can be re-written as follows

$$
\begin{aligned}
g(r) &= \frac{1}{(2V(r))^{1/4}}\left(A\exp(\int_{r_1}^{r} N\sqrt{2V})e^{-\beta} + B\exp(-\int_{r_1}^{r} N\sqrt{2V})e^{\beta}\right)\\
&= \frac{Ae^{-\beta}}{(2V(r))^{1/4}}\left(\exp(\int_{r_1}^{r} N\sqrt{2V}) + \frac{B}{A}e^{2\beta}\exp(-\int_{r_1}^{r} N\sqrt{2V})\right),
\end{aligned}
\tag{I.1}
$$

where

$$
\beta = \int_{r_1}^{r^*} N\sqrt{2V},
\tag{I.2}
$$

and $r_1$ is the second turning point.

Near the turning point the WKB approximation fails and hence we used the linear potential at the turning point to solve for the wavefunction exactly. The Schrodinger equation near the turning point $r_1$ is given by

$$
\left(-\frac{1}{2N^2}\frac{\partial}{\partial r} + V'(r_1)(r - r_1)\right)g(r) = 0.
\tag{I.3}
$$

The solution then very famously becomes the linear combination of Airy functions as below:

$$
g(r) = c_1 \text{Bi}\left(N^{2/3}V'(r_1)^{1/3}(r - r_1)\right) + c_2 \text{Ai}\left(N^{2/3}V'(r_1)^{1/3}(r - r_1)\right).
\tag{I.4}
$$

In the large $N^{2/3}V'(r_1)^{1/3}(r - r_1)$ limit the above solution behaves as

$$
g(r) \approx c_1 \frac{e^{\frac{2}{3}N\sqrt{V'(r_1)}r^{3/2}}}{\sqrt{\pi}\sqrt[12]{V'(r_1)}\sqrt[4]{r}} + c_2 \frac{e^{-\frac{2}{3}N\sqrt{V'(r_1)}r^{3/2}}}{2\sqrt{\pi}\sqrt[12]{V'(r_1)}\sqrt[4]{r}}.
\tag{I.5}
$$

Matching the above expression with (I.1)(after replacing $V$ by $V'(r_1)(r - r_1)$) we obtain

$$
c_1 = \frac{\sqrt{\pi}}{V'(r_1)^{1/6}}Ae^{-\beta}, \qquad c_2 = \frac{2\sqrt{\pi}}{V'(r_1)^{1/6}}Be^{\beta}.
\tag{I.6}
$$

Now taking the small $r$ limit (i.e. $a^{1/3}(r - r_1) \to -\infty$ ) of this solution, we obtain

$$
g(r) \sim \frac{Ae^{-\beta}}{(2V)^{-\frac{1}{4}}}\left(\sin\left(\int_{r_1}^{r} N\sqrt{-2V} + \frac{\pi}{4}\right) + \frac{2B}{A}e^{2\beta}\cos\left(\int_{r_1}^{r} N\sqrt{-2V} + \frac{\pi}{4}\right)\right).
\tag{I.7}
$$

In the main text, we match this solution to the solution near the horizon.

## J   LLM details

In this Appendix, we specialize the half-BPS LLM solution [23] to the case of interest in this paper i.e. 10 dimensional supergravity solution with a dual giant graviton moving along some angle in $S^5$. We consider the background solution to be $AdS_5 \times S^5$ and compute the $\frac{1}{N}$ corrections to this solution by expanding the corresponding LLM solution to leading order in $N$.

## J.1 Supergravity solution with dual giant graviton

In this subsection, we review the 1/2 BPS supergravity solution derived in [23]. In the next subsection, we will specialize to the case of interest of this paper.

The half BPS solution preserves $SO(4) \times SO(4)$ symmetry. Therefore LLM used the following Ansatz for the spacetime metric and the five-form field strength

$$ds^2 = g_{\mu\nu}dx^\mu dx^\nu + e^{H+G}d\Omega_3^2 + e^{H-G}d\tilde{\Omega}_3^2,$$
$$F_{(5)} = 4\left(F \wedge d\Omega_3 + \tilde{F} \wedge d\tilde{\Omega}_3\right),$$
(J.1)

where the indices $\mu, \nu$ run over $0, 1, 2, 3$. The dilaton and axion are constant and the three form field strengths are zero. The two forms $F$ and $\tilde{F}$ are given by

$$F = dB_t \wedge (dt + V) + B_t dV + d\hat{B},$$
$$\tilde{F} = d\tilde{B}_t \wedge (dt + V) + \tilde{B}_t dV + d\hat{\tilde{B}}.$$
(J.2)

These fields are not independent of each other, but are related via

$$F = e^{3G} \star_4 \tilde{F}$$
(J.3)

(this relationship reflects the self duality of the 10 dimensional five form field strength). By imposing the killing spinor equation, LLM found that the four dimensional metric $g_{\mu\nu}$, and the fields $B$, $\hat{B}$, $\tilde{B}$ and $\hat{\tilde{B}}$ are given by

$$ds^2 = -h^{-2}(dt + V_i dx^i)^2 + h^2(dy^2 + dx^i dx^i) + ye^G d\Omega_3^2 + ye^{-G}d\tilde{\Omega}_3^2,$$
$$B_t = -\frac{1}{4}y^2 e^{2G}, \qquad \tilde{B}_t = -\frac{1}{4}y^2 e^{-2G},$$
$$d\hat{B} = -\frac{1}{4}y^3 \star_3 \left(\frac{z+\frac{1}{2}}{y^2}\right), \qquad d\hat{\tilde{B}} = -\frac{1}{4}y^3 \star_3 \left(\frac{z-\frac{1}{2}}{y^2}\right),$$
(J.4)

where

$$z = \frac{1}{2}\tanh G, \qquad\qquad h^{-2} = 2y\cosh G,$$
$$y\partial_y V_i = \epsilon_{ij}\partial_j z, \qquad\qquad y(\partial_i V_j - \partial_j V_i) = \epsilon_{ij}\partial_y z.$$
(J.5)

As described in [23], the complete information about the half-BPS supergravity solutions lies in the boundary conditions of the function $z$ at $y = 0$. On this plane $z$ takes only two values $\pm\frac{1}{2}$. This ensures that the solution is non-singular as $y$ is taken to be zero. At non-zero values of $y$, the function $z$ is obtained by solving a Laplace's equation (see eq. 2.15 of [23]). Knowing $z$ at $y = 0$ also fixes the two dimensional vector field $V^i$ (see eq. 2.12 of [23]). We list the formulae that determine $z(x_1, x_2, y)$ and $V_i(x_1, x_2, y)$ in terms of boundary data $z(x_1, x_2, 0)$ below:

$$z(x_1, x_2, y) = \frac{y^2}{\pi}\int_{R^2} \frac{z(x_1', x_2', 0)dx_1'dx_2'}{((\mathbf{x}-\mathbf{x}')^2 + y^2)^2},$$
$$V_i = \frac{\epsilon_{ij}}{\pi}\int_{R^2} \frac{z(x_1', x_2', 0)(x^j - x'^j)dx_1'dx_2'}{((\mathbf{x}-\mathbf{x}')^2 + y^2)^2}.$$
(J.6)

The boundary conditions on $z$ at $y = 0$ are depicted by coloring the regions with $z = \frac{1}{2}$ as gray and the regions with $z = -\frac{1}{2}$ as white. The pure $AdS_5 \times S^5$ solution can be obtained by putting the boundary conditions such that $z = \frac{1}{2}$ in a circular region with radius $\rho_0$ and $z = -\frac{1}{2}$ everywhere else. In pictures, this looks like a disk of radius $\rho_0$ in an otherwise white

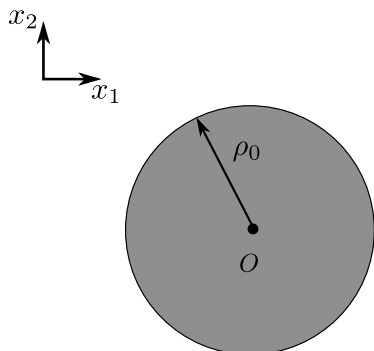

Figure 15: A figure denoting the shading of the $y = 0$ LLM plane that corresponds to a vacuum $AdS_5 \times S^5$ geometry. $z = \frac{1}{2}$ on the shaded disk centered at the origin (O), but $z = -\frac{1}{2}$ everywhere else.

region (see fig.15). The $y = 0$ surface has 5 spatial directions.[160] In the black blob, the $S^3$ in $AdS_5$ shrinks to zero size and the full blob is an $S^5$. Outside the blob, the $S^3$ in $S^5$ shrinks to zero and the five dimensional space is topologically $S^3 \times S^1 \times R$, where the $S^3$ lies in $AdS_5$, $R$ is the radial distance from the center of $AdS$, and $S^1$ is a circle in $S^5$.

With the boundary conditions described above, one gets the following metric which corresponds to metric on $AdS_5 \times S^5$:

$$\frac{ds^2}{l_N^2} = \rho_0 \left( -(1+r^2)dt^2 + \frac{dr^2}{1+r^2} + r^2 d\Omega_3^2 + d\theta^2 + \cos^2\theta d\phi^2 + \sin^2\theta d\tilde{\Omega}_3^2 \right). \qquad (J.7)$$

The coordinates in (J.4) are related to the coordinates used in the metric above as follows:

$$\begin{aligned}
\rho &= \rho_0 \sqrt{1+r^2} \cos\theta\,, \\
y &= \rho_0 r \sin\theta\,, \\
\tilde{\phi} &= \phi - t\,,
\end{aligned} \qquad (J.8)$$

where $\rho, \tilde{\phi}$ are the polar coordinates in the $y = 0$ plane.[161]

## J.2 The LLM solution for a single dual giant

We are interested in SUGRA solutions which correspond to a dual giant graviton that is localized and moving on $S^5$ and which wraps an $S^3$ in $AdS_5$. If one puts a small black blob of size $\rho_1 = \frac{1}{\sqrt{N}}$, outside the blob of size $\rho_0 = \sqrt{\frac{N-1}{N}}$,[162] it corresponds to solution with a dual giant described above (see fig. 16). The gravity solution with these boundary conditions can again be found by first finding $z$ and $V_i$ everywhere in terms of $z(x_1, x_2, 0)$.

We find that the function $z$ for this solution is given by the linear combination of the solution

---

[160]At $y = 0$, one of the $S^3$ in (J.4) shrinks to zero size. In the region with $z = \frac{1}{2}$, $S^3$ shrinks and in the region with $z = -\frac{1}{2}$, $\tilde{S}^3$ shrinks. Along with the shrinking of an $S^3$, there is one more condition, i.e. $y = 0$, hence it is a codimension 4 surface.

[161]Note that there the third transformation above has a different sign than (2.26) in [23]

[162]without loss of generality, we put the small blob on the X-axis at a distance $d$ from the center of the bigger blob

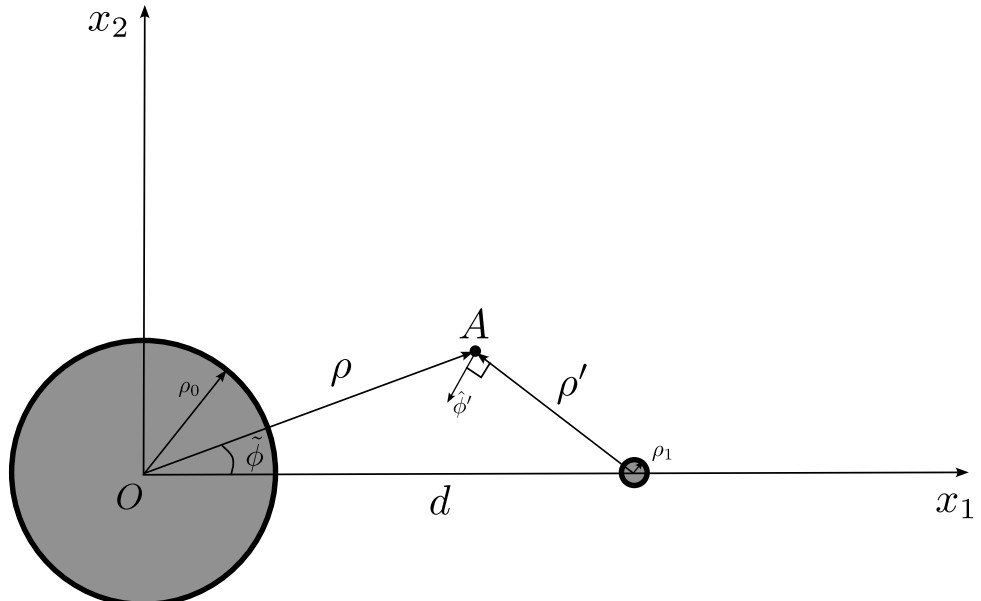

Figure 16: The boundary conditions in the $y = 0$ plane for a configuration with one dual giant graviton in $AdS$ background. The big blob of radius $\rho_0$ gives rise to pure $AdS_5 \times S^5$, the small blob of radius $\rho_1$ corresponds to one $D3$ brane at a fixed radius in $AdS$ i.e. a dual giant graviton.

for the big blob and the small blob i.e.

$$
\begin{aligned}
z &= \tilde{z}_1 + \tilde{z}_2 + \frac{1}{2}, \\
\tilde{z}_1 &= \frac{\left(\rho^2 - \rho_0^2 + y^2\right)}{2\sqrt{(\rho^2 + \rho_0^2 + y^2)^2 - 4\rho^2\rho_0^2}} - \frac{1}{2} = z_1 - \frac{1}{2}, \\
\tilde{z}_2 &= \frac{\left(\rho'^2 - \rho_1^2 + y^2\right)}{2\sqrt{\left(\rho'^2 + \rho_1^2 + y^2\right)^2 - 4\rho'^2\rho_1^2}} - \frac{1}{2} = z_2 - \frac{1}{2} \approx -\frac{(\rho_1)^2 y^2}{(\rho'^2 + y^2)^2},
\end{aligned}
\tag{J.9}
$$

where $r'$ is the distance between the center of the small blob from the point of interest. Similarly, the function $V_i$ is given by

$$
\begin{aligned}
V_1 &= V_\phi \hat{\phi} = -\frac{1}{2}\left(\frac{\rho^2 + y^2 + \rho_0^2}{\sqrt{(\rho^2 + \rho_0^2 + y^2)^2 - 4\rho^2\rho_0^2}} - 1\right)\left(-\sin\tilde{\phi}\,\hat{x}_1 + \cos\tilde{\phi}\,\hat{x}_2\right), \\
V_2 &= V'_\phi \hat{\phi}' = -\frac{1}{2}\left(\frac{\rho'^2 + y^2 + \rho_1^2}{\sqrt{(\rho'^2 + \rho_1^2 + y^2)^2 - 4\rho'^2\rho_1^2}} - 1\right)\frac{1}{\rho'}\left(-\rho\sin\tilde{\phi}\,\hat{x}_1 + (\rho\cos\tilde{\phi} - d)\hat{x}_2\right) \\
&= V_2^\rho \hat{\rho} + V_2^\phi \hat{\phi},
\end{aligned}
\tag{J.10}
$$

where $\vec{\rho}'$ is the position vector from the dual giant to the probe point and $\hat{\phi}'$ is the unit vector normal to $\vec{\rho}'$ i.e.

$$
\vec{\rho}' = \vec{\rho} - \vec{d} = (\rho\cos\tilde{\phi} - d, \rho\sin\tilde{\phi}),
\tag{J.11}
$$

and

$$
\rho' = \sqrt{\rho^2 + d^2 - 2\rho d \cos\tilde{\phi}}, \qquad \hat{\phi}' = \frac{(-\rho\sin\tilde{\phi}, \rho\cos\tilde{\phi} - d)}{\rho'}.
\tag{J.12}
$$

Therefore in $\rho$ and $\phi$ components $V_2$ takes the following form

$$V_2^\rho = \frac{d \sin \tilde{\phi}}{2} \left( \frac{\rho'^2 + y^2 + \rho_1^2}{\sqrt{(\rho'^2 + \rho_1^2 + y^2)^2 - 4\rho'^2 \rho_1^2}} - 1 \right) \frac{1}{\rho'^2} \approx \frac{(\rho_1)^2 d \sin \tilde{\phi}}{(\rho'^2 + y^2)^2},$$

$$V_2^\phi = -\frac{\rho^2 - d\rho \cos \tilde{\phi}}{2} \left( \frac{\rho'^2 + y^2 + \rho_1^2}{\sqrt{(\rho'^2 + \rho_1^2 + y^2)^2 - 4\rho'^2 \rho_1^2}} - 1 \right) \frac{1}{\rho'^2} \approx -\frac{(\rho_1)^2 (\rho^2 - d\rho \cos \tilde{\phi})}{(\rho'^2 + y^2)^2}.$$

(J.13)

Let us first take a strict large $N$ limit and check if we get pure $AdS_5 \times S^5$. In this limit, $\tilde{z}_2 = 0$ and $V_2 = 0$. Hence we get,

$$z = z_1 = \frac{1}{2} \frac{r^2 - \sin^2 \theta}{r^2 + \sin^2 \theta},$$

$$V = V_1 = \frac{-\cos^2 \theta}{r^2 + \sin^2 \theta} \hat{\phi}.$$

(J.14)

Using $V$ and $z$, we can find the functions $G$ and $h$ using the following relations:

$$h^{-2} = \frac{2y}{\sqrt{1 - 4z^2}} = \rho_0 (r^2 + \sin^2 \theta),$$

(J.15)

$$\cosh G = \frac{1}{\sqrt{4z^2 - 1}} = \frac{r^2 + \sin^2 \theta}{2r \sin \theta},$$

$$e^G = \frac{r}{\sin \theta}.$$

(J.16)

Substituting $V$, $h$ and $e^G$ in (J.4) and using (J.8) we can easily see that we get the $AdS_5 \times S^5$ metric with unit AdS length as in (J.7).

To look at the backreaction of the brane, we need to include subleading terms in $N$. Let us Taylor expand various functions appearing in the metric in $\rho_1$,

$$h^{-2} = \frac{2y}{\sqrt{1 - 4(z_1 + \tilde{z}_2)^2}} \approx \rho_0 (r^2 + \sin^2 \theta) \left( 1 - \frac{\rho_1^2}{2} \frac{(r^4 - \sin^4 \theta)}{x^4} \right),$$

$$h^2 = \frac{\sqrt{1 - 4(z_1 + \tilde{z}_2)^2}}{2y} \approx \frac{1}{\rho_0 (r^2 + \sin^2 \theta)} \left( 1 + \frac{\rho_1^2}{2} \frac{(r^4 - \sin^4 \theta)}{x^4} \right),$$

$$e^G = \sqrt{\frac{1 + 2(z_1 + \tilde{z}_2)}{1 - 2(z_1 + \tilde{z}_2)}} \approx \frac{r}{\sin \theta} \left( 1 - \frac{\rho_1^2}{2} \frac{(r^2 + \sin^2 \theta)^2}{x^4} \right),$$

$$e^{-G} = \sqrt{\frac{1 - 2(z_1 + \tilde{z}_2)}{1 + 2(z_1 + \tilde{z}_2)}} \approx \frac{\sin \theta}{r} \left( 1 + \frac{\rho_1^2}{2} \frac{(r^2 + \sin^2 \theta)^2}{x^4} \right),$$

$$V^\rho \approx \rho_1^2 \frac{d \sin(\phi - t)}{x^4},$$

$$V^\phi \approx \frac{-\cos^2 \theta}{r^2 + \sin^2 \theta} - \rho_1^2 \frac{\sqrt{1 + r^2} \cos \theta (\sqrt{1 + r^2} \cos \theta - d \cos(\phi - t))}{x^4},$$

(J.17)

where

$$x^2 = r^2 + \cos^2 \theta + d^2 - 2d \sqrt{1 + r^2} \cos \theta \cos(\phi - t).$$

(J.18)

Using the above expansions, we find corrections to the coefficients of various components of the metric. Substituting the above in (J.4), we get the following correction to pure AdS

metric:[163]

$$\frac{ds^2}{l_{N-1}^2} = r^2 d\Omega_3^2 \left(1 - \frac{\rho_1^2}{2} \frac{\left(r^2 + \sin^2\theta\right)^2}{x^4}\right) + \sin^2\theta\, d\tilde{\Omega}_3^2 \left(1 + \frac{\rho_1^2}{2} \frac{\left(r^2 + \sin^2\theta\right)^2}{x^4}\right) \tag{J.19}$$

$$-(1+r^2)dt^2\left(1 - \frac{\rho_1^2}{2} \frac{(r^2+\cos^2\theta)^2 - 4(1+r^2)\cos^2\theta - 1 + 4d\sqrt{1+r^2}\cos\theta\cos(\phi-t)}{x^4}\right)$$

$$+\frac{dr^2}{1+r^2}\left(1 + \frac{\rho_1^2}{2} \frac{r^4 - \sin^4\theta}{x^4}\right) + d\theta^2\left(1 + \frac{\rho_1^2}{2} \frac{r^4 - \sin^4\theta}{x^4}\right)$$

$$+\cos^2\theta\, d\phi^2\left(1 + \frac{\rho_1^2}{2} \frac{(r^2+\cos^2\theta)^2 - 4(1+r^2)\cos^2\theta - 1 + 4d\sqrt{1+r^2}\cos\theta\cos(\phi-t)}{x^4}\right)$$

$$-\rho_1^2 \frac{2d\sin(\phi-t)}{x^4}\left(dt(1+r^2) - \cos^2\theta\, d\phi\right)\left(\frac{r\cos\theta\, dr}{\sqrt{1+r^2}} - \sqrt{1+r^2}\sin\theta\, d\theta\right)$$

$$+2\,d\phi\, dt\,\rho_1^2 \frac{\cos\theta\left(2(1+r^2)\cos\theta - d\sqrt{1+r^2}(1+r^2+\cos^2\theta)\cos(\phi-t)\right)}{x^4} + O(\rho_1^4).$$

### J.2.1 Five-form field strength

The $\frac{1}{N}$ corrections to the five-form field strength can be found by substituting (J.9) in (J.4). Below, we list various components of the two forms $F$ and $\tilde{F}$ which appear in the five form field strength as written in (J.1).

$$F_{tr} = \frac{2\left(d^2-1\right)r\sin^4\theta\left(r^2+\sin^2\theta\right)}{2Nx^6},$$

$$F_{t\phi} = \frac{d\sqrt{r^2+1}\sin^3\theta\sin 2\theta\left(\sin^2\theta+r^2\right)^2\sin(t-\phi)}{2Nx^6},$$

$$F_{t\theta} = \frac{\sqrt{1+r^2}\sin^3\theta}{8Nx^6}\left[8\sqrt{r^2+1}\cos\theta\left(\left(d^2\left(r^2+2\right)+2r^2+1\right)\cos\theta + d^2\cos^2\theta\cos(2t-2\phi)\right)\right.$$
$$\left.-4d\cos(t-\phi)\left(\left(d^2+3r^2+2\right)\cos 2\theta + d^2 + 2r^4 + 7r^2 + 4\right)\right],$$

$$F_{r\theta} = \frac{dr\sin^3\theta\sin(t-\phi)\left(-\left(d^2+3r^2+2\right)\cos 2\theta - d^2 + 4d\sqrt{r^2+1}\cos^3\theta\cos(t-\phi) + 2r^4 + r^2\right)}{2N\sqrt{r^2+1}\,x^6},$$

$$F_{r\phi} = \frac{2r\sin^4\theta\cos\theta\left(r^2+\sin^2\theta\right)\left(d\left(\cos 2\theta + 2r^2 + 3\right)\cos(t-\phi) - 4\sqrt{r^2+1}\cos\theta\right)}{4N\sqrt{r^2+1}\,x^6},$$

$$F_{\theta\phi} = \frac{\sin^3\theta\cos\theta}{4Nx^6}\left[2\cos^2\theta\left(2d^2\left(2\left(r^2+1\right)\cos^2(t-\phi) + 2r^2 + 1\right)\right.\right.$$
$$+\sin^2\theta\left(4d^2 + 3r^4 + 2r^2 - 1\right) + 2\left(2r^2+1\right)\sin^4\theta\right)$$
$$+2d\sqrt{r^2+1}\cos(t-\phi)\cos\theta\left(2\cos^2\theta\left(2\cos 2\theta - 5\left(r^2+1\right)\right)\right.$$
$$\left.-\left(2\left(d^2 - r^4\right) + \sin^2\theta\left(-\cos 2\theta + 2r^2 + 1\right)\right)\right)$$
$$-\frac{1}{4}\left(r^2+1\right)\left(16r^4 - 8r^2\cos 2\theta + \left(r^2+1\right)\cos 4\theta + 7r^2 - 1\right.$$
$$\left.\left.+4\cos^4\theta\left(-\cos 2\theta + 2r^2 + 2\right)\right)\right], \tag{J.20}$$

---

[163]we have used the fact that $l_N^2 \rho_0 = l_{N-1}^2$ and put the leading order value of $\rho_0 \approx 1 + O(\rho_1^2)$ in the terms proportional to $\rho_1^2$.

where $x$ is defined in (J.18). The components of $\tilde{F}_{\mu\nu}$ are given by

$$
\begin{aligned}
\tilde{F}_{tr} &= \frac{r^3}{32Nx^6}\Big[\big(48d^2\big(r^2+1\big)-16r^4+17\big)\cos 2\theta - \cos 6\theta + 2\big(6r^2+5\big)\cos 4\theta \\
&\quad +2\big(-8\big(4d^2+3\big)r^4-6\big(4d^2+1\big)r^2 \\
&\quad +2d\cos\theta\big(\sqrt{r^2+1}\cos(t-\phi)\big(-8d^2+\cos 4\theta - 8\big(2r^2+3\big)\cos 2\theta + 24\big(r^4+r^2\big)-1\big) \\
&\quad +8d\big(r^2+1\big)\cos\theta\cos(2t-2\phi)\big)+8d^2-16r^6+3\big)\Big], \\
\tilde{F}_{t\theta} &= -\frac{4r^4\sin\theta\big(r^2+\sin^2\theta\big)\big(4\big(r^2+1\big)\cos\theta - d\sqrt{r^2+1}\big(\cos 2\theta + 2r^2+3\big)\cos(t-\phi)\big)}{8Nx^6}, \\
\tilde{F}_{t\phi} &= -\frac{dr^4\sqrt{r^2+1}\cos\theta\big(\sin^2\theta+r^2\big)^2\sin(t-\phi)}{Nx^6}, \\
\tilde{F}_{r\theta} &= \frac{-dr^3\sin\theta\sin(t-\phi)}{8N\sqrt{r^2+1}x^6}\Big[8d\big(r^2+1\big)\big(d-2\big(r^2+1\big)^{1/2}\cos\theta\cos(t-\phi)\big) \\
&\quad -\cos 4\theta + 4\big(3r^2+2\big)\cos 2\theta + 4r^2+1\Big], \\
\tilde{F}_{r\phi} &= -\frac{r^3\cos\theta}{8Nx^6}\Big[\frac{\cos(t-\phi)\big(d\big(8d^2\big(r^2+1\big)+\cos 4\theta + 4\big(3r^2+4\big)\cos 2\theta + 4r^2+7\big)\big)}{\sqrt{r^2+1}} \\
&\quad -4\cos\theta\big(\big(d^2+2\big)\cos 2\theta - 2d^2\big(r^2+1\big)\cos(2t-2\phi)+3d^2\big)\Big], \\
\tilde{F}_{\theta\phi} &= \frac{2\big(d^2-1\big)r^4\sin(2\theta)\big(-\cos(2\theta)+2r^2+1\big)}{8Nx^6}.
\end{aligned}
$$

$$(J.21)$$

## K  Spherical harmonics on $S^5$

In this section, we compute the $SO(4)$ invariant scalar spherical harmonics on $S^5$. The $S^5$ metric is given by

$$ds^2 = d\theta^2 + \cos^2\theta\, d\phi^2 + \sin^2\theta\, d\Omega_3^2. \tag{K.1}$$

Since we are looking for $SO(4)$ invariant spherical harmonics, these are only functions of $\theta$ and $\phi$ and satisfy the following differential equation

$$\left(\frac{1}{\sin^3\theta\cos\theta}\frac{\partial}{\partial\theta}\left(\sin^3\theta\cos\theta\frac{\partial}{\partial\theta}\right)+\frac{\partial^2}{\partial\phi^2}\right)Y(\theta,\phi)=-k(k+4), \tag{K.2}$$

where $k$ is an integer. Using

$$Y(\theta,\phi)=Y_{k,m}(\theta)e^{im\phi}, \tag{K.3}$$

we obtain

$$\frac{\partial^2 Y_{k,m}(\theta)}{\partial\theta^2}-\frac{m^2 Y_{k,m}(\theta)}{\cos^2\theta}+\left(\frac{3\cos^2\theta-\sin^2\theta}{\sin\theta\cos\theta}\right)\frac{\partial Y_{k,m}(\theta)}{\partial\theta}=-k(k+4). \tag{K.4}$$

The solution to the above equation is given by

$$
\begin{aligned}
Y_{k,m}(\theta) &= a(i\cos\theta)^{-m}\,_2F_1\left(-\frac{k}{2}-\frac{m}{2},\frac{k}{2}-\frac{m}{2}+2;1-m;\cos^2\theta\right) \\
&\quad + b(i\cos\theta)^m\,_2F_1\left(\frac{m}{2}-\frac{k}{2},\frac{k}{2}+\frac{m}{2}+2;m+1;\cos^2(\theta)\right).
\end{aligned}
\tag{K.5}
$$

For $m > 0$, imposing regularity at $\theta = \frac{\pi}{2}$ sets $a = 0$ and we obtain

$$Y_{k,m}(\theta) = b(i\cos\theta)^m\,_2F_1\left(\frac{m}{2}-\frac{k}{2},\frac{k}{2}+\frac{m}{2}+2;m+1;\cos^2\theta\right). \tag{K.6}$$

Near $\theta = 0$, the above solution takes the following form

$$\lim_{\theta \to 0} Y_{k,m}(\theta) = b \frac{i^m \Gamma(m+1)}{\theta^2 \Gamma\left(\frac{m-k}{2}\right)\Gamma\left(\frac{1}{2}(k+m+4)\right)} . \tag{K.7}$$

Regularity at $\theta = 0$ imposes $k \geq m$ and $m = k - 2\mathbb{Z}$. Finally, the coefficient $b$ can be fixed by imposing

$$\frac{2}{\pi} \int d\theta d\phi \, \sin^3 \theta \cos\theta \, Y(\theta, \phi) Y^*(\theta, \phi) = \delta_{k,k'} \delta_{m,m'} . \tag{K.8}$$

A similar analysis can be performed for $m < 0$, we finally obtain

$$Y(\theta, \phi) = b(i\cos\theta)^{|m|} {}_2F_1\left(\frac{|m|}{2} - \frac{k}{2}, \frac{k}{2} + \frac{|m|}{2} + 2; |m| + 1; \cos^2\theta\right) e^{im\phi} . \tag{K.9}$$

The spherical harmonic with $m = k$ takes the following simple form

$$Y_{k,k}(\theta, \phi) = \sqrt{\frac{(k+1)(k+2)}{2}} (i\cos\theta)^k e^{ik\phi} . \tag{K.10}$$

## K.1 Expectation value of $\mathrm{Tr}(Z^n)$

As explained in §6.5, we can compute the expectation values of the chiral primary operators from the asymptotic behavior of the part of the five-form field strength that has all legs on $S^5$. Using (218) and (245), we compute the $\frac{1}{N}$ expansion of the scalar function $\chi$ and it is given by

$$\begin{aligned}
\chi = 4\Bigg(-1 - \frac{(\cos 2\theta - 2r^2 - 1)}{64N\sqrt{r^2+1}x^6}\Bigg[&4\cos 2\theta\left(\sqrt{r^2+1}\left(5d^2+10r^2+6\right)-6d\left(r^2+1\right)\cos\theta\cos(t-\phi)\right) \\
&+\sqrt{r^2+1}\left(8\left(5d^2+2\right)r^2+28d^2+8r^4+23\right) \\
&-24d\left(2r^4+5r^2+3\right)\cos\theta\cos(t-\phi)+\sqrt{r^2+1}\cos 4\theta\Bigg]\Bigg).
\end{aligned} \tag{K.11}$$

In the large $r$ expansion of the function $\chi$, the $\frac{1}{r^k}$ piece appear with the $k^{th}$ (and lower) scalar spherical harmonics. Keeping only the $\frac{1}{N}$ piece, at leading order in $\frac{1}{r}$, we obtain

$$\begin{aligned}
\chi(\theta, \phi) &\approx 4\left(\frac{-6\cos^2\theta\left(8d^2\cos^2(t-\phi)+1\right)+8d^2+15\cos 2\theta+7}{16Nr^2}\right) + O\left(\frac{1}{r^3}\right) \\
&\approx \frac{1}{r^2}\left(\frac{d^2\sqrt{6}}{2}\left(Y_{2,2}(\theta,\tilde{\phi})+Y_{2,-2}(\theta,\tilde{\phi})\right)+\sqrt{2}\left(d^2-1\right)Y_{2,0}(\theta,\tilde{\phi})\right)+O\left(\frac{1}{r^3}\right),
\end{aligned} \tag{K.12}$$

where $\tilde{\phi} = \phi - t$. As expected, the $\frac{1}{r^2}$ term appears with the $k = 2$ spherical harmonics and hence contains the information of the operators quadratic in $X, Y$ and $Z$. The expectation value of the operator $\mathrm{Tr}(Z^2)$ is proportional to $\frac{d^2\sqrt{6}}{2}$ (since it has charge $m = 2$) and the expectation value of $\mathrm{Tr}(X^2 + Y^2)$ is proportional to $\sqrt{2}(d^2-1)$. The expectation values of other chiral primary operators can be found by integrating $\chi$ with the orthogonal spherical harmonics.

The expectation value of $\mathrm{Tr}(Z^k)$ is given by

$$s^I = \frac{1}{k(k+4)}\left(\frac{2}{\pi}\int d\theta d\phi \, \sin^3\theta \cos\theta \, Y_{k,k}^*(\theta,\phi)\chi(\theta,\phi)\right). \tag{K.13}$$

To perform the $\phi$ integral, we define $z = e^{-i\phi}$ and perform the complex integral by computing the residue at the poles. The poles appear when $x$ defined in (215) vanishes i.e.

$$z_\pm = \frac{\sec\theta\left(2d^2 + \cos 2\theta + 2r^2 + 1 \pm \sqrt{(2d^2 + \cos 2\theta + 2r^2 + 1)^2 - 16d^2(r^2+1)\cos^2\theta}\right)}{4d\sqrt{r^2+1}}.$$

At large $r$, only the $z_-$ pole lies inside the contour. After performing the $\phi$ integral, at large $r$, we obtain

$$s^I = -\frac{4i^k}{N\,k(k+4)}\left(\int d\theta\left(\sqrt{2(k+1)(k+2)}\,(k^2-1)\sin^3\theta\cos^{k+1}\theta\right)\left(\frac{d\cos\theta}{r}\right)^k\right). \quad \text{(K.14)}$$

We finally perform $\theta$ integral to obtain

$$s^I = -\frac{4i^k}{N\,k(k+4)}\left(\frac{(k^2-1)}{\sqrt{2(k+1)(k+2)}}\right)\left(\frac{d}{r}\right)^k. \quad \text{(K.15)}$$

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
