# Peer review of "Dual Dressed Black Holes as the end point of the Charged Superradiant instability in N = 4 Yang Mills"

_SciPost Physics, doi:SciPost Phys. 18, 137 (2025)_

## Round 1 · Referee Report · Anonymous (Referee 1) · 2025-1-26

Strengths

Constructs novel solutions of black holes + dual giant gravitons in string theory on $AdS_5\times S^5$, analyzes them in detail and argues that they dominate the micro-canonical ensemble in some range of energies

Report

The paper discusses charged black holes in type IIB string theory in $AdS_5\times S^5$, and argues that in some range of energies they are unstable towards emitting dual giant gravitons, and that the generic solutions in this range of energies involve a black hole together with one, two or three dual giant gravitons that are far from it in the radial direction. Previous resolutions of the instability by "hairy black hole" solutions of 5d SUGRA are argued to be irrelevant. The new proposed solutions are analyzed in great detail, and shown to be under computational control (for a large range of energies) thanks to the large separation between the black hole and the dual giant gravitons (that justifies treating them as probes). One interesting future application of these solutions will be to shed light on the range of values for which 1/16-BPS states exist in this theory. The paper is very nice and clear, it explains the dynamical and thermodynamical instabilities of the various configurations clearly, and I am happy to recommend its publication.

There are two minor changes listed below which the authors may want to consider before the publication.

Requested changes

1) In page 5 the authors say that they address the fate of the black holes in type IIB SUGRA, but in fact they address it in the full type IIB string theory, since the dual giant gravitons are D-branes that are not well-described just in type IIB SUGRA. The authors explain clearly why a 5d SUGRA analysis is definitely not enough since higher KK modes are involved, but it seems to me that a 10d SUGRA analysis is also not enough since it does not include the dual giant gravitons. The authors should consider rephrasing some of the statements on this page to clarify that their analysis uses the full type IIB string theory and not just type IIB SUGRA. (The authors mention on page 11 that SUGRA is not enough, but there is no reason not to say this already on page 5.) And also in other places they refer to the solution with 1,2,3 dual giant gravitons as a "supergravity solution", even though it involves extra D-branes beyond supergravity.

2) Somewhat related to this, the first paragraph of page 11 is not completely clear to me. In this paragraph the computation of a non-zero VEV for $tr(Z^n)$ is presented as evidence that the solution cannot be found just within 5d SUGRA. As mentioned above, this is clear anyway since the solution is not even in 10d SUGRA; but I am not sure if just the non-zero VEVs are enough to argue that the solution is not captured by a consistent trunction to 5d SUGRA, since the uplift of solutions from 5d SUGRA to 10d SUGRA is non-linear, and a priori I could imagine that it turns on non-zero VEVs for all $tr(Z^n)$ operators, even if the corresponding fields are not part of the 5d SUGRA. Of course, it is highly plausible that the 10d solution indeed cannot be lifted from 5d, but it is not clear to me if the VEVs themselves give evidence for this; if the authors want to keep this statement, perhaps they can give more justifications for why such VEVs cannot arise by uplifting 5d SUGRA solutions.

Recommendation

Publish (surpasses expectations and criteria for this Journal; among top 10%)

  • validity: top
  • significance: high
  • originality: top
  • clarity: top
  • formatting: excellent
  • grammar: excellent

Author:  Diksha Jain  on 2025-03-22  [id 5305]

(in reply to Report 1 on 2025-01-26)

Dear Referee

Thanks a lot for your questions/comments, we address them below :

Point 1: We thank the referee for pointing out our imprecise wording here. In the revised version we
changed the term 'supergravity solutions’ to `bulk solutions’ all through the paper (including in the
abstract).

Point 2: We thank the referee for this point. We have added the new footnote number 22 to address this
point. This footnote reads
Gauged supergravity is believed to form a consistent truncation of the full 10 d supergravity in the
following sense. If we decompose 10 d supergravity into 5d fields, denote those fields that lie in gauged
supergravity a $a$ type fields, and those fields that lie outside gauged supergravity as $b$ type fields,
then it is believed that all couplings between $a$ and $b$ type fields are of quadratic or higher order in
$b$. As a consequence, a solution of gauged supergravity never sources $b$ type fields including all those
dual to $Tr (Z^n)$ for all $n \geq 3.$ .
We hope this is sufficient.

---

## Round 1 · Referee Report · Anonymous (Referee 2) · 2025-2-14

Strengths

  1. A nice demonstration of the instability of previously known SUGRA solutions.
  2. A novel proposal for the phase in the mu>1 part of the phase diagram of N=4 SYM.
  3. A thorough exploration of the resulting phase diagram.
  4. A construction, using reasonable approximations, of the backreacted solutions.

Weaknesses

While the generality of the paper is impressive, it makes it somewhat imposing, and it is at times hard to extract the truly important ideas. (I admit I have not had the time to go through it all in detail.) In the future, I would recommend the authors to consider a shorter companion paper, going through the main results in the simplest possible case only (perhaps all charges equal and zero ang. mom.).

Report

Understanding the phase diagrams of gauge theories at non-zero charge density (or non-zero chemical potential) is an important question with several pressing applications. Holographic duality has for the last few decades been an interesting tool for physicists interested in this question, with the added benefit of connecting with interesting problems of gravitational physics.

Many interesting results have come out of such work. However, top-down holographic theories are complicated beasts, even in the supergravity limit. In order to construct solutions, attention is usually focused to small truncations of the full theory. Even if the resulting solutions are stable within the truncation, they might not be within the full supergravity theory. And even if that is the case, there remains the stringy sectors not accounted for by SUGRA.

The present paper attempts to account for such stringy instabilities. It gives a nice thermodynamic argument for the instability of previously known SUGRA solutions dual to finite-density N=4 SYM. It argues that the end state of this instability is represented by a gravity solution formed by a central BH with mu=1 plus a small number of D3-branes (dual giant gravitons) at large radius, carrying a large fraction of the total R-charge. It goes on to discuss the thermodynamics of such solution, and the resulting phase diagram, using the approximation that the central BH and the D3-branes weakly interact, so can be treated independently.

It goes on to discuss a probe analysis of such D3-branes, and quantizes these probe branes to study how they tunnel to the global minimum of their effective potential. The backreaction is then analyzed using results from Lin, Lunin, and Maldacena.

The paper provides an important step towards accounting for stringy instabilities in holographic duals of gauge theories at finite charge density. I am happy to recommend it for publication in this journal. I include a few proposed (very) minor revisions.

Requested changes

  1. Under (48), "...mu_i is an increasing function of Q_i" should be "...of Q_i^{BH}" I think.
  2. The analysis in sec. 3 assumes from the beginning that the number of dual GG's is small (compared to N) and that they are at large radius, motivating the weak coupling to the central black hole. This is fine for proving the existence of the instability and for discussing the solutions in the paper. But as far as I can tell this leaves open the possibility that there are other solutions with an order N number of branes sharing the extra charge. I think this would be worth to note more clearly already in the beginning of that section.
  3. On page 26, the authors note that in the planer limit, the decay must happen through bubble nucleations. It might be worth pointing out that what in their paper is ref. 14 (as well as the recent https://arxiv.org/abs/2411.19667) studies exactly such bubble nucleation in the probe limit, albeit in a different holographic theory.
  4. It is said throughout the paper that mu>1 BH's are thermodynamically unstable. To clarify, are they "locally" unstable or metastable? The effective potential analysis, and e.g. the discussion about bubble nucleation, make it seem like it must be metastable?

Recommendation

Publish (easily meets expectations and criteria for this Journal; among top 50%)

  • validity: high
  • significance: high
  • originality: high
  • clarity: high
  • formatting: excellent
  • grammar: excellent

Author:  Diksha Jain  on 2025-03-22  [id 5306]

(in reply to Report 2 on 2025-02-14)

Dear Referee

Thanks a lot for your comments. We address them below:
Point 1: We thank the referee for pointing this out. We have corrected the typo.
Point 2: We thank the referee for this point. We have added footnote 39 to address this point. We
reproduce the text of that footnote
The analysis presented in this subsection applies whenever $\frac{m}{N}$ is small. This is certainly the
case when $m$ is of order unity, as assumed in the main text in the rest of this subsection. However it is
also the case when $m=\zeta N$ with $\zeta \ll 1$ (see section 7 for a discussion of bulk solutions
corresponding to this case). The probe analysis of the rest of this section (see (50)) tells us that
the reduction of flux at the centre of the black hole results in a fractional lowering of entropy of order
${\cal O}(\zeta)$. The analysis of 6.8.2 then tells us that interaction effects correct probe
estimates at order
$\frac{\zeta}{d^4} = {\cal O}(\zeta^3)$, and so are negligible at small $\zeta$. This discussion strongly
suggests that the solutions described in this paper maximize entropy locally (in configuration space). We
emphasize that nothing presented in this paper rules out the possible existence of new nonlinear solutions
of supergravity that
have even higher entropy than the solutions presented in this paper. For instance, we cannot rule out the
possibility that thermodynamics is dominated by a new nonlinear solution - which can be, in some sense,
thought of as `$\zeta$ of order unity'.
We hope this clarifies the status of our claims here.
Point 3: We thank the referee for alerting us to these papers. We have added citations to these papers in
our manuscript.
Point 4: The referee is correct that the instability studied in this paper is of the tunneling variety, and we
thank them for suggesting that this be emphasized. We have added a footnote 10 to alert the reader to this
point. The text of this footnote is reproduced below
As we explain in
section 5, this instability involves tunneling through a barrier. Consequently, it is perhaps more
accurate to say that black holes with $\mu_i>1$ are metastable in the microcanonical ensemble. We
expect the same black holes to have a simpler `roll down the hill' instability in the grand canonical
ensemble (see 3.2).

---

## Round 1 · Referee Report · Anonymous (Referee 3) · 2025-2-17

Report

This paper identifies a new class of black holes in AdS space, termed dual dressed black holes, and proposes that they serve as the endpoints of the superradiant instabilities of charged black holes. The previously known endpoints, identified as hairy black holes, were argued by the authors to be unstable in the ten-dimensional IIB supergravity, though they may remain stable in lower-dimensional gauged supergravity. The author provides a thorough discussion of the dual dressed black hole, including classical and quantum analyses in the probe limit, as well as the backreacted solutions in IIB supergravity. Given the significance of this discovery, I strongly recommend this paper for publication in SciPost Physics.

Requested changes

(The equation numbers and line numbers refer to those in the version scipost_202501_00001v1.)

  1. In Section 3.4, it may be helpful for readers to include a phase diagram on a two-dimensional plane with fixed $Q_2$ and $Q_3$, similar to the one in Figure 2.
  2. In equations (90) and (96), there should not be an explicit $\epsilon$ in the first term. Even if the author chooses to retain $\epsilon$, the power of $\epsilon$ in the denominator should be 1, not 2.
  3. In line 840, the “mass to energy ratio” should be “charge to energy ratio”.
  4. In equation (96), an overall factor of $N$ is missed.

Section 5 is not very well-written and is somewhat difficult for me to follow. I have noticed several typos/errors: 1. In line 990, it should be $V_{cl}(r) = NV(r)+\omega$. 2. In line 1099, the local wave number should be $N\sqrt{-2V}$. 3. In equations (154) and (155), the $\epsilon$ should be $\epsilon / N$. 4. In lines 1149 and 1306, the reference to (5.4) is incorrect. 5. In line 1181, the $\omega^q_*$ should be $\omega_*$. 6. In equations (187) and (188), $W’(r)$ should be $W’(R)$.

In line 1016, I do not quite understand why $V_{-2}$ is of order $\frac{\min(\frac{\omega}N,\frac 1N)^2}{q_n^2}$. From (128), I thought it should be of order $\frac{\max(\frac{\omega}N,\frac 1N)^2}{q_n^2}$ instead.

Recommendation

Publish (surpasses expectations and criteria for this Journal; among top 10%)

  • validity: top
  • significance: top
  • originality: top
  • clarity: good
  • formatting: -
  • grammar: -

Author:  Diksha Jain  on 2025-03-22  [id 5307]

(in reply to Report 3 on 2025-02-17)

Dear Referee

Thanks a lot for your comments. We address them below:
Point 1: We thank the referee for this suggestion. We have added 4 new figures, Figs 2, 3, 11 and 12
in response to this suggestion.

Point 2: We thank the referee for pointing this out. We have fixed the typo.
Point 3: We thank the referee for pointing this out. We have fixed the typo.
Point 4: We thank the referee for pointing this out. We have fixed the typo.
Point 5: We like to especially thank the referee, for their careful reading of section 5, for
detecting several mis prints and typos (we have fixed all of them), and especially for
correctly pointing out that the first subsection of section 5 was confusingly written.
In response to the referees criticism, we have rewritten parts of section 5.1 to make this
section clearer and easier to read. We feel this exercise has improved the presentation in our
paper and would like, again, to express our gratitude to the referee for prompting us to do this.

---

## Round 2 · Author Response

In the resubmitted version, we made some changes to address the comments made by the referees.

---

## Round 2 · List of Changes

1. In the revised version we changed the term supergravity solutions’ tobulk solutions’ all through the paper (including in the abstract).
  2. We have added the new footnote number 22 . This footnote reads "Gauged supergravity is believed to form a consistent truncation of the full 10 d supergravity in the following sense. If we decompose 10 d supergravity into 5d fields, denote those fields that lie in gauged supergravity a $a$ type fields, and those fields that lie outside gauged supergravity as $b$ type fields, then it is believed that all couplings between $a$ and $b$ type fields are of quadratic or higher order in $b$. As a consequence, a solution of gauged supergravity never sources $b$ type fields including all those dual to $Tr (Z^n)$ for all $n \geq 3.$ .
  3. We have added footnote 39 which reads "The analysis presented in this subsection applies whenever $\frac{m}{N}$ is small. This is certainly the case when $m$ is of order unity, as assumed in the main text in the rest of this subsection. However it is also the case when $m=\zeta N$ with $\zeta \ll 1$ (see \S \ref{disc} for a discussion of bulk solutions corresponding to this case). The probe analysis of the rest of this section (see \eqref{feng}) tells us that the reduction of flux at the centre of the black hole results in a fractional lowering of entropy of order ${\cal O}(\zeta)$. The analysis of \S \ref{estsub} then tells us that interaction effects correct probe estimates at order $\frac{\zeta}{d^4} = {\cal O}(\zeta^3)$, and so are negligible at small $\zeta$. This discussion strongly suggests that the solutions described in this paper maximize entropy locally (in configuration space). We emphasize that nothing presented in this paper rules out the possible existence of new nonlinear solutions of supergravity that have even higher entropy than the solutions presented in this paper. For instance, we cannot rule out the possibility that thermodynamics is dominated by a new nonlinear solution - which can be, in some sense, thought of as `$\zeta$ of order unity'."
  4. We have added a footnote 10 which reads " As we explain in \S \ref{quant}, this instability involves tunneling through a barrier. Consequently, it is perhaps more accurate to say that black holes with $\mu_i>1$ are metastable in the microcanonical ensemble. We expect the same black holes to have a simpler `roll down the hill' instability in the grand canonical ensemble (see \S \ref{thermins}).
  5. We have added 4 new figures, Figs 2, 3, 11 and 12.
  6. We have rewritten parts of section 5.1 to make this section clearer and easier to read.
  7. We fixed various typos.

---

## Editorial Decision

published